# CAUCHY-SCHWARZ DIVERGENCE INFORMATION BOTTLENECK FOR REGRESSION

**Shujian Yu**[1,3]    **Xi Yu**[2]    **Sigurd Løkse**[4]    **Robert Jenssen**[3,6]    **Jose C. Principe**[5]

[1]Vrije Universiteit Amsterdam    [2]Brookhaven National Laboratory
[3]UiT - The Arctic University of Norway    [4]NORCE Norwegian Research Centre
[5]University of Florida    [6]University of Copenhagen
s.yu3@vu.nl; xyu1@bnl.gov; sigl@norceresearch.no
robert.jenssen@uit.no; principe@cnel.ufl.edu

## ABSTRACT

The information bottleneck (IB) approach is popular to improve the generalization, robustness and explainability of deep neural networks. Essentially, it aims to find a minimum sufficient representation $\mathbf{t}$ by striking a trade-off between a compression term $I(\mathbf{x}; \mathbf{t})$ and a prediction term $I(y; \mathbf{t})$, where $I(\cdot; \cdot)$ refers to the mutual information (MI). MI is for the IB for the most part expressed in terms of the Kullback-Leibler (KL) divergence, which in the regression case corresponds to prediction based on mean squared error (MSE) loss with Gaussian assumption and compression approximated by variational inference. In this paper, we study the IB principle for the regression problem and develop a new way to parameterize the IB with deep neural networks by exploiting favorable properties of the Cauchy-Schwarz (CS) divergence. By doing so, we move away from MSE-based regression and ease estimation by avoiding variational approximations or distributional assumptions. We investigate the improved generalization ability of our proposed CS-IB and demonstrate strong adversarial robustness guarantees. We demonstrate its superior performance on six real-world regression tasks over other popular deep IB approaches. We additionally observe that the solutions discovered by CS-IB always achieve the best trade-off between prediction accuracy and compression ratio in the information plane. The code is available at `https://github.com /SJYuCNEL/Cauchy-Schwarz-Information-Bottleneck`.

## 1 INTRODUCTION

The information bottleneck (IB) principle was proposed by (Tishby et al., 1999) as an information-theoretic framework for representation learning. It considers extracting information about a target variable $y$ through a correlated variable $\mathbf{x}$. The extracted information is characterized by another variable $\mathbf{t}$, which is (a possibly randomized) function of $\mathbf{x}$. Formally, the IB objective is to learn a representation $\mathbf{t}$ that maximizes its predictive power to $y$ subject to some constraints on the amount of information that it carries about $\mathbf{x}$:

$$\max_{p(\mathbf{t}|\mathbf{x})} I(y; \mathbf{t}) \quad \text{s.t.} \quad I(\mathbf{x}; \mathbf{t}) \leq R, \tag{1}$$

where $I(\cdot; \cdot)$ denotes the mutual information. By introducing a Lagrange multiplier $\beta > 0$, $\mathbf{t}$ is found by optimizing a so-called IB Lagrangian (Gilad-Bachrach et al., 2003; Shamir et al., 2010):

$$\min_{p(\mathbf{t}|\mathbf{x})} -I(y; \mathbf{t}) + \beta I(\mathbf{x}; \mathbf{t}). \tag{2}$$

Maximizing $I(y; \mathbf{t})$ ensures the sufficiency of $\mathbf{t}$ to predict $y$, whereas minimizing $I(\mathbf{x}; \mathbf{t})$ encourages the minimality (or complexity) of $\mathbf{t}$ and prevents it from encoding irrelevant bits. The parameter $\beta$ controls the fundamental tradeoff between these two information terms. In this sense, the IB principle also provides a natural approximation of minimal sufficient statistic (Gilad-Bachrach et al., 2003).

Traditionally, the IB principle and its variants (e.g., (Strouse & Schwab, 2017; Creutzig et al., 2009)) have found applications in document clustering (Slonim & Tishby, 2000), image segmentation (Bardera et al., 2009), biomolecular modeling (Wang et al., 2019), etc. Recent studies have

established close connections between IB and DNNs, especially in a supervised learning scenario. In this context, $\mathbf{x}$ denotes input feature vectors, $y$ denotes desired response such as class labels, and $\mathbf{t}$ refers to intermediate latent representations or activations of hidden layers. Theoretically, it was observed that the layer representation $\mathbf{t}$ undergoes two separate training phases: a fitting or memorization phase in which both $I(\mathbf{x}; \mathbf{t})$ and $I(y; \mathbf{t})$ increase, and a compression phase in which $I(\mathbf{x}; \mathbf{t})$ decreases while $I(y; \mathbf{t})$ continues to increase or remains consistent (see (Shwartz-Ziv & Tishby, 2017; Saxe et al., 2018; Chelombiev et al., 2019; Yu et al., 2020; Lorenzen et al., 2022) for a series of work in this direction; although the argument itself is still under a debate). The existence of compression also provides new insights to the generalization behavior of DNNs (Wang et al., 2022; Kawaguchi et al., 2023). Practically, the intermediate representations learned with the IB objective have been demonstrated to be more robust to adversarial attacks (Wang et al., 2021; Pan et al., 2021) and distributional shift (Ahuja et al., 2021). In a parallel line of research (Bang et al., 2021; Kim et al., 2021), the IB approach has been leveraged to identify the most informative features (to a certain decision) by learning a differentiable mask $m$ on the input, i.e., $\mathbf{t} = \mathbf{x} \odot m$, in which $\odot$ refers to element-wise product.

Unfortunately, optimizing the IB Lagrangian remains a challenge due to its computational intractability. Although scalable methods of IB are feasible thanks to variational bounds of mutual information (Alemi et al., 2017; Kolchinsky et al., 2019b; Poole et al., 2019) as well as Gaussian or discrete data assumptions (Chechik et al., 2003; Tishby et al., 1999), the choice of such bounds, the imposed data distributional assumptions, as well as specific details on their implementations may introduce strong inductive bias that competes with the original objective (Ngampruetikorn & Schwab, 2023).

In this paper, we propose a new method for performing nonlinear IB on arbitrarily distributed $p(\mathbf{x}, y)$, by exploiting favorable properties of the Cauchy-Schwarz (CS) divergence (Principe et al., 2000; Yu et al., 2023). We focus our attention on the regression setup, which is far less investigated than classification, except for (Ngampruetikorn & Schwab, 2022) that analyzes high-dimensional linear regression from an IB perspective. For $I(y; \mathbf{t})$, we demonstrate that the commonly used mean squared error (MSE) loss is not an ideal approximation. Rather, the CS divergence offers a new prediction term which does not take any distributional assumptions on the decoder. Similarly, for $I(\mathbf{x}; \mathbf{t})$, we demonstrate that it can be explicitly estimated from samples without variational or non-parametric upper bound approximations, by making use of the Cauchy-Schwarz quadratic mutual information (CS-QMI). Both terms can be optimized with gradient-based approaches. To summarize, we make the following major contributions:

- We show that nonlinear IB for regression on arbitrarily distributed $p(\mathbf{x}, y)$ can be carried out, with least dependence on variational approximations and no distributional assumptions with the aid of CS divergence estimated in a non-parametric way. The resulting prediction term is no longer a simple MSE loss, as in the Kullback-Leibler (KL) case; the compression term measures the true mutual information value, rather than its upper bound.

- We demonstrate that the CS divergence induced prediction and compression terms can be estimated elegantly from observations with closed-form expressions. We also establish the close connection between CS divergence with respect to maximum mean discrepancy (MMD) (Gretton et al., 2012) and KL divergence.

- We provide in-depth analysis on the generalization error and adversarial robustness of our developed CS-IB. We demonstrate the superior performance of CS-IB on six benchmark regression datasets against five popular deep IB approaches, such as nonlinear IB (NIB) (Kolchinsky et al., 2019b) and Hilbert-Schmidt Independence Criterion Bottleneck (HSIC-bottleneck) (Wang et al., 2021), in terms of generalization error, adversarial robustness, and the trade-off between prediction accuracy and compression ratio in the information plane (Shwartz-Ziv & Tishby, 2017).

## 2 BACKGROUND KNOWLEDGE

### 2.1 PROBLEM FORMULATION AND VARIANTS OF IB LAGRANGIAN

In supervised learning, we have a training set $\mathcal{D} = \{\mathbf{x}_i, y_i\}_{i=1}^N$ of input feature $\mathbf{x}$ and described response $y$. We assume $\mathbf{x}_i$ and $y_i$ are sampled *i.i.d.* from a true data distribution $p(\mathbf{x}, y) = p(y|\mathbf{x})p(x)$. The usual high-level goal of supervised learning is to use the dataset $\mathcal{D}$ to learn a particular conditional

distribution $q_\theta(\hat{y}|x)$ of the task outputs given the input features parameterized by $\theta$ which is a good approximation of $p(y|\mathbf{x})$, in which $\hat{y}$ refers to the predicted output.

If we measure the closeness between $p(y|\mathbf{x})$ and $q_\theta(\hat{y}|\mathbf{x})$ with KL divergence, the learning objective becomes:

$$\min D_{\mathrm{KL}}(p(y|\mathbf{x}); q_\theta(\hat{y}|\mathbf{x})) = \min \mathbb{E}\left(-\log(q_\theta(\hat{y}|\mathbf{x}))\right) - H(y|\mathbf{x}) \Leftrightarrow \min \mathbb{E}\left(-\log(q_\theta(\hat{y}|\mathbf{x}))\right), \quad (3)$$

where $H(y|\mathbf{x})$ only depends on $\mathcal{D}$ that is independent to parameters $\theta$.

For classification, $q_\theta(\hat{y}|\mathbf{x})$ is characterized by a discrete distribution which could be the output of a neural network $h_\theta(x)$, and Eq. (3) is exactly the cross-entropy loss. For regression, suppose $q_\theta(\hat{y}|\mathbf{x})$ is distributed normally $\mathcal{N}(h_\theta(\mathbf{x}), \sigma^2 I)$, and the network $h_\theta(\mathbf{x})$ gives the prediction of the mean of the Gaussian, the objective reduces to $\mathbb{E}\left(\|y - h_\theta(\mathbf{x})\|_2^2\right)$, which amounts to the mean squared error (MSE) loss[1] and is empirically estimated by $\frac{1}{N}\sum_{i=1}^{N}(y - \hat{y})^2$.

Most supervised learning methods aim to learn an intermediate representation $\mathbf{t}$ before making predictions about $y$. Examples include activations of hidden layers in neural networks and transformations in a feature space through the kernel trick in SVM. Suppose $\mathbf{t}$ is a possibly stochastic function of input $\mathbf{x}$, as determined by some parameterized conditional distribution $q_\theta(\mathbf{t}|\mathbf{x})$, the overall estimation of the conditional probability $p(y|\mathbf{x})$ is given by the marginalization over the representations:

$$q_\theta(\hat{y}|\mathbf{x}) = \int q_\theta(\hat{y}|\mathbf{t})q_\theta(\mathbf{t}|\mathbf{x})d\mathbf{t} \quad \text{or} \quad q_\theta(\hat{y}|\mathbf{x}) = \sum_t q_\theta(\hat{y}|\mathbf{t})q_\theta(\mathbf{t}|\mathbf{x}). \quad (4)$$

There are multiple ways to learn $\mathbf{t}$ to obtain $q_\theta(\hat{y}|\mathbf{x})$. From an information-theoretic perspective, a particular appealing framework is the IB Lagrangian in Eq. (2), which, however, may involve intractable integrals. Recently, scalable methods of IB on continuous and non-Gaussian data using DNNs became possible thanks to different mutual information approximation techniques. In the following, we briefly introduce existing approaches on approximating $I(y;\mathbf{t})$ and $I(\mathbf{x};\mathbf{t})$.

### 2.1.1 APPROXIMATION TO $I(y;\mathbf{t})$

In regression, maximizing $I_\theta(y;\mathbf{t})$ is commonly approximated by minimizing MSE loss.

**Proposition 1.** *(Rodriguez Galvez, 2019) With a Gaussian assumption on $q_\theta(\hat{y}|t)$, maximizing $I_\theta(y;\mathbf{t})$ essentially minimizes $D_{KL}(p(y|\mathbf{x}); q_\theta(\hat{y}|\mathbf{x}))$, both of which could be approximated by minimizing a MSE loss.*

*Proof.* All proofs can be found in Appendix A. $\qquad\square$

### 2.1.2 APPROXIMATION TO $I(\mathbf{x};\mathbf{t})$

The approximation to $I(\mathbf{x};\mathbf{t})$ differs for each method. For variational IB (VIB) (Alemi et al., 2017) and similar works (Achille & Soatto, 2018), $I(\mathbf{x};\mathbf{t})$ is upper bounded by:

$$I(\mathbf{x};\mathbf{t}) = \mathbb{E}_{p(x,t)}\log p(t|x) - \mathbb{E}_{p(t)}\log p(t) \leq \mathbb{E}_{p(x,t)}\log p(t|x) - \mathbb{E}_{p(t)}\log v(t) = D_{\mathrm{KL}}(p(t|x); v(t)), \quad (5)$$

where $v$ is some prior distribution such as Gaussian. On the other hand, the nonlinear information bottleneck (NIB) (Kolchinsky et al., 2019b) uses a non-parametric upper bound of mutual information (Kolchinsky & Tracey, 2017):

$$I(\mathbf{x};\mathbf{t}) \leq -\frac{1}{N}\sum_{i=1}^{N}\log\frac{1}{N}\sum_{j=1}^{N}\exp\left(-D_{\mathrm{KL}}(p(t|x_i); p(t|x_j))\right). \quad (6)$$

Recently, (Kolchinsky et al., 2019a) showed that optimizing the IB Lagrangian for different values of $\beta$ cannot explore the IB curve when $y$ is a deterministic function of $\mathbf{x}$. Therefore, the authors of (Kolchinsky et al., 2019a) propose a simple modification to IB Lagrangian, which is also called the squared-IB Lagrangian:

$$\min -I(y;\mathbf{t}) + \beta I(\mathbf{x};\mathbf{t})^2. \quad (7)$$

---

[1]Note that, $\log(q_\theta(\hat{y}|\mathbf{x})) = \log\left(\frac{1}{\sqrt{2\pi}\sigma}\exp\left(-\frac{\|y - h_\theta(\mathbf{x})\|_2^2}{2\sigma^2}\right)\right) = -\log\sigma - \frac{1}{2}\log(2\pi) - \frac{\|y - f_\theta(\mathbf{x})\|_2^2}{2\sigma^2}$.

The convex-IB (Rodríguez Gálvez et al., 2020) further showed that applying any monotonically increasing and strictly convex function $u$ on $I(\mathbf{x}; \mathbf{t})$ can explore the IB curve. For example, by instantiating $u$ with shifted exponential function, the objective of convex-IB is expressed as:

$$\min -I(y; \mathbf{t}) + \beta \exp\left(\eta(I(\mathbf{x}; \mathbf{t}) - r^*)\right), \eta > 0, r^* \in [0, \infty). \tag{8}$$

In both squared-IB and convex-IB, $I(\mathbf{x}; \mathbf{t})$ is evaluated with Eq. (6). There are other approaches that do not estimate $I(\mathbf{x}; \mathbf{t})$. The conditional entropy bottleneck (Fischer, 2020) and the deterministic information bottleneck (Strouse & Schwab, 2017) replace $I(\mathbf{x}; \mathbf{t})$, respectively, with $I(\mathbf{x}; \mathbf{t}|y)$ and $H(\mathbf{t})$. The decodable information bottleneck (Dubois et al., 2020) makes use of the $\mathcal{V}$-information (Xu et al., 2020) and defines "minimality" in classification as not being able to distinguish between examples with the same labels. One can also replace $I(\mathbf{x}; \mathbf{t})$ with other nonlinear dependence measures like the Hilbert–Schmidt independence criterion (HSIC) (Gretton et al., 2007).

## 2.2 CAUCHY-SCHWARZ DIVERGENCE AND ITS INDUCED MEASURES

Motivated by the famed Cauchy-Schwarz (CS) inequality for square integrable functions:

$$\left(\int p(\mathbf{x})q(\mathbf{x})d\mathbf{x}\right)^2 \leq \int p(\mathbf{x})^2 d\mathbf{x} \int q(\mathbf{x})^2 d\mathbf{x}, \tag{9}$$

with equality iff $p(\mathbf{x})$ and $q(\mathbf{x})$ are linearly dependent, a measure of the "distance" between $p(\mathbf{x})$ and $q(\mathbf{x})$ can be defined, which was named CS divergence (Principe et al., 2000; Yu et al., 2023), with:

$$D_{\mathrm{CS}}(p; q) = -\log\left(\frac{\left(\int p(\mathbf{x})q(\mathbf{x})d\mathbf{x}\right)^2}{\int p(\mathbf{x})^2 d\mathbf{x} \int q(\mathbf{x})^2 d\mathbf{x}}\right). \tag{10}$$

The CS divergence is symmetric for any two probability density functions (PDFs) $p$ and $q$, such that $0 \leq D_{\mathrm{CS}} < \infty$, where the minimum is obtained iff $p(\mathbf{x}) = q(\mathbf{x})$. Given samples $\{\mathbf{x}_i^p\}_{i=1}^m$ and $\{\mathbf{x}_i^q\}_{i=1}^n$, drawn i.i.d. from respectively $p(\mathbf{x})$ and $q(\mathbf{x})$, CS divergence can be empirically estimated with the kernel density estimator (KDE) (Parzen, 1962) as:

$$\widehat{D}_{\mathrm{CS}}(p; q) = \log\left(\frac{1}{m^2}\sum_{i,j=1}^m \kappa(\mathbf{x}_i^p, \mathbf{x}_j^p)\right) + \log\left(\frac{1}{n^2}\sum_{i,j=1}^n \kappa(\mathbf{x}_i^q, \mathbf{x}_j^q)\right) - 2\log\left(\frac{1}{mn}\sum_{i=1}^m \sum_{j=1}^n \kappa(\mathbf{x}_i^p, \mathbf{x}_j^q)\right). \tag{11}$$

where $\kappa$ is a kernel function such as Gaussian $\kappa_\sigma(\mathbf{x}, \mathbf{x}') = \exp(-\|\mathbf{x} - \mathbf{x}'\|_2^2/2\sigma^2)$.

**Remark 1.** *The CS divergence is also a special case of the generalized divergence defined by (Lutwak et al., 2005) which relies on a modification of the Hölder inequality[2], when $\alpha = 2$:*

$$D_\alpha(p; q) = \log\left(\frac{\left(\int q(\mathbf{x})^{\alpha-1}p(\mathbf{x})\right)^{\frac{1}{1-\alpha}}\left(\int q(\mathbf{x})^\alpha\right)^{\frac{1}{\alpha}}d\mathbf{x}}{\left(\int p(\mathbf{x})^\alpha\right)^{\frac{1}{\alpha(1-\alpha)}}d\mathbf{x}}\right). \tag{12}$$

*The KL divergence $D_{KL}(p\|q) = \int p\log\left(\frac{p}{q}\right)$ is obtained when $\alpha \to 1$.*

**Remark 2.** *The CS divergence is closely related to the maximum mean discrepancy (MMD) (Gretton et al., 2012). In fact, given a characteristic kernel $\kappa(\mathbf{x}, \mathbf{x}') = \langle\phi(\mathbf{x}), \phi(\mathbf{x}')\rangle_{\mathcal{H}}$, let us denote the (empirical) mean embedding for $\{\mathbf{x}_i^p\}_{i=1}^m$ and $\{\mathbf{x}_i^q\}_{i=1}^n$ as $\boldsymbol{\mu}_p = \frac{1}{m}\sum_{i=1}^m \phi(\mathbf{x}_i^p)$ and $\boldsymbol{\mu}_q = \frac{1}{n}\sum_{i=1}^n \phi(\mathbf{x}_i^q)$, the empirical estimators of CS divergence and MMD can be expressed as:*

$$\widehat{D}_{CS}(p; q) = -2\log\left(\frac{\langle\boldsymbol{\mu}_p, \boldsymbol{\mu}_q\rangle_{\mathcal{H}}}{\|\boldsymbol{\mu}_p\|_{\mathcal{H}}\|\boldsymbol{\mu}_q\|_{\mathcal{H}}}\right) = -2\log\cos(\boldsymbol{\mu}_p, \boldsymbol{\mu}_q), \tag{13}$$

$$\widehat{MMD}^2(p; q) = \langle\boldsymbol{\mu}_p, \boldsymbol{\mu}_q\rangle_{\mathcal{H}}^2 = \|\boldsymbol{\mu}_p\|_{\mathcal{H}}^2 + \|\boldsymbol{\mu}_q\|_{\mathcal{H}}^2 - 2\langle\boldsymbol{\mu}_p, \boldsymbol{\mu}_q\rangle_{\mathcal{H}}$$

$$= \frac{1}{m^2}\sum_{i,j=1}^m \kappa(\mathbf{x}_i^p, \mathbf{x}_j^p) + \frac{1}{n^2}\sum_{i,j=1}^n \kappa(\mathbf{x}_i^q, \mathbf{x}_j^q) - \frac{2}{mn}\sum_{i=1}^m \sum_{j=1}^n \kappa(\mathbf{x}_i^p, \mathbf{x}_j^q). \tag{14}$$

*That is, CS divergence measures the cosine similarity between $\boldsymbol{\mu}_p$ and $\boldsymbol{\mu}_q$ in a Reproducing kernel Hilbert space (RKHS) $\mathcal{H}$, whereas MMD uses Euclidean distance.*

Eq. (10) can be extended to measure the independence between two random variables $\mathbf{x}$ and $y$.

---

[2]See Appendix B.1 for an in-depth discussion on the relationship between CS & KL divergences and MMD.

**Cauchy-Schwarz Quadratic Mutual Information (CS-QMI)** The independence between $\mathbf{x}$ and $y$ can be measured by any (valid) distance or divergence measure over the joint distribution $p(\mathbf{x}, y)$ with respect to the product of marginal distributions $p(\mathbf{x})p(y)$. If we substitute $p(\mathbf{x})$ and $q(\mathbf{x})$ in Eq. (10) with $p(\mathbf{x}, y)$ and $p(\mathbf{x})p(y)$, we obtain the Cauchy-Schwarz quadratic mutual information (CS-QMI) (Principe et al., 2000):

$$I_{\mathrm{CS}}(\mathbf{x}, y) = D_{\mathrm{CS}}(p(\mathbf{x}, y); p(\mathbf{x})p(y)) = -\log\left(\frac{\left|\int p(\mathbf{x}, y)p(\mathbf{x})p(y)d\mathbf{x}dy\right|^2}{\int p^2(\mathbf{x}, y)d\mathbf{x}dy \int p^2(\mathbf{x})p^2(y)d\mathbf{x}dy}\right). \quad (15)$$

Distinct to KL divergence that is notoriously hard to estimate, we will show that both CS-QMI and the divergence between $p(y|\mathbf{x})$ and $q_\theta(\hat{y}|\mathbf{x})$ can be elegantly estimated in a non-parametric way with closed-form expressions, enabling efficient implementation of deep IB without approximations.

## 3 THE CAUCHY-SCHWARZ DIVERGENCE INFORMATION BOTTLENECK

As has been discussed in Section 2.1, existing deep IB approaches rely on Shannon's definition of information and are essentially minimizing the following objective:

$$\min_{p(\mathbf{t}|\mathbf{x})} D_{\mathrm{KL}}(p(y|\mathbf{x}); q_\theta(\hat{y}|\mathbf{x})) + \beta I(\mathbf{x}; \mathbf{t}), \quad (16)$$

where $I(\mathbf{x}; \mathbf{t})$ is defined also in a KL divergence sense. Both terms in Eq. (16) are hard to estimate. In case of regression, $\min D_{\mathrm{KL}}(p(y|\mathbf{x}); q_\theta(\hat{y}|\mathbf{x}))$ gets back to the MSE loss under a parametric Gaussian assumption. However, we could also uncover the mean absolute error (MAE) loss if we take a Laplacian assumption. On the other hand, the use of an upper bound to $I(\mathbf{x}; \mathbf{t})$ also makes the solution of IB sub-optimal. Moreover, the choices of bounds or assumptions are hard to decide for practitioners. We refer interested readers to Appendix B.2 for a detailed discussion on different approximation biases of previous literature. We also demonstrate the advantage of CS divergence over variational KL divergence in terms of optimization via a toy example in Appendix C.5.

In this paper, we consider a new formulation of IB that entirely based on the CS divergence:

$$\min_{p(\mathbf{t}|\mathbf{x})} D_{\mathrm{CS}}(p(y|\mathbf{x}); q_\theta(\hat{y}|\mathbf{x})) + \beta I_{\mathrm{CS}}(\mathbf{x}; \mathbf{t}). \quad (17)$$

The first term of Eq. (17) measures the conditional CS divergence between $p(y|\mathbf{x})$ and $q_\theta(\hat{y}|\mathbf{x})$, whereas the second term is the CS-QMI between $\mathbf{x}$ and $\mathbf{t}$. We will show in next subsections how the favorable properties of CS divergence possibly affect the IB's computation and performance, compared to its KL divergence counterpart.

### 3.1 ESTIMATION OF CS DIVERGENCE INDUCED TERMS

Both terms in Eq. (17) (i.e., $D_{\mathrm{CS}}(p(y|\mathbf{x}); q_\theta(\hat{y}|\mathbf{x}))$ and $I_{\mathrm{CS}}(\mathbf{x}; \mathbf{t})$) can be efficiently and non-parametrically estimated from given samples $\{y_i, \mathbf{x}_i, \mathbf{t}_i, \hat{y}_i\}_{i=1}^n$ with the kernel density estimator (KDE) in a closed-form expression. When parameterizing IB with a DNN, $\mathbf{t}$ refers to the latent representation of one hidden layer (i.e., $\mathbf{t} = f(\mathbf{x})$), $n$ denotes mini-batch size.

**Proposition 2** (Empirical Estimator of $D_{\mathrm{CS}}(p(y|\mathbf{x}); q_\theta(\hat{y}|\mathbf{x}))$). *Given observations $\{(\mathbf{x}_i, y_i, \hat{y}_i)\}_{i=1}^N$, where $\mathbf{x} \in \mathbb{R}^p$ denotes a $p$-dimensional input variable, $y$ is the desired response, and $\hat{y}$ is the predicted output generated by a model $f_\theta$. Let $K$, $L^1$ and $L^2$ denote, respectively, the Gram matrices for the variable $\mathbf{x}$, $y$, and $\hat{y}$ (i.e., $K_{ij} = \kappa(\mathbf{x}_i, \mathbf{x}_j)$, $L_{ij}^1 = \kappa(y_i, y_j)$ and $L_{ij}^2 = \kappa(\hat{y}_i, \hat{y}_j)$, in which $\kappa = \exp\left(-\frac{\|\cdot\|^2}{2\sigma^2}\right)$ is a Gaussian kernel function). Further, let $L^{21}$ denote the Gram matrix between $\hat{y}$ and $y$ (i.e., $L_{ij}^{21} = \kappa(\hat{y}_i, y_j)$). The empirical estimation of $D_{CS}(p(y|\mathbf{x}); q_\theta(\hat{y}|\mathbf{x}))$ is given by:*

$$\widehat{D}_{CS}(p(y|\mathbf{x}); q_\theta(\hat{y}|\mathbf{x})) = \log\left(\sum_{j=1}^N \left(\frac{\sum_{i=1}^N K_{ji}L_{ji}^1}{(\sum_{i=1}^N K_{ji})^2}\right)\right)$$

$$+ \log\left(\sum_{j=1}^N \left(\frac{\sum_{i=1}^N K_{ji}L_{ji}^2}{(\sum_{i=1}^N K_{ji})^2}\right)\right) - 2\log\left(\sum_{j=1}^N \left(\frac{\sum_{i=1}^N K_{ji}L_{ji}^{21}}{(\sum_{i=1}^N K_{ji})^2}\right)\right). \quad (18)$$

**Remark 3.** *The empirical estimator of $D_{CS}(p(y|\mathbf{x}); q_\theta(\hat{y}|\mathbf{x}))$ in Eq. (18) is non-negative. Compared to MSE, it assumes and encourages that if two points $\mathbf{x}_i$ and $\mathbf{x}_j$ are sufficiently close, their predictions ($\hat{y}_i$ and $\hat{y}_j$) will be close as well, which also enhances the numerical stability (Nguyen & Raff, 2019) of the trained model. Moreover, when the kernel width $\sigma$ reduces to 0, it gets back to MSE.*

**Remark 4.** *A precise estimation on the divergence between $p(y|\mathbf{x})$ and $q_\theta(\hat{y}|\mathbf{x})$ is a non-trivial task. The use of KL divergence combined with Gaussian assumption is likely to introduce inductive bias. Another alternative is the conditional MMD by (Ren et al., 2016) with the expression $\widehat{D}_{MMD}(p(y|\mathbf{x}); q_\theta(\hat{y}|\mathbf{x})) = \mathrm{tr}(K\tilde{K}^{-1}L^1\tilde{K}^{-1}) + \mathrm{tr}(K\tilde{K}^{-1}L^2\tilde{K}^{-1}) - 2\,\mathrm{tr}(K\tilde{K}^{-1}L^{21}\tilde{K}^{-1})$, in which $\tilde{K} = K + \lambda I$. Obviously, CS divergence avoids introducing an additional hyperparametr $\lambda$ and the necessity of matrix inverse, which improves computational efficiency and stability. Moreover, Eq. (18) does not rely on any parametric distributional assumption on $q_\theta(\hat{y}|\mathbf{x})$.*

We provide detailed justifications regarding Remark 3 and Remark 4 in Appendix B.3.

**Proposition 3** (Empirical Estimator of CS-QMI). *Given $N$ pairs of observations $\{(\mathbf{x}_i, \mathbf{t}_i)\}_{i=1}^N$, each sample contains two different types of measurements $\mathbf{x} \in \mathcal{X}$ and $\mathbf{t} \in \mathcal{T}$ obtained from the same realization. Let $K$ and $Q$ denote, respectively, the Gram matrices for variable $\mathbf{x}$ and variable $\mathbf{t}$, which are also symmetric. The empirical estimator of CS-QMI is given by:*

$$\widehat{I}_{CS}(\mathbf{x}; \mathbf{t}) = \log\left(\frac{1}{N^2}\sum_{i,j}^N K_{ij}Q_{ij}\right) + \log\left(\frac{1}{N^4}\sum_{i,j,q,r}^N K_{ij}Q_{qr}\right) - 2\log\left(\frac{1}{N^3}\sum_{i,j,q}^N K_{ij}Q_{iq}\right)$$

$$= \log\left(\frac{1}{N^2}\mathrm{tr}(KQ)\right) + \log\left(\frac{1}{N^4}\mathbb{1}^T K\mathbb{1}\mathbb{1}^T Q\mathbb{1}\right) - 2\log\left(\frac{1}{N^3}\mathbb{1}^T KQ\mathbb{1}\right), \tag{19}$$

*where $\mathbb{1}$ is a $N \times 1$ vector of ones. The second line of Eq. (19) reduces the complexity to $\mathcal{O}(N^2)$.*

**Remark 5.** *Our empirical estimator of CS-QMI in Eq. (19) is closely related to the most widely used biased (or V-statistics) estimator of HSIC, which can be expressed as (Gretton et al., 2007):*

$$\widehat{HSIC}_b(\mathbf{x}; \mathbf{t}) = \frac{1}{N^2}\sum_{i,j}^N K_{ij}Q_{ij} + \frac{1}{N^4}\sum_{i,j,q,r}^N K_{ij}Q_{qr} - \frac{2}{N^3}\sum_{i,j,q}^N K_{ij}Q_{iq} = \frac{1}{N^2}\mathrm{tr}(KHQH), \tag{20}$$

*where $H = I - \frac{1}{N}\mathbb{1}\mathbb{1}^T$ is a $N \times N$ centering matrix. Comparing Eq. (19) with Eq. (20), it is easy to observe that CS-QMI adds a logarithm operator on each term of HSIC. This is not surprising. CS-QMI equals the CS divergence between $p(\mathbf{x}, \mathbf{t})$ and $p(\mathbf{x})p(\mathbf{t})$; whereas HSIC is equivalent to MMD between $p(\mathbf{x}, \mathbf{t})$ and $p(\mathbf{x})p(\mathbf{t})$. According to Remark 2, similar rule could be expected. However, different to HSIC, the CS-QMI is a rigorous definition of mutual information. It measures the mutual dependence between variables $\mathbf{x}$ and $y$ in units like bit (with $\log_2$) or nat (with $\ln$).*

## 3.2 The Rationality of the Regularization Term $I_{\mathrm{CS}}(\mathbf{x}; \mathbf{t})$

The advantages of replacing MSE (or MAE) loss with $D_{\mathrm{CS}}(p(y|\mathbf{x}); q_\theta(\hat{y}|\mathbf{x}))$ is elaborated in Remarks 3 and 4. Hence, we justify the rationality of the regularization term $I_{\mathrm{CS}}(\mathbf{x}; \mathbf{t})$.

### 3.2.1 Effects of $I_{\mathrm{CS}}(\mathbf{x}; \mathbf{t})$ on Generalization

Theorem 1 suggests that CS divergence is upper bounded by the smaller value between forward and reverse KL divergences, which improves the stability of training. Additionally, Corollary 1 implies that $I_{\mathrm{CS}}(\mathbf{x}; \mathbf{t})$ encourages a smaller value on the dependence between $\mathbf{x}$ and $\mathbf{t}$, i.e., a heavier penalty on the information compression. These results can be extended to arbitrary square-integral densities as shown in Appendix B.5. Note that, this is distinct to the mainstream IB optimization idea that just minimizes an upper bound of Shannon's mutual information $I(\mathbf{x}; \mathbf{t})$ due to the difficulty of estimation. Similar to Eq. (18), Eq. (19) makes the estimation of $I_{\mathrm{CS}}(\mathbf{x}; \mathbf{t})$ straightforward without any approximation or parametric assumptions on the underlying data distribution.

**Theorem 1.** *For arbitrary $d$-variate Gaussian distributions $p \sim \mathcal{N}(\mu_1, \Sigma_1)$ and $q \sim \mathcal{N}(\mu_2, \Sigma_2)$,*

$$D_{CS}(p; q) \leq \min\{D_{KL}(p; q), D_{KL}(q; p)\}. \tag{21}$$

**Corollary 1.** *For two random vectors $\mathbf{x}$ and $\mathbf{t}$ which follow a joint Gaussian distribution $\mathcal{N}\left(\begin{pmatrix}\mu_x \\ \mu_t\end{pmatrix}, \begin{pmatrix}\Sigma_x & \Sigma_{xt} \\ \Sigma_{tx} & \Sigma_t\end{pmatrix}\right)$, the CS-QMI is no greater than the Shannon's mutual information:*

$$I_{CS}(\mathbf{x}; \mathbf{t}) \leq I(\mathbf{x}; \mathbf{t}). \tag{22}$$

|     | $I_{\text{CS}}(\mathbf{x}; \mathbf{t})$ | $I_{\text{CS}}(\mathbf{x}; \mathbf{t}|y)$ |     | $I_{\text{CS}}(\mathbf{x}; \mathbf{t})$ | $I_{\text{CS}}(\mathbf{x}; \mathbf{t}|y)$ |
|-----|------|------|-----|------|------|
| $\tau$ | 0.30 | 0.31 | $\tau$ | 0.40 | 0.44 |
| MIC | 0.38 | 0.47 | MIC | 0.46 | 0.54 |

Table 1: The dependence between $I_{\text{CS}}(\mathbf{x}; \mathbf{t})$ (or $I_{\text{CS}}(\mathbf{x}; \mathbf{t}|y)$) and generalization gap (measured by Kendall's $\tau$ and MIC) on synthetic data (left) and real-world California housing data (right).

Minimizing unnecessary information (by minimizing the dependence between $\mathbf{x}$ and $\mathbf{t}$) to control generalization error has inspired lots of deep learning algorithms. In classification setup, recent study states that given $m$ training samples, with probability $1 - \delta$, the generalization gap could be upper bounded by $\sqrt{\frac{2^{I(\mathbf{x};\mathbf{t})} + \log(1/\delta)}{2m}}$ (Shwartz-Ziv et al., 2019; Galloway et al., 2023), which has been rigorously justified in (Kawaguchi et al., 2023) by replacing $2^{I(\mathbf{x};\mathbf{t})}$ with $I(\mathbf{x}; \mathbf{t}|y)$. It is natural to ask if similar observations hold for regression. We hypothesize that the compression on the dependence between $\mathbf{x}$ and $\mathbf{t}$ also plays a fundamental role to predict the generalization of a regression model, and $I_{\text{CS}}(\mathbf{x}; \mathbf{t})$ or $I_{\text{CS}}(\mathbf{x}; \mathbf{t}|y)$ correlates well with the generalization performance of a trained network.

To test this, we train nearly one hundred fully-connected neural networks with varying hyperparameters (depth, width, batch size, dimensionality of $\mathbf{t}$) on both a synthetic nonlinear regression data with 30 dimensional input and a real-world California housing data, and retain models that reach a stable convergence. For all models, we measure the dependence between $I_{\text{CS}}(\mathbf{x}; \mathbf{t})$ (or $I_{\text{CS}}(\mathbf{x}; \mathbf{t}|y)$) and the generalization gap (i.e., the performance difference in training and test sets in terms of rooted mean squared error) with both Kendall's $\tau$ and maximal information coefficient (MIC) (Reshef et al., 2011), as shown in Table 1. For Kendall's $\tau$, values of $\tau$ close to 1 indicate strong agreement of two rankings for samples in variables $x$ and $y$, that that is, if $x_i > x_j$, then $y_i > y_j$. Kendall's $\tau$ matches our motivation well, since we would like to evaluate if a small value of $I_{\text{CS}}(\mathbf{x}; \mathbf{t})$ (or $I_{\text{CS}}(\mathbf{x}; \mathbf{t}|y)$) is likely to indicate a smaller generalization gap. This result is in line with that in (Kawaguchi et al., 2023) and corroborates our hypothesis. See Appendix C.1 for details and additional results.

Finally, we would like emphasize that when $p$ and $q$ are sufficiently small, Theorem 1 could be extended without Gaussian assumption, which may enable us to derive tighter generalization error bound in certain learning scenarios. We refer interested readers to Appendix B.5 for more discussions.

### 3.2.2 ADVERSARIAL ROBUSTNESS GUARANTEE

Given a network $h_\theta = g(f(\mathbf{x}))$, where $f : \mathbb{R}^{d_X} \mapsto \mathbb{R}^{d_T}$ maps the input to an intermediate layer representation $\mathbf{t}$, and $g : \mathbb{R}^{d_T} \mapsto \mathbb{R}$ maps this intermediate representation $\mathbf{t}$ to the final layer, we assume all functions $h_\theta$ and $g$ we consider are uniformly bounded by $M_\mathcal{X}$ and $M_\mathcal{Z}$, respectively. Let us denote $\mathcal{F}$ and $\mathcal{G}$ the induced RKHSs for kernels $\kappa_X$ and $\kappa_Z$, and assume all functions in $\mathcal{F}$ and $\mathcal{G}$ are uniformly bounded by $M_\mathcal{F}$ and $M_\mathcal{G}$. Based on Remark 2 and the result by (Wang et al., 2021), let $\mu(\mathbb{P}_{XT})$ and $\mu(\mathbb{P}_X \otimes \mathbb{P}_T)$ denote, respectively, the (empirical) kernel mean embedding of $\mathbb{P}_{XT}$ and $\mathbb{P}_X \otimes \mathbb{P}_T$ in the RHKS $\mathcal{F} \otimes \mathcal{G}$, CS-QMI bounds the power of an arbitrary adversary in $\mathcal{S}_r$ when $\sqrt{N}$ is sufficiently large, in which $\mathcal{S}_r$ is a $\ell_\infty$-ball of radius $r$, i.e., $\mathcal{S}_r = \{\delta \in \mathbb{R}^{d_X}, \delta_\infty \leq r\}$.

**Proposition 4.** *Denote* $\gamma = \frac{\sigma M_\mathcal{F} M_\mathcal{G}}{r\sqrt{-2\log o(1) d_X M_\mathcal{Z}}} \left( \mathbb{E}[|h_\theta(\mathbf{x} + \delta) - h_\theta(\mathbf{x})|] - o(r) \right)$, *if* $\mathbf{x} \sim \mathcal{N}(0, \sigma^2 I)$ *and* $\|\mu(\mathbb{P}_{XT})\|_{\mathcal{F} \otimes \mathcal{G}} = \|\mu(\mathbb{P}_X \otimes \mathbb{P}_T)\|_{\mathcal{F} \otimes \mathcal{G}} = \|\mu\|$, *when* $\sqrt{N} \gg |g'(HSIC(\mathbf{x}; \mathbf{t}))|\sigma_H$, *then:*

$$\mathbb{P}\left( \widehat{I}_{CS}(\mathbf{x}; \mathbf{t}) \geq g(\gamma) \right) \approx 1 - \Phi\left( \frac{\sqrt{N}(g(\gamma) - g(HSIC(\mathbf{x}; \mathbf{t})))}{|g'(HSIC(\mathbf{x}; \mathbf{t}))|\sigma_H} \right) \to 1, \tag{23}$$

*in which* $g(x) = -2\log(1 - x/(2\|\mu\|^2))$ *is a monotonically increasing function,* $\Phi$ *is the cumulative distribution function of a standard Gaussian, and* $\sqrt{N}(\widehat{HSIC}_b(\mathbf{x}; \mathbf{t}) - HSIC(\mathbf{x}; \mathbf{t})) \xrightarrow{D} \mathcal{N}(0, \sigma_H^2)$.

## 4 EXPERIMENTS

We perform experiments on four benchmark regression datasets: California Housing, Appliance Energy, Beijing PM2.5, and Bike Sharing from the UCI repository. To showcase the scalability of CS-IB to high-dimensional data (e.g., images), we additionally report its performance on rotation

MNIST and UTKFace (Zhang et al., 2017), in which the tasks are respectively predicting the rotation angle of MNIST digits and estimating the age of persons by their face images. We compare CS-IB to popular deep IB approaches that could be used for regression tasks. These include VIB (Alemi et al., 2017), NIB (Kolchinsky et al., 2019b), squared-NIB (Kolchinsky et al., 2019a), convex-NIB with exponential function (Rodríguez Gálvez et al., 2020) and HSIC-bottleneck (Wang et al., 2021). Similar to (Kolchinsky et al., 2019b;a; Rodríguez Gálvez et al., 2020), we optimize all competing methods for different values of $\beta$, producing a series of models that explore the trade-off between compression and prediction.

The network $h_\theta = g(f(\mathbf{x}))$ is consists of two parts: a stochastic encoder $\mathbf{t} = f_{\text{enc}}(\mathbf{x}) + w$ where $w$ is zero-centered Gaussian noise with covariance $\texttt{diag}(\sigma_\theta^2)$ and $p(\mathbf{t}|\mathbf{x}) = \mathcal{N}(f_{\text{enc}}(\mathbf{x}), \texttt{diag}(\sigma_\theta^2))$, and a deterministic decoder $y = g_{\text{dec}}(\mathbf{t})$. For benchmark regression datasets, the encoder $f_{\text{enc}}$ is a 3-layer fully-connected network or LSTM. For rotation MNIST and UTKFace, we use VGG-16 (Simonyan & Zisserman, 2015) as the backbone architecture. Detail on experimental setup is in Appendix C.2.

## 4.1 Behaviors in the Information Plane

For each IB approach, we traverse different values of $\beta \geq 0$. Specifically, we vary $\beta \in [10^{-3}, 10]$ for CS-IB, $\beta \in [10^{-2}, 1]$ for NIB, square-NIB, and exp-NIB, $\beta \in [10^{-6}, 10^{-3}]$ for VIB, and $\beta \in [1, 10]$ for the HSIC$(\mathbf{x}; \mathbf{t})$ term in HSIC-bottleneck. These ranges were chosen empirically so that the resulting models fully explore the IB curve. To fairly compare the capability of prediction under different compression levels, we define the compression ratio $r$ at $\beta = \beta^*$ as $1 - I(\mathbf{x}; \mathbf{t})_{\beta=\beta^*}/I(\mathbf{x}; \mathbf{t})_{\beta=0}$. Hence, $r$ equals 0 when $\beta = 0$ (i.e., no compression term in the IB objective). Intuitively, a large $\beta$ would result in small value of $I(\mathbf{x}; \mathbf{t})$, and hence large $r$. Here, $I(\mathbf{x}; \mathbf{t})$ is calculated by each approach's own estimator, whereas the true value of $I(y; \mathbf{t})$ is approximated with $\frac{1}{2}\log(\text{var}(y)/\text{MSE})$ (Kolchinsky et al., 2019b). The results are summarized in Fig. 1 and Table 2. Interestingly, when $r = 0$, our CS-IB have already demonstrated an obvious performance gain, which implies that our prediction term (i.e., Eq. (18)) alone is more helpful than MSE to extract more usable information from input $\mathbf{x}$ to predict $y$. For each data, we additionally report the best performance achieved when $r \neq 0$ (see Table 7 in Appendix C.4). Again, our CS-IB outperforms others. We also perform an ablation study, showing that Eq. (18) could also improve the performances of NIB, etc. (by replacing their MSE counterpart), although they are still inferior to CS-IB. See Appendix C.4 for additional results.

Table 2: RMSE for different deep IB approaches with compression ratio $r = 0$ and $r = 0.5$. When $r = 0$, CS-IB uses prediction term $D_{\text{CS}}(p(y|\mathbf{x}); q_\theta(\hat{y}|\mathbf{x}))$ in Eq. (18), whereas others use MSE.

| Model | Housing | | Energy | | PM2.5 | | Bike | | Rotation MNIST | | UTKFace | |
|---|---|---|---|---|---|---|---|---|---|---|---|---|
| | 0 | 0.5 | 0 | 0.5 | 0 | 0.5 | 0 | 0.5 | 0 | 0.5 | 0 | 0.5 |
| VIB | 0.258 | 0.347 | 0.059 | 0.071 | 0.025 | 0.038 | 0.428 | 0.523 | 4.351 | 5.358 | 8.870 | 9.258 |
| NIB | 0.258 | 0.267 | 0.059 | 0.060 | 0.025 | 0.034 | 0.428 | 0.435 | 4.351 | 4.102 | 8.870 | 8.756 |
| Square-NIB | 0.258 | 0.293 | 0.059 | 0.063 | 0.025 | 0.028 | 0.428 | 0.447 | 4.351 | 4.257 | 8.870 | 8.712 |
| Exp-NIB | 0.258 | 0.287 | 0.059 | 0.061 | 0.025 | 0.030 | 0.428 | 0.458 | 4.351 | 4.285 | 8.870 | 8.917 |
| HSIC-bottlenck | 0.258 | 0.371 | 0.059 | 0.065 | 0.025 | 0.031 | 0.428 | 0.451 | 4.351 | 4.573 | 8.870 | 8.852 |
| CS-IB | **0.251** | **0.245** | **0.056** | **0.058** | **0.022** | **0.027** | **0.404** | **0.412** | **4.165** | **3.930** | **8.702** | **8.655** |

## 4.2 Adversarial Robustness

We then evaluate the adversarial robustness of the model trained with our CS-IB objective. Different types of adversarial attacks have been proposed to "fool" models by adding small carefully designed perturbations on the input. Despite extensive studies on adversarial robustness of classification networks, the adversarial robustness in regression setting is scarcely investigated but of crucial importance (Nguyen & Raff, 2019; Gupta et al., 2021). In this section, we use the most basic way to evaluate adversarial robustness, that is the regression performance on adversarially perturbed versions of the test set, also called the adversarial examples.

There are no formal definitions on adversarial attacks in the regression setting, we follow (Nguyen & Raff, 2019) and consider adversarial attack as a potential symptom of numerical instability in the learned function. That is, we aim to learn a numerically stable function $f_\theta$ such that the output of two points that are near each other should be similar:

$$|f_\theta(\mathbf{x}) - f_\theta(\mathbf{x} + \Delta_{\mathbf{x}})| \leq \delta, \quad \text{s.t.,} \quad \|\Delta_{\mathbf{x}}\|_p < \epsilon. \tag{24}$$

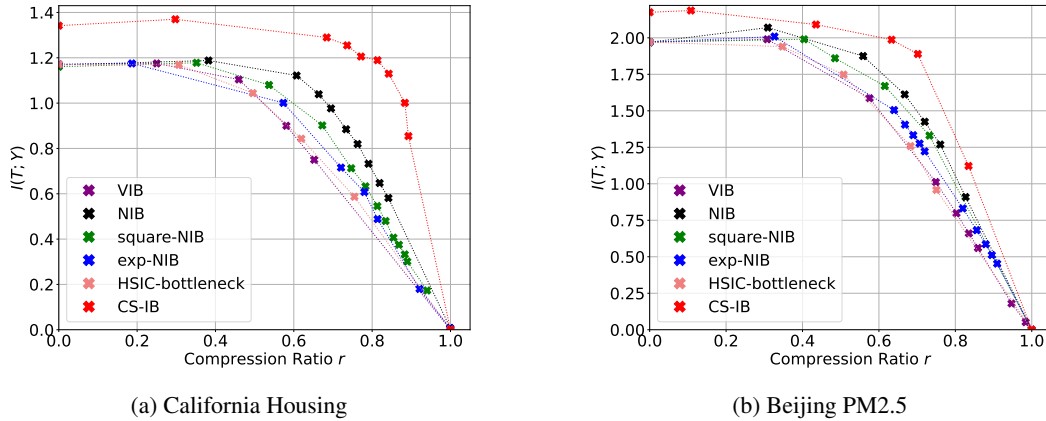

(a) California Housing                  (b) Beijing PM2.5

Figure 1: Information plane diagrams on California Housing and Beijing PM2.5 datasets.

Different to classification, there are no natural margins in regression tasks. Hence, we just consider untargeted attack and define an adversarial example $\tilde{\mathbf{x}}$ as a point within a $\ell_p$ ball with radius $\epsilon$ around $\mathbf{x}$ that causes the learned function to produce an output with the largest deviation:

$$\tilde{\mathbf{x}} = \underset{\|\mathbf{x}-\mathbf{x}'\|_p < \epsilon}{\arg\max} \ \mathcal{L}(f_\theta(\mathbf{x}'), y), \tag{25}$$

where $\mathcal{L}$ is a loss function such as mean squared error.

We apply two commonly used ways to solve Eq. (25): the Fast Gradient Sign Attack (FGSM) (Goodfellow et al., 2014) and the Projected Gradient Descent (PGD) (Madry et al., 2018). In our experiments, we evaluate performance against both white-box FGSM attack with perturbation $\epsilon = 1/255$ for two image datasets (i.e, Rotation MNIST and UTKFace), and $\epsilon = 0.1$ for the remaining four benchmark regression datasets and a white-box 5-step PGD ($\ell_\infty$) attack with perturbation $\rho = 0.3$ and step size $\alpha = 0.1$. The RMSE in the test set is shown in Table 3. As can be seen, our CS-IB outperforms other IB approaches for both types of attacks. This result corroborates our analysis in Section 3.2.2.

Table 3: White-box robustness (in terms of RMSE) with FGSM and PGD attacks. The best performance is highlighted.

| Method | Housing | | Energy | | PM2.5 | | Bike | | Rotation MNIST | | UTKFace | |
|---|---|---|---|---|---|---|---|---|---|---|---|---|
| | FGSM | PGD$^5$ | FGSM | PGD$^5$ | FGSM | PGD$^5$ | FGSM | PGD$^5$ | FGSM | PGD$^5$ | FGSM | PGD$^5$ |
| VIB | 0.706 | 0.917 | 0.502 | 0.543 | 0.432 | 0.464 | 1.633 | 2.072 | 6.754 | 7.109 | 12.151 | 13.172 |
| NIB | 0.641 | 0.732 | 0.394 | 0.471 | 0.381 | 0.415 | 1.487 | 1.840 | 5.577 | 6.898 | 11.375 | 12.654 |
| Square-NIB | 0.649 | 0.789 | 0.441 | 0.463 | 0.383 | 0.423 | 1.592 | 1.937 | 5.413 | 6.813 | 11.056 | 12.785 |
| Exp-NIB | 0.661 | 0.784 | 0.434 | 0.491 | 0.367 | 0.435 | 1.532 | 1.952 | 5.850 | 6.845 | 11.687 | 12.895 |
| HSIC-bottleneck | 0.651 | 0.765 | 0.385 | 0.485 | 0.311 | 0.349 | 1.519 | 2.011 | 5.785 | 6.923 | 11.457 | 12.776 |
| Ours | **0.635** | **0.755** | **0.290** | **0.402** | **0.278** | **0.324** | **1.478** | **1.786** | **5.023** | **6.543** | **10.824** | **12.058** |

## 5 CONCLUSION

We discuss the implementation of deep information bottleneck (IB) for the regression setup on arbitrarily distributed $p(\mathbf{x}, y)$. By making use of the Cauchy-Schwarz (CS) divergence, we obtain a new prediction term that enhances numerical stability of the trained model and also avoids Gaussian assumption on the decoder. We also obtain a new compression term that estimates the true mutual information values (rather than an upper bound) and has theoretical guarantee on adversarial robustness. Besides, we show that CS divergence is always smaller than the popular Kullback-Leibler (KL) divergence, thus enabling tighter generalization error bound. Experiments on four benchmark datasets and two high-dimensional image datasets against other five deep IB approaches over a variety of deep architectures (e.g., LSTM and VGG-16) suggest that our prediction term is scalable and helpful for extracting more usable information from input $\mathbf{x}$ to predict $y$; the compression term also improves generalization. Moreover, our model always achieves the best trade-off in terms of prediction accuracy and compression ratio. Limitations and future work are discussed in Appendix D.

ACKNOWLEDGMENTS

The authors would like to thank the anonymous reviewers for constructive comments. The authors would also like to thank Dr. Yicong Lin from the Vrije Universiteit Amsterdam for helpful discussions on proofs in Appendix B, and Mr. Kaizhong Zheng from the Xi'an Jiaotong University for performing initial study on extending CS-IB to predict the age of patients with a graph neural network in Appendix D. This work was funded in part by the Research Council of Norway (RCN) under grant 309439, and the U.S. ONR under grants N00014-18-1-2306, N00014-21-1-2324, N00014-21-1-2295, the DARPA under grant FA9453-18-1-0039.

**Reproducibility Statement.** To ensure reproducibility, we include complete proofs to our theoretical results in Appendix A, thorough justifications on the properties or advantages of our method in Appendix B, detailed explanations of our experimental setup in Appendix C. Our code is available at `https://github.com/SJYuCNEL/Cauchy-Schwarz-Information-Bottleneck`.

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

The appendix is organized into the following topics and sections:

## TABLE OF CONTENTS

## A  PROOFS

### A.1  PROOF TO PROPOSITION 1

**Proposition 1.** *(Rodriguez Galvez, 2019) With a Gaussian assumption on $q_\theta(\hat{y}|t)$, maximizing $I_\theta(y; \mathbf{t})$ essentially minimizes $D_{KL}(p(y|\mathbf{x}); q_\theta(\hat{y}|\mathbf{x}))$, both of which could be approximated by minimizing a MSE loss.*

*Proof.* A complete proof is in (Rodriguez Galvez, 2019). Intuitively, given a feed-forward neural network $h_\theta = f \circ g$, where $f$ is the encoder and $g$ is the decoder, minimizing $D_{\mathrm{KL}}(p(y|\mathbf{x}); q_\theta(\hat{y}|\mathbf{x}))$ can be approximated by minimizing a MSE loss $\mathcal{L}_{\mathrm{MSE}}(\theta) = \mathbb{E}[(y - h_\theta(\mathbf{x}))^2] = \mathbb{E}[(y - g_\theta(\mathbf{t}))^2]$ (under Gaussian assumption) (Vera et al., 2023), whereas minimizing $\mathcal{L}_{\mathrm{CE}}(\theta)$ or $\mathcal{L}_{\mathrm{MSE}}(\theta)$ maximizes $I(y; \mathbf{t})$ (Proposition 3.2 in (Rodriguez Galvez, 2019)). $\qquad\square$

### A.2  PROOF TO PROPOSITION 2

**Proposition 2** (Empirical Estimator of $D_{\mathrm{CS}}(p(y|\mathbf{x}); q_\theta(\hat{y}|\mathbf{x}))$). *Given observations $\{(\mathbf{x}_i, y_i, \hat{y}_i)\}_{i=1}^N$, where $\mathbf{x} \in \mathbb{R}^p$ denotes a $p$-dimensional input variable, $y$ is the desired response, and $\hat{y}$ is the predicted output generated by a model $f_\theta$. Let $K$, $L^1$ and $L^2$ denote, respectively, the Gram matrices[3] for the variable $\mathbf{x}$, $y$, and $\hat{y}$ (i.e., $K_{ij} = \kappa(\mathbf{x}_i, \mathbf{x}_j)$, $L^1_{ij} = \kappa(y_i, y_j)$ and $L^2_{ij} = \kappa(\hat{y}_i, \hat{y}_j)$, in which $\kappa$ is a Gaussian kernel and takes the form of $\kappa = \exp\left(-\frac{\|\cdot\|^2}{2\sigma^2}\right)$). Further, let $L^{21}$ denote the Gram matrix*

---

[3]In kernel learning, the Gram or kernel matrix is a symmetric matrix where each entry is the inner product of the corresponding data points in a reproducing kernel Hilbert space (RKHS), defined by kernel function $\kappa$.

between $\hat{y}$ and $y$ (i.e., $L_{ij}^{21} = \kappa(\hat{y}_i, y_j)$). The empirical estimation of $D_{CS}(p(y|\mathbf{x}); q_\theta(\hat{y}|\mathbf{x}))$ is given by:

$$
\begin{aligned}
\widehat{D}_{CS}(p(y|\mathbf{x}); q_\theta(\hat{y}|\mathbf{x})) &= \log\left(\sum_{j=1}^{N}\left(\frac{\sum_{i=1}^{N} K_{ji} L_{ji}^1}{(\sum_{i=1}^{N} K_{ji})^2}\right)\right) \\
&+ \log\left(\sum_{j=1}^{N}\left(\frac{\sum_{i=1}^{N} K_{ji} L_{ji}^2}{(\sum_{i=1}^{N} K_{ji})^2}\right)\right) - 2\log\left(\sum_{j=1}^{N}\left(\frac{\sum_{i=1}^{N} K_{ji} L_{ji}^{21}}{(\sum_{i=1}^{N} K_{ji})^2}\right)\right).
\end{aligned}
\tag{26}
$$

*Proof.* The derivation is largely inspired by (Yu et al., 2023). By definition, we have[4]:

$$
\begin{aligned}
D_{CS}(p(y|\mathbf{x}); q_\theta(\hat{y}|\mathbf{x})) &= \log\left(\int p^2(y|\mathbf{x}) d\mathbf{x} dy\right) + \log\left(\int q_\theta^2(\hat{y}|\mathbf{x}) d\mathbf{x} dy\right) \\
&\quad - 2\log\left(\int p(y|\mathbf{x}) q_\theta(\hat{y}|\mathbf{x}) d\mathbf{x} dy\right) \\
&= \log\left(\int \frac{p^2(\mathbf{x}, y)}{p^2(\mathbf{x})} d\mathbf{x} dy\right) + \log\left(\int \frac{q_\theta^2(\mathbf{x}, \hat{y})}{p^2(\mathbf{x})} d\mathbf{x} dy\right) \\
&\quad - 2\log\left(\int \frac{p(\mathbf{x}, y) q_\theta(\mathbf{x}, \hat{y})}{p^2(\mathbf{x})} d\mathbf{x} dy\right).
\end{aligned}
\tag{27}
$$

**[Estimation of the conditional quadratic terms $\int \frac{p^2(\mathbf{x},y)}{p^2(\mathbf{x})} d\mathbf{x} dy$ and $\int \frac{q_\theta^2(\mathbf{x},\hat{y})}{p^2(\mathbf{x})} d\mathbf{x} dy$]**

The empirical estimation of $\int \frac{p^2(\mathbf{x},y)}{p^2(\mathbf{x})} d\mathbf{x} dy$ can be expressed as:

$$
\int \frac{p^2(\mathbf{x}, y)}{p^2(\mathbf{x})} d\mathbf{x} dy = \mathbb{E}_{p(\mathbf{x},y)}\left[\frac{p(\mathbf{x}, y)}{p^2(\mathbf{x})}\right] \approx \frac{1}{N}\sum_{j=1}^{N} \frac{p(\mathbf{x}_j, y_j)}{p^2(\mathbf{x}_j)}.
\tag{28}
$$

By kernel density estimator (KDE), we have:

$$
\frac{p(\mathbf{x}_j, y_j)}{p^2(\mathbf{x}_j)} \approx N \frac{\sum_{i=1}^{N} \kappa_\sigma(\mathbf{x}_j - \mathbf{x}_i) \kappa_\sigma(y_j - y_i)}{\left(\sum_{i=1}^{N} \kappa_\sigma(\mathbf{x}_j - \mathbf{x}_i)\right)^2}.
\tag{29}
$$

Therefore,

$$
\int \frac{p^2(\mathbf{x}, y)}{p^2(\mathbf{x})} d\mathbf{x} dy \approx \sum_{j=1}^{N}\left(\frac{\sum_{i=1}^{N} \kappa_\sigma(\mathbf{x}_j - \mathbf{x}_i) \kappa_\sigma(y_j - y_i)}{\left(\sum_{i=1}^{N} \kappa_\sigma(\mathbf{x}_j - \mathbf{x}_i)\right)^2}\right) = \sum_{j=1}^{N}\left(\frac{\sum_{i=1}^{N} K_{ji} L_{ji}^1}{(\sum_{i=1}^{N} K_{ji})^2}\right).
\tag{30}
$$

Similarly, the empirical estimation of $\int \frac{q_\theta^2(\mathbf{x},\hat{y})}{p^2(\mathbf{x})} d\mathbf{x} dy$ is given by:

$$
\int \frac{q_\theta^2(\mathbf{x}, \hat{y})}{p^2(\mathbf{x})} d\mathbf{x} dy \approx \sum_{j=1}^{N}\left(\frac{\sum_{i=1}^{N} \kappa_\sigma(\mathbf{x}_j - \mathbf{x}_i) \kappa_\sigma(\hat{y}_j - \hat{y}_i)}{\left(\sum_{i=1}^{N} \kappa_\sigma(\mathbf{x}_j - \mathbf{x}_i)\right)^2}\right) = \sum_{j=1}^{N}\left(\frac{\sum_{i=1}^{N} K_{ji} L_{ji}^2}{(\sum_{i=1}^{N} K_{ji})^2}\right).
\tag{31}
$$

**[Estimation of the cross term $\int \frac{p(\mathbf{x},y) q_\theta(\mathbf{x},\hat{y})}{p^2(\mathbf{x})} d\mathbf{x} dy$]**

The empirical estimation of $\int \frac{p(\mathbf{x},y) q_\theta(\mathbf{x},\hat{y})}{p^2(\mathbf{x})} d\mathbf{x} dy$ can be expressed as:

$$
\int \frac{p(\mathbf{x}, y) q_\theta(\mathbf{x}, \hat{y})}{p^2(\mathbf{x})} d\mathbf{x} dy = \mathbb{E}_{p(\mathbf{x},y)}\left[\frac{q_\theta(\mathbf{x}, \hat{y})}{p^2(\mathbf{x})}\right] \approx \frac{1}{N}\sum_{j=1}^{N} \frac{q_\theta(\mathbf{x}_j, \hat{y}_j)}{p^2(\mathbf{x}_j)}.
\tag{32}
$$

---

[4]$p(\mathbf{x}, y) = p(y|\mathbf{x}) p(\mathbf{x})$ and $q_\theta(\mathbf{x}, \hat{y}) = q_\theta(\hat{y}|\mathbf{x}) p(\mathbf{x})$.

By KDE, we further have:

$$\frac{q_\theta(\mathbf{x}_j, \hat{y}_j)}{p^2(\mathbf{x}_j)} \approx N \frac{\sum_{i=1}^N \kappa_\sigma(\mathbf{x}_j - \mathbf{x}_i) \kappa_\sigma(\hat{y}_j - y_i)}{\left(\sum_{i=1}^N \kappa_\sigma(\mathbf{x}_j - \mathbf{x}_i)\right)^2}. \tag{33}$$

Therefore,

$$\int_\mathcal{X} \int_\mathcal{Y} \frac{p_s(\mathbf{x}, \mathbf{y}) p_t(\mathbf{x}, \mathbf{y})}{p_s(\mathbf{x}) p_t(\mathbf{x})} d\mathbf{x} d\mathbf{y} \approx \sum_{j=1}^N \left( \frac{\sum_{i=1}^N \kappa_\sigma(\mathbf{x}_j - \mathbf{x}_i) \kappa_\sigma(\hat{y}_j - y_i)}{\left(\sum_{i=1}^N \kappa_\sigma(\mathbf{x}_j - \mathbf{x}_i)\right)^2} \right) = \sum_{j=1}^N \left( \frac{\sum_{i=1}^N K_{ji} L_{ji}^{21}}{(\sum_{i=1}^N K_{ji})^2} \right). \tag{34}$$

Combine Eqs. (30), (31) and (34) with Eq. (27), an empirical estimation to $D_{\mathrm{CS}}(p(y|\mathbf{x}); q_\theta(\hat{y}|\mathbf{x}))$ is given by:

$$\widehat{D}_{\mathrm{CS}}(p(y|\mathbf{x}); q_\theta(\hat{y}|\mathbf{x})) = \log \left( \sum_{j=1}^N \left( \frac{\sum_{i=1}^N K_{ji} L_{ji}^1}{(\sum_{i=1}^N K_{ji})^2} \right) \right)$$
$$+ \log \left( \sum_{j=1}^N \left( \frac{\sum_{i=1}^N K_{ji} L_{ji}^2}{(\sum_{i=1}^N K_{ji})^2} \right) \right) - 2 \log \left( \sum_{j=1}^N \left( \frac{\sum_{i=1}^N K_{ji} L_{ji}^{21}}{(\sum_{i=1}^N K_{ji})^2} \right) \right). \tag{35}$$

$\square$

### A.3 PROOF TO PROPOSITION 3

**Proposition 3** (Empirical Estimator of CS-QMI). *Given $N$ pairs of observations $\{(\mathbf{x}_i, \mathbf{t}_i)\}_{i=1}^N$, each sample contains two different types of measurements $\mathbf{x} \in \mathcal{X}$ and $\mathbf{t} \in \mathcal{T}$ obtained from the same realization. Let $K$ and $Q$ denote, respectively, the Gram matrices for variable $\mathbf{x}$ and variable $\mathbf{t}$. The empirical estimator of CS-QMI is given by:*

$$\widehat{I}_{CS}(\mathbf{x}; \mathbf{t}) = \log \left( \frac{1}{N^2} \sum_{i,j}^N K_{ij} Q_{ij} \right) + \log \left( \frac{1}{N^4} \sum_{i,j,q,r}^N K_{ij} Q_{qr} \right) - 2 \log \left( \frac{1}{N^3} \sum_{i,j,q}^N K_{ij} Q_{iq} \right)$$
$$= \log \left( \frac{1}{N^2} \mathrm{tr}(KQ) \right) + \log \left( \frac{1}{N^4} \mathbb{1}^T K \mathbb{1} \mathbb{1}^T Q \mathbb{1} \right) - 2 \log \left( \frac{1}{N^3} \mathbb{1}^T K Q \mathbb{1} \right). \tag{36}$$

*Proof.* By definition, we have:

$$I_{\mathrm{CS}}(\mathbf{x}, \mathbf{t}) = D_{\mathrm{CS}}(p(\mathbf{x}, \mathbf{t}); p(\mathbf{x}) p(\mathbf{t})) = -\log \left( \frac{\left| \int p(\mathbf{x}, \mathbf{t}) p(\mathbf{x}) p(\mathbf{t}) d\mathbf{x} d\mathbf{t} \right|^2}{\int p^2(\mathbf{x}, \mathbf{t}) d\mathbf{x} d\mathbf{t} \int p^2(\mathbf{x}) p^2(\mathbf{t}) d\mathbf{x} d\mathbf{t}} \right)$$
$$= \log \left( \int p^2(\mathbf{x}, \mathbf{t}) d\mathbf{x} d\mathbf{t} \right) + \log \left( \int p^2(\mathbf{x}) p^2(\mathbf{t}) d\mathbf{x} d\mathbf{t} \right) - 2 \log \left( \int p(\mathbf{x}, \mathbf{t}) p(\mathbf{x}) p(\mathbf{t}) d\mathbf{x} d\mathbf{t} \right) \tag{37}$$

Again, all three terms inside the "log" can be estimated by KDE as follows,

$$\int p^2(\mathbf{x}, \mathbf{t}) d\mathbf{x} d\mathbf{t} = \frac{1}{N^2} \sum_{i=1}^N \sum_{j=1}^N \kappa(\mathbf{x}_i - \mathbf{x}_j) \kappa(\mathbf{t}_i - \mathbf{t}_j) = \frac{1}{N^2} \sum_{i,j}^N K_{ij} Q_{ij}, \tag{38}$$

$$\int p(\mathbf{x}, \mathbf{t}) p(\mathbf{x}) p(\mathbf{t}) d\mathbf{x} d\mathbf{t} = \mathbb{E}_{p(\mathbf{x}, \mathbf{t})} \left[ p(\mathbf{x}) p(\mathbf{t}) \right]$$

$$= \frac{1}{N} \sum_{i=1}^{N} \left[ \left( \frac{1}{N} \sum_{j=1}^{N} \kappa(\mathbf{x}_i - \mathbf{x}_j) \right) \left( \frac{1}{N} \sum_{q=1}^{N} \kappa(\mathbf{t}_i - \mathbf{t}_q) \right) \right]$$

$$= \frac{1}{N^3} \sum_{i=1}^{N} \sum_{j=1}^{N} \sum_{q=1}^{N} \kappa(\mathbf{x}_i - \mathbf{x}_q) \kappa(\mathbf{t}_i - \mathbf{t}_q) \tag{39}$$

$$= \frac{1}{N^3} \sum_{i,j,q}^{N} K_{ij} Q_{iq},$$

$$\int p^2(\mathbf{x}) p^2(\mathbf{t}) d\mathbf{x} d\mathbf{t} = \left[ \frac{1}{N^2} \sum_{i=1}^{N} \sum_{j=1}^{N} \kappa(\mathbf{x}_i - \mathbf{x}_j) \right] \left[ \frac{1}{N^2} \sum_{q=1}^{N} \sum_{r=1}^{N} \kappa(\mathbf{t}_q - \mathbf{t}_r) \right] \tag{40}$$

$$= \frac{1}{N^4} \sum_{i=1}^{N} \sum_{j=1}^{N} \sum_{q=1}^{N} \sum_{r=1}^{N} \kappa(\mathbf{x}_i - \mathbf{x}_j) \kappa(\mathbf{t}_q - \mathbf{t}_r) = \frac{1}{N^4} \sum_{i,j,q,r}^{N} K_{ij} Q_{qr}.$$

By plugging Eqs. (38)-(40) into Eq. (37), we obtain:

$$\widehat{I}_{\text{CS}}(\mathbf{x}; \mathbf{t}) = \log \left( \frac{1}{N^2} \sum_{i,j}^{N} K_{ij} Q_{ij} \right) + \log \left( \frac{1}{N^4} \sum_{i,j,q,r}^{N} K_{ij} Q_{qr} \right) - 2 \log \left( \frac{1}{N^3} \sum_{i,j,q}^{N} K_{ij} Q_{iq} \right). \tag{41}$$

An exact and naïve computation of Eq. (41) would require $\mathcal{O}(N^4)$ operations. However, an equivalent form which needs $\mathcal{O}(n^2)$ operations can be formulated as (both $K$ and $Q$ are symmetric):

$$\widehat{I}_{\text{CS}}(\mathbf{x}; \mathbf{t}) = \log \left( \frac{1}{N^2} \operatorname{tr}(KQ) \right) + \log \left( \frac{1}{N^4} \mathbb{1}^T K \mathbb{1} \mathbb{1}^T Q \mathbb{1} \right) - 2 \log \left( \frac{1}{N^3} \mathbb{1}^T K Q \mathbb{1} \right). \tag{42}$$

where $\mathbb{1}$ is a vector of 1s of relevant dimension.

$\square$

### A.4  PROOF TO PROPOSITION 4

Given a network $h_\theta = g(f(\mathbf{x}))$, where $f : \mathbb{R}^{d_X} \mapsto \mathbb{R}^{d_T}$ maps the input to an intermediate layer representation $\mathbf{t}$, and $g : \mathbb{R}^{d_T} \mapsto \mathbb{R}$ maps this intermediate representation $\mathbf{t}$ to the final layer, we assume all functions $h_\theta$ and $g$ we consider are uniformly bounded by $M_\mathcal{X}$ and $M_\mathcal{Z}$, respectively. Let us denote $\mathcal{F}$ and $\mathcal{G}$ the induced RKHSs for kernels $\kappa_X$ and $\kappa_Z$, and assume all functions in $\mathcal{F}$ and $\mathcal{G}$ are uniformly bounded by $M_\mathcal{F}$ and $M_\mathcal{G}$. Based on Remark 2 and the result by (Wang et al., 2021), let $\mu(\mathbb{P}_{XT})$ and $\mu(\mathbb{P}_X \otimes \mathbb{P}_T)$ denote, respectively, the (empirical) kernel mean embedding of $\mathbb{P}_{XT}$ and $\mathbb{P}_X \otimes \mathbb{P}_T$ in the RHKS $\mathcal{F} \otimes \mathcal{G}$, CS-QMI bounds the power of an arbitrary adversary in $\mathcal{S}_r$ when $\sqrt{N}$ is sufficiently large, in which $\mathcal{S}_r$ is a $\ell_\infty$-ball of radius $r$, i.e., $\mathcal{S}_r = \{\delta \in \mathbb{R}^{d_X}, \delta_\infty \leq r\}$.

**Proposition 4.** *Denote* $\gamma = \frac{\sigma M_\mathcal{F} M_\mathcal{G}}{r \sqrt{-2 \log o(1) d_X M_\mathcal{Z}}} \left( \mathbb{E}[|h_\theta(\mathbf{x} + \delta) - h_\theta(\mathbf{x})|] - o(r) \right)$, *if* $\mathbf{x} \sim \mathcal{N}(0, \sigma^2 I)$ *and* $\|\hat{\mu}(\mathbb{P}_{XT})\|_{\mathcal{F} \otimes \mathcal{G}} = \|\hat{\mu}(\mathbb{P}_X \otimes \mathbb{P}_T)\|_{\mathcal{F} \otimes \mathcal{G}} = \|\hat{\mu}\|$, *then:*

$$\mathbb{P} \left( \widehat{I}_{CS}(\mathbf{x}; \mathbf{t}) \geq g(\gamma) \right) = 1 - \Phi \left( \frac{\sqrt{N}(g(\gamma) - g(HSIC(\mathbf{x}; \mathbf{t})))}{|g'(HSIC(\mathbf{x}; \mathbf{t}))|\sigma_H} \right), \tag{43}$$

*in which* $g(x) = -2 \log(1 - x/(2\|\hat{\mu}\|^2))$ *is a monotonically increasing function,* $\Phi$ *is the cumulative distribution function of a standard Gaussian, and* $\sqrt{N}(\widehat{HSIC}_b(\mathbf{x}; \mathbf{t}) - HSIC(\mathbf{x}; \mathbf{t})) \xrightarrow{D} \mathcal{N}(0, \sigma_H^2)$.

*Proof.* We first provide the following two Lemmas.

**Lemma 1** (asymptotic distribution of $HSIC_b(\mathbf{x};\mathbf{t})$ when $\mathbb{P}_{XT} \neq \mathbb{P}_X \otimes \mathbb{P}_T$ (Gretton et al., 2007))**.** *Let*

$$h_{ijqr} = \frac{1}{4!} \sum_{t,u,v,w}^{i,j,q,r} K_{tu}Q_{tu} + K_{tu}Q_{vw} - 2K_{tu}Q_{tv}, \tag{44}$$

*where the sum represents all ordered quadruples $(t, u, v, w)$ drawn without replacement from $(i, j, q, r)$, and assume $\mathbb{E}(h^2) \leq \infty$. Under $\mathcal{H}_1$ that is $\mathbb{P}_{XT} \neq \mathbb{P}_X \otimes \mathbb{P}_T$[5], $HSIC_b(\mathbf{x};\mathbf{t})$ converges in distribution as $N \to \infty$ to a Gaussian according to:*

$$\sqrt{N}(\widehat{HSIC}_b(\mathbf{x};\mathbf{t}) - HSIC(\mathbf{x};\mathbf{t})) \xrightarrow{D} \mathcal{N}(0, \sigma_H^2), \tag{45}$$

*where $\sigma_H^2 = 16 \left( \mathbb{E}_i (\mathbb{E}_{j,q,r} H_{ijqr})^2 - HSIC(\mathbf{x};\mathbf{t}) \right)$.*

**Lemma 2** (adversarial robustness guarantee of $HSIC(\mathbf{x};\mathbf{t})$ (Wang et al., 2021))**.** *Assume $\mathbf{x} \sim \mathcal{N}(0, \sigma^2 I)$. Also assume all functions $h_\theta$ and $g$ we consider are uniformly bounded respectively by $M_{\mathcal{X}}$ and $M_{\mathcal{Z}}$, and all functions in $\mathcal{F}$ and $\mathcal{G}$ are uniformly bounded respectively by $M_{\mathcal{F}}$ and $M_{\mathcal{G}}$[6], then:*

$$\frac{r\sqrt{-2\log o(1)}d_X M_{\mathcal{Z}}}{\sigma M_{\mathcal{F}} M_{\mathcal{G}}} HSIC(\mathbf{x};\mathbf{t}) + o(r) \geq \mathbb{E}[|h_\theta(\mathbf{x}+\delta) - h_\theta(\mathbf{x})|], \tag{46}$$

*for all $\delta \in \mathcal{S}_r$.*

By Remark 2, for samples $\{\mathbf{x}_i^p\}_{i=1}^m$ and $\{\mathbf{x}_i^q\}_{i=1}^n$, drawn *i.i.d.* from respectively any two square-integral distributions $p$ and $q$,

$$\widehat{D}_{\mathrm{CS}}(p;q) = -2\log\left(\frac{\langle \boldsymbol{\mu}_p, \boldsymbol{\mu}_q \rangle_{\mathcal{H}}}{\|\boldsymbol{\mu}_p\|_{\mathcal{H}}\|\boldsymbol{\mu}_q\|_{\mathcal{H}}}\right) = -2\log\cos(\boldsymbol{\mu}_p, \boldsymbol{\mu}_q), \tag{47}$$

and

$$\widehat{\mathrm{MMD}}^2(p;q) = \langle \boldsymbol{\mu}_p, \boldsymbol{\mu}_q \rangle_{\mathcal{H}}^2 = \|\boldsymbol{\mu}_p\|_{\mathcal{H}}^2 + \|\boldsymbol{\mu}_q\|_{\mathcal{H}}^2 - 2\langle \boldsymbol{\mu}_p, \boldsymbol{\mu}_q \rangle_{\mathcal{H}}, \tag{48}$$

in which $\boldsymbol{\mu}_p$ and $\boldsymbol{\mu}_q$ refer to, respectively, the (empirical) kernel mean embeddings of $\{\mathbf{x}_i^p\}_{i=1}^m$ and $\{\mathbf{x}_i^q\}_{i=1}^n$ in the Reproducing kernel Hilbert space (RKHS) $\mathcal{H}$.

Let $p = \mathbb{P}_{XT}$ (the joint distribution) and $q = \mathbb{P}_X \otimes \mathbb{P}_T$ (the product of marginal distributions). Further, let $\mu(\mathbb{P}_{XT})$ and $\mu(\mathbb{P}_X \otimes \mathbb{P}_T)$ denote, respectively, the (empirical) kernel mean embedding of $\mathbb{P}_{XT}$ and $\mathbb{P}_X \otimes \mathbb{P}_T$ in RKHS $\mathcal{F} \otimes \mathcal{G}$, we have:

$$\widehat{I}_{\mathrm{CS}}(\mathbf{x};\mathbf{t}) = \widehat{D}_{\mathrm{CS}}(\mathbb{P}_{XT}; \mathbb{P}_X \otimes \mathbb{P}_T) = -2\log\left(\frac{\langle \mu(\mathbb{P}_{XT}), \mu(\mathbb{P}_X \otimes \mathbb{P}_T) \rangle_{\mathcal{F} \otimes \mathcal{G}}}{\|\mu(\mathbb{P}_{XT})\|_{\mathcal{F} \otimes \mathcal{G}}\|\mu(\mathbb{P}_X \otimes \mathbb{P}_T)\|_{\mathcal{F} \otimes \mathcal{G}}}\right), \tag{49}$$

and

$$\begin{aligned}
\widehat{\mathrm{HSIC}}(\mathbf{x};\mathbf{t}) &= \widehat{\mathrm{MMD}}^2(\mathbb{P}_{XT}; \mathbb{P}_X \otimes \mathbb{P}_T) \\
&= \|\mu(\mathbb{P}_{XT})\|_{\mathcal{F} \otimes \mathcal{G}}^2 + \|\mu(\mathbb{P}_X \otimes \mathbb{P}_T)\|_{\mathcal{F} \otimes \mathcal{G}}^2 - 2\langle \mu(\mathbb{P}_{XT}), \mu(\mathbb{P}_X \otimes \mathbb{P}_T) \rangle_{\mathcal{F} \otimes \mathcal{G}}.
\end{aligned} \tag{50}$$

If $\|\mu(\mathbb{P}_{XT})\|_{\mathcal{F} \otimes \mathcal{G}} = \|\mu(\mathbb{P}_X \otimes \mathbb{P}_T)\|_{\mathcal{F} \otimes \mathcal{G}} = \|\mu\|$ (i.e., the norm of the (empirical) kernel mean embeddings for $\mathbb{P}_{XT}$ and $\mathbb{P}_X \otimes \mathbb{P}_T$ is the same, please also refer to Fig. 2 for an geometrical interpretation), then:

$$\widehat{I}_{\mathrm{CS}}(\mathbf{x};\mathbf{t}) = g(\widehat{\mathrm{HSIC}}(\mathbf{x};\mathbf{t})) = -2\log(1 - \frac{\widehat{\mathrm{HSIC}}(\mathbf{x};\mathbf{t})}{2\|\hat{\mu}\|^2}). \tag{51}$$

Here, $g(x)$ is a monotonically increasing function.

---

[5]In our application, $\mathbf{x}$ and $\mathbf{t}$ will never be completely independent. Otherwise, the latent representation $\mathbf{t}$ learns no meaningful information from $\mathbf{x}$ and the training fails.

[6]We refer interested readers to Assumptions 1 and 2 in (Wang et al., 2021) for the mathematical formulations regarding these two assumptions.

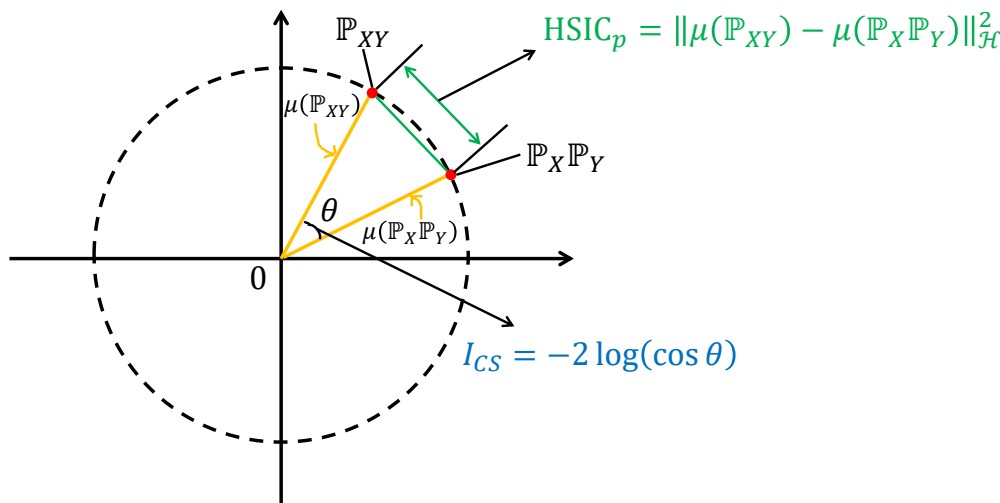

Figure 2: Geometrical interpretation of CS-QMI and HSIC, in which $\mathcal{H} = \mathcal{F} \otimes \mathcal{G}$. When $\|\mu(\mathbb{P}_{XT})\|_{\mathcal{H}} = \|\mu(\mathbb{P}_X \otimes \mathbb{P}_T)\|_{\mathcal{H}} = \|\mu\|$, CS-QMI and HSIC has a monotonic relationship.

By applying the delta method to Lemma 1, we obtain:

$$\sqrt{N}(\widehat{I}_{\text{CS}}(\mathbf{x}; \mathbf{t}) - g(\text{HSIC}(\mathbf{x}; \mathbf{t})) \xrightarrow{D} \mathcal{N}(0, [g'(\text{HSIC}(\mathbf{x}; \mathbf{t}))]^2 \sigma_H^2). \tag{52}$$

Here, $g'(x) = \frac{1}{\|\mu\|^2 - x/2}$ exists and is always non-zero if $\|\hat{\mu}(\mathbb{P}_{XT})\|_{\mathcal{F} \otimes \mathcal{G}}$ and $\|\hat{\mu}(\mathbb{P}_X \otimes \mathbb{P}_T)\|_{\mathcal{F} \otimes \mathcal{G}}$ are not orthogonal.

Then, by applying Lemma 2,

$$\text{HSIC}(\mathbf{x}; \mathbf{t}) \geq \frac{\sigma M_{\mathcal{F}} M_{\mathcal{G}}}{r\sqrt{-2\log o(1)} d_X M_{\mathcal{Z}}} \left(\mathbb{E}[|h_\theta(\mathbf{x} + \delta) - h_\theta(\mathbf{x})|] - o(r)\right) = \gamma. \tag{53}$$

Hence,

$$g(\text{HSIC}(\mathbf{x}; \mathbf{t})) \geq g(\gamma). \tag{54}$$

Then,

$$\mathbb{P}\left(\widehat{I}_{\text{CS}}(\mathbf{x}; \mathbf{t}) \geq g(\gamma)\right) = \mathbb{P}(\sqrt{N}(\widehat{I}_{\text{CS}}(\mathbf{x}; \mathbf{t}) - g(\text{HSIC}(\mathbf{x}; \mathbf{t})) \geq \sqrt{N}(g(\gamma) - g(\text{HSIC}(\mathbf{x}; \mathbf{t})))$$

$$\approx 1 - \Phi\left(\frac{\sqrt{N}(g(\gamma) - g(\text{HSIC}(\mathbf{x}; \mathbf{t})))}{|g'(\text{HSIC}(\mathbf{x}; \mathbf{t}))|\sigma_H}\right). \quad \text{By applying Eq. (52).} \tag{55}$$

By Lemma 2,

$$g(\gamma) - g(\text{HSIC}(\mathbf{x}; \mathbf{t})) \leq 0. \tag{56}$$

Hence, $\mathbb{P}\left(\widehat{I}_{\text{CS}}(\mathbf{x}; \mathbf{t}) \geq g(\gamma)\right)$ is at least $0.5$.

On the other hand, when $\sqrt{N} \gg |g'(\text{HSIC}(\mathbf{x}; \mathbf{t}))|\sigma_H$, we have:

$$\mathbb{P}\left(\widehat{I}_{\text{CS}}(\mathbf{x}; \mathbf{t}) \geq g(\gamma)\right) \to 1. \tag{57}$$

$\square$

In fact, even when the condition $\|\mu(\mathbb{P}_{XT})\|_{\mathcal{H}} = \|\mu(\mathbb{P}_X \otimes \mathbb{P}_T)\|_{\mathcal{H}} = \|\mu\|$ does not hold, the following Lemma provides a supplement.

**Lemma 3.** *Minimizing the empirical estimator of CS-QMI (i.e., Eq. (19)) implies the minimization of that of HSIC.*

*Proof.* First observe that the functional relationship between the estimator of CS-QMI and HSIC is equivalent to the functional relationship between the estimator of the Cauchy–Schwarz divergence and squared MMD. This is due to HSIC being defined as MMD between two specific distributions, and CS-QMI being defined as the CS-Divergence between the same two distributions. Without loss of generality, we base this analysis on the Cauchy-Schwarz divergence and squared MMD.

Given samples $\mathcal{X}^{(p)} = \{\mathbf{x}_i^p\}_{i=1}^m$, $\mathcal{X}^{(q)} = \{\mathbf{x}_i^q\}_{i=1}^n$ and $\mathcal{X} = \mathcal{X}^{(p)} \cup \mathcal{X}^{(q)} = \{\mathbf{x}_j\}_{j=1}^{m+n}$, we have:

$$
\begin{aligned}
\widehat{D}_{\mathrm{CS}}(p;q) &= \log\left(\|\boldsymbol{\mu}_p\|_{\mathcal{H}}^2\right) + \log\left(\|\boldsymbol{\mu}_q\|_{\mathcal{H}}^2\right) - 2\log\left(\langle\boldsymbol{\mu}_p,\boldsymbol{\mu}_q\rangle_{\mathcal{H}}\right) \\
&= -2\log\left(\frac{\langle\boldsymbol{\mu}_p,\boldsymbol{\mu}_q\rangle_{\mathcal{H}}}{\|\boldsymbol{\mu}_p\|_{\mathcal{H}}\|\boldsymbol{\mu}_q\|_{\mathcal{H}}}\right) = -2\log\cos(\boldsymbol{\mu}_p,\boldsymbol{\mu}_q),
\end{aligned}
\tag{58}
$$

and

$$
\widehat{\mathrm{MMD}}^2(p;q) = \|\boldsymbol{\mu}_p - \boldsymbol{\mu}_q\|_{\mathcal{H}}^2 = \|\boldsymbol{\mu}_p\|_{\mathcal{H}}^2 + \|\boldsymbol{\mu}_q\|_{\mathcal{H}}^2 - 2\langle\boldsymbol{\mu}_p,\boldsymbol{\mu}_q\rangle_{\mathcal{H}},
\tag{59}
$$

in which $\boldsymbol{\mu}_p = \frac{1}{m}\sum_{i=1}^m \phi(\mathbf{x}_i^p)$ and $\boldsymbol{\mu}_q = \frac{1}{n}\sum_{i=1}^n \phi(\mathbf{x}_i^q)$ refer to, respectively, the (empirical) mean embeddings of $\mathcal{X}^{(p)}$ and $\mathcal{X}^{(q)}$ in $\mathcal{H}$.

Therefore, we have:

$$
\frac{\partial\widehat{D}_{\mathrm{CS}}}{\partial\mathbf{x}_j} = \frac{1}{\|\boldsymbol{\mu}_p\|_{\mathcal{H}}^2}\frac{\partial\|\boldsymbol{\mu}_p\|_{\mathcal{H}}^2}{\partial\mathbf{x}_j} + \frac{1}{\|\boldsymbol{\mu}_q\|_{\mathcal{H}}^2}\frac{\partial\|\boldsymbol{\mu}_q\|_{\mathcal{H}}^2}{\partial\mathbf{x}_j} - \frac{2}{\langle\boldsymbol{\mu}_p,\boldsymbol{\mu}_q\rangle_{\mathcal{H}}}\frac{\partial\langle\boldsymbol{\mu}_p,\boldsymbol{\mu}_q\rangle_{\mathcal{H}}}{\partial\mathbf{x}_j},
\tag{60}
$$

and

$$
\frac{\partial\widehat{\mathrm{MMD}}^2}{\partial\mathbf{x}_j} = \frac{\partial\|\boldsymbol{\mu}_p\|_{\mathcal{H}}^2}{\partial\mathbf{x}_j} + \frac{\partial\|\boldsymbol{\mu}_q\|_{\mathcal{H}}^2}{\partial\mathbf{x}_j} - \frac{2\partial\langle\boldsymbol{\mu}_p,\boldsymbol{\mu}_q\rangle_{\mathcal{H}}}{\partial\mathbf{x}_j}.
\tag{61}
$$

Thus, for each term in $\frac{\partial\widehat{\mathrm{MMD}}^2}{\partial\mathbf{x}_j}$, the "log" on CS just scales the gradient contribution, but does not change the direction. That is, taking one step with gradient descent of $\widehat{D}_{\mathrm{CS}}$ will decrease $\widehat{\mathrm{MMD}}^2$.

To corroborate our analysis, we implement a CS Divergence optimization simulation. The setup is as follows: initializing two datasets, one fixed (denote by $\mathbf{X}_1$), and one for optimization (denoted by $\mathbf{X}_2$). The goal is to minimize the CS divergence between the probability density function (PDF) of $\mathbf{X}_2$ and the PDF of $\mathbf{X}_1$ by performing updates on $\mathbf{X}_2$ using gradient descent.

$\mathbf{X}_1$ was initialized as a $2d$, from a mixture of two Gaussian distributions, *i.e.*, $\mathcal{N}_1(\mu_1, \Sigma_1)$ and $\mathcal{N}_2(\mu_2, \Sigma_2)$, where $\mu_1 = [-4, -4]^\top$, $\mu_2 = [4, 4]^\top$, and $\Sigma_1 = \Sigma_2 = \mathrm{diag}(\sigma^2)$, where $\sigma = 1$. The number of samples $N_1 = 400$.

$\mathbf{X}_2$ was initialized as a $2d$, from a single Gaussian distribution, *i.e.*, $\mathcal{N}(\mu, \Sigma)$, where $\mu = [0, 0]^\top$, and $\Sigma = \mathrm{diag}(\sigma^2)$, where $\sigma = 1$. The number of samples $N_2 = 200$. The kernel width was set to 1, and the learning rate was set to 10.

We compute MMD for each step in the optimization. The results as shown in Fig. 3 are in line with our Lemma, as MMD is decreasing while performing gradient descent on $D_{\mathrm{CS}}$.

$\square$

## A.5   PROOF TO THEOREM 1

**Theorem 1.** *For two arbitrary $d$-variate Gaussian distributions $p \sim \mathcal{N}(\mu_1, \Sigma_1)$ and $q \sim \mathcal{N}(\mu_2, \Sigma_2)$,*

$$
D_{CS}(p;q) \leq \min\{D_{KL}(p;q), D_{KL}(q;p)\}.
\tag{62}
$$

*Proof.* Given two $d$-dimensional Gaussian distributions $p \sim \mathcal{N}(\mu_1, \Sigma_1)$ and $q \sim \mathcal{N}(\mu_2, \Sigma_2)$, the KL divergence for $p$ and $q$ is given by:

$$
D_{\mathrm{KL}}(p;q) = \frac{1}{2}\left(\mathrm{tr}(\Sigma_2^{-1}\Sigma_1) - d + (\mu_2 - \mu_1)^T\Sigma_2^{-1}(\mu_2 - \mu_1) + \ln\left(\frac{|\Sigma_2|}{|\Sigma_1|}\right)\right),
\tag{63}
$$

in which $|\cdot|$ is the determinant of matrix.

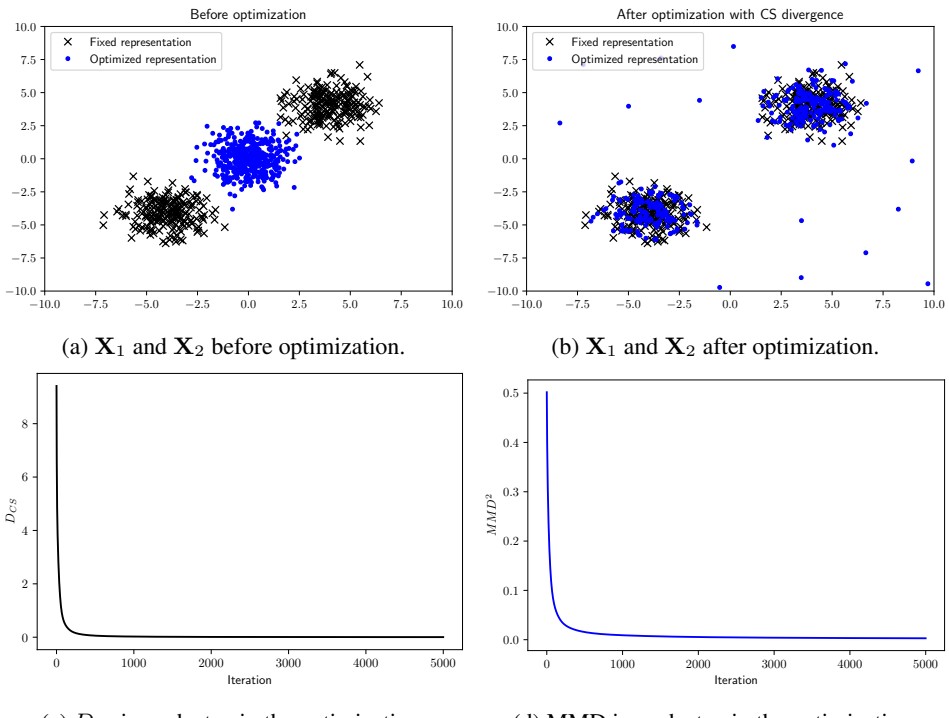

(a) $\mathbf{X}_1$ and $\mathbf{X}_2$ before optimization.

(b) $\mathbf{X}_1$ and $\mathbf{X}_2$ after optimization.

(c) $D_{\text{CS}}$ in each step in the optimization.

(d) MMD in each step in the optimization.

Figure 3: Lemma 3 supporting simulation.

The CS divergence for $p$ and $q$ is given by (Kampa et al., 2011)[7]:

$$D_{\text{CS}}(p;q) = -\log(z_{12}) + \frac{1}{2}\log(z_{11}) + \frac{1}{2}\log(z_{22}), \tag{64}$$

where

$$z_{12} = \mathcal{N}(\mu_1|\mu_2, (\Sigma_1 + \Sigma_2)) = \frac{\exp(-\frac{1}{2}(\mu_1 - \mu_2)^T)(\Sigma_1 + \Sigma_2)^{-1}(\mu_1 - \mu_2)}{\sqrt{(2\pi)^d|\Sigma_1 + \Sigma_2|}}$$

$$z_{11} = \frac{1}{\sqrt{(2\pi)^d|2\Sigma_1|}} \tag{65}$$

$$z_{22} = \frac{1}{\sqrt{(2\pi)^d|2\Sigma_2|}}.$$

Therefore,

$$
\begin{aligned}
D_{\text{CS}}(p;q) &= \frac{1}{2}(\mu_2 - \mu_1)^T(\Sigma_1 + \Sigma_2)^{-1}(\mu_2 - \mu_1) + \log(\sqrt{(2\pi)^d|\Sigma_1 + \Sigma_2|}) - \\
&\quad \frac{1}{2}\log(\sqrt{(2\pi)^d|2\Sigma_1|}) - \frac{1}{2}\log(\sqrt{(2\pi)^d|2\Sigma_2|}) \\
&= \frac{1}{2}(\mu_2 - \mu_1)^T(\Sigma_1 + \Sigma_2)^{-1}(\mu_2 - \mu_1) + \frac{1}{2}\ln\left(\frac{|\Sigma_1 + \Sigma_2|}{2^d\sqrt{|\Sigma_1||\Sigma_2|}}\right).
\end{aligned} \tag{66}
$$

We first consider the difference between $D_{\text{CS}}(p;q)$ and $D_{\text{KL}}(p;q)$ results from mean vector discrepancy, i.e., $\mu_1 - \mu_2$.

**Lemma 4.** *(Horn & Johnson, 2012) For any two positive semi-definite Hermitian matrices $A$ and $B$ of size $n \times n$, then $A - B$ is positive semi-definite if and only if $B^{-1} - A^{-1}$ is also positive semi-definite.*

---

[7]Note that, we apply a constant $1/2$ on the definition in the main manuscript.

Applying Lemma 4 to $(\Sigma_1 + \Sigma_2) - \Sigma_2$ (which is positive semi-definite), we obtain that $\Sigma_2^{-1} - (\Sigma_1 + \Sigma_2)^{-1}$ is also positive semi-definite, from which we obtain:

$$
\begin{aligned}
(\mu_2 - \mu_1)^T \Sigma_2^{-1} (\mu_2 - \mu_1) &- (\mu_2 - \mu_1)^T (\Sigma_1 + \Sigma_2)^{-1} (\mu_2 - \mu_1) \\
&= (\mu_2 - \mu_1)^T \left[ \Sigma_2^{-1} - (\Sigma_1 + \Sigma_2)^{-1} \right] (\mu_2 - \mu_1) \geq 0.
\end{aligned}
\tag{67}
$$

We then consider the difference between $D_{\text{CS}}(p;q)$ and $D_{\text{KL}}(p;q)$ results from covariance matrix discrepancy, i.e., $\Sigma_1 - \Sigma_2$.

We have,

$$
\begin{aligned}
2(D_{\text{CS}}(p;q) - D_{\text{KL}}(p;q))_{\mu_1 = \mu_2} &= \log\left( \frac{|\Sigma_1 + \Sigma_2|}{2^d \sqrt{|\Sigma_1||\Sigma_2|}} \right) - \log\left( \frac{|\Sigma_2|}{|\Sigma_1|} \right) - \text{tr}(\Sigma_2^{-1} \Sigma_1) + d. \\
&= -d \log 2 + \log\left( |\Sigma_1 + \Sigma_2| \right) - \frac{1}{2}(\log|\Sigma_1| + \log|\Sigma_2|) \\
&\quad - \log|\Sigma_2| + \log|\Sigma_1| - \text{tr}(\Sigma_2^{-1} \Sigma_1) + d \\
&= -d \log 2 + \log\left( \frac{|\Sigma_1 + \Sigma_2|}{|\Sigma_2|} \right) + \frac{1}{2}\log\left( \frac{|\Sigma_1|}{|\Sigma_2|} \right) - \text{tr}(\Sigma_2^{-1} \Sigma_1) + d \\
&= -d \log 2 + \log\left( |\Sigma_2^{-1} \Sigma_1 + I| \right) + \frac{1}{2}\log\left( |\Sigma_2^{-1} \Sigma_1| \right) - \text{tr}(\Sigma_2^{-1} \Sigma_1) + d
\end{aligned}
\tag{68}
$$

Let $\{\lambda_i\}_{i=1}^d$ denote the eigenvalues of $\Sigma_2^{-1} \Sigma_1$, which are non-negative, since $\Sigma_2^{-1} \Sigma_1$ is also positive semi-definite.

We have:

$$
|\Sigma_2^{-1} \Sigma_1| = \left[ \left( \prod_{i=1}^d \lambda_i \right)^{1/d} \right]^d \leq \left[ \frac{1}{d} \sum_{i=1}^d \lambda_i \right]^d = \left( \frac{1}{d} \text{tr}(\Sigma_2^{-1} \Sigma_1) \right)^d,
\tag{69}
$$

in which we use the property that geometric mean is no greater than the arithmetic mean.

Similarly, we have:

$$
|\Sigma_2^{-1} \Sigma_1 + I| = \prod_{i=1}^d (1 + \lambda_i) \leq \left[ \frac{1}{d} \sum_{i=1}^d (1 + \lambda_i) \right]^d = \left( 1 + \frac{1}{d} \text{tr}(\Sigma_2^{-1} \Sigma_1) \right)^d.
\tag{70}
$$

By plugging Eqs. (69) and (70) into Eq. (68), we obtain:

$$
\begin{aligned}
2(D_{\text{CS}}(p;q) - D_{\text{KL}}(p;q))_{\mu_1 = \mu_2} &= -d \log 2 + \log\left( |\Sigma_2^{-1} \Sigma_1 + I| \right) + \frac{1}{2}\log\left( |\Sigma_2^{-1} \Sigma_1| \right) - \text{tr}(\Sigma_2^{-1} \Sigma_1) + d \\
&\leq -d \log 2 + d \log(1 + \frac{1}{d}\text{tr}(\Sigma_2^{-1}\Sigma_1)) + \frac{d}{2}\log(\frac{1}{d}\text{tr}(\Sigma_2^{-1}\Sigma_1)) - \text{tr}(\Sigma_2^{-1}\Sigma_1) + d.
\end{aligned}
\tag{71}
$$

Let us denote $x = \text{tr}(\Sigma_2^{-1}\Sigma_1) = \sum_{i=1}^d \lambda_i \geq 0$, then

$$
2(D_{\text{CS}}(p;q) - D_{\text{KL}}(p;q))_{\mu_1 = \mu_2} = f(x) = -d\log 2 + d\log(1 + \frac{x}{d}) + \frac{d}{2}\log(\frac{x}{d}) - x + d. \tag{72}
$$

Let $f'(x) = 0$, we have $x = d$. Since $f''(x = d) < 0$, we have,

$$
2(D_{\text{CS}}(p;q) - D_{\text{KL}}(p;q))_{\mu_1 = \mu_2} = f(x) \leq f(x = d) = 0. \tag{73}
$$

Combining Eq. (67) with Eq. (73), we obtain:

$$
D_{\text{CS}}(p;q) - D_{\text{KL}}(p;q) \leq 0. \tag{74}
$$

The above analysis also applies to $D_{\text{KL}}(q;p)$.

Specifically, we have:

$$2(D_{\text{KL}}(q;p) - D_{\text{CS}}(p;q))_{\Sigma_1=\Sigma_2} = (\mu_2 - \mu_1)^T \left[\Sigma_1^{-1} - (\Sigma_1 + \Sigma_2)^{-1}\right](\mu_2 - \mu_1) \geq 0, \quad (75)$$

$$\begin{aligned}
2(D_{\text{CS}}(p;q) - D_{\text{KL}}(q;p))_{\mu_1=\mu_2} &= \log\left(\frac{|\Sigma_1 + \Sigma_2|}{2^d\sqrt{|\Sigma_1||\Sigma_2|}}\right) - \log\left(\frac{|\Sigma_1|}{|\Sigma_2|}\right) - \text{tr}(\Sigma_1^{-1}\Sigma_2) + d. \\
&= -d\log 2 + \log\left(|\Sigma_1 + \Sigma_2|\right) - \frac{1}{2}(\log|\Sigma_1| + \log|\Sigma_2|) \\
&\quad - \log|\Sigma_1| + \log|\Sigma_2| - \text{tr}(\Sigma_1^{-1}\Sigma_2) + d \\
&= -d\log 2 + \log\left(\frac{|\Sigma_1 + \Sigma_2|}{|\Sigma_1|}\right) + \frac{1}{2}\log\left(\frac{|\Sigma_2|}{|\Sigma_1|}\right) - \text{tr}(\Sigma_1^{-1}\Sigma_2) + d \\
&= -d\log 2 + \log\left(|\Sigma_1^{-1}\Sigma_2 + I|\right) + \frac{1}{2}\log\left(|\Sigma_1^{-1}\Sigma_2|\right) - \text{tr}(\Sigma_1^{-1}\Sigma_2) + d \\
&\leq -d\log 2 + d\log(1 + \frac{1}{d}\text{tr}(\Sigma_1^{-1}\Sigma_2)) + \frac{d}{2}\log(\frac{1}{d}\text{tr}(\Sigma_1^{-1}\Sigma_2)) - \text{tr}(\Sigma_1^{-1}\Sigma_2) + d \\
&\leq 0.
\end{aligned}$$
$$(76)$$

Hence,
$$D_{\text{CS}}(p;q) - D_{\text{KL}}(q;p) \leq 0. \tag{77}$$

Combining Eq. (74) and Eq. (77), we obtain:
$$D_{\text{CS}}(p;q) \leq \min\{D_{\text{KL}}(p;q), D_{\text{KL}}(q;p)\}. \tag{78}$$

$\square$

### A.6  PROOF TO COROLLARY 1

**Corollary 1.** *For two random vectors* $\mathbf{x}$ *and* $\mathbf{t}$ *which follow a joint Gaussian distribution* $\mathcal{N}\left(\begin{pmatrix}\mu_x \\ \mu_t\end{pmatrix}, \begin{pmatrix}\Sigma_x & \Sigma_{xt} \\ \Sigma_{tx} & \Sigma_t\end{pmatrix}\right)$, *the CS-QMI is no greater than the Shannon's mutual information:*

$$I_{CS}(\mathbf{x}; \mathbf{t}) \leq I(\mathbf{x}; \mathbf{t}). \tag{79}$$

*Proof.*
$$I_{\text{CS}}(\mathbf{x}; \mathbf{t}) = D_{\text{CS}}(p(\mathbf{x}, \mathbf{t}); p(\mathbf{x})p(\mathbf{t})), \tag{80}$$

$$I_{\text{KL}}(\mathbf{x}; \mathbf{t}) = D_{\text{KL}}(p(\mathbf{x}, \mathbf{t}); p(\mathbf{x})p(\mathbf{t})), \tag{81}$$

When $\mathbf{x}$ and $\mathbf{t}$ are jointly Gaussian,

$$p(\mathbf{x}, \mathbf{t}) \sim \mathcal{N}\left(\begin{pmatrix}\mu_x \\ \mu_t\end{pmatrix}, \begin{pmatrix}\Sigma_x & \Sigma_{xt} \\ \Sigma_{tx} & \Sigma_t\end{pmatrix}\right), \tag{82}$$

$$p(\mathbf{x})p(\mathbf{t}) \sim \mathcal{N}\left(\begin{pmatrix}\mu_x \\ \mu_t\end{pmatrix}, \begin{pmatrix}\Sigma_x & 0 \\ 0 & \Sigma_t\end{pmatrix}\right), \tag{83}$$

Applying Theorem 1 to $\mathcal{N}\left(\begin{pmatrix}\mu_x \\ \mu_t\end{pmatrix}, \begin{pmatrix}\Sigma_x & \Sigma_{xt} \\ \Sigma_{tx} & \Sigma_t\end{pmatrix}\right)$ and $\mathcal{N}\left(\begin{pmatrix}\mu_x \\ \mu_t\end{pmatrix}, \begin{pmatrix}\Sigma_x & 0 \\ 0 & \Sigma_t\end{pmatrix}\right)$, we obtain:

$$I_{\text{CS}}(\mathbf{x}; \mathbf{t}) \leq I(\mathbf{x}; \mathbf{t}). \tag{84}$$

$\square$

# B PROPERTIES AND ANALYSIS

## B.1 THE RELATIONSHIP BETWEEN CS DIVERGENCE, KL DIVERGENCE AND MMD

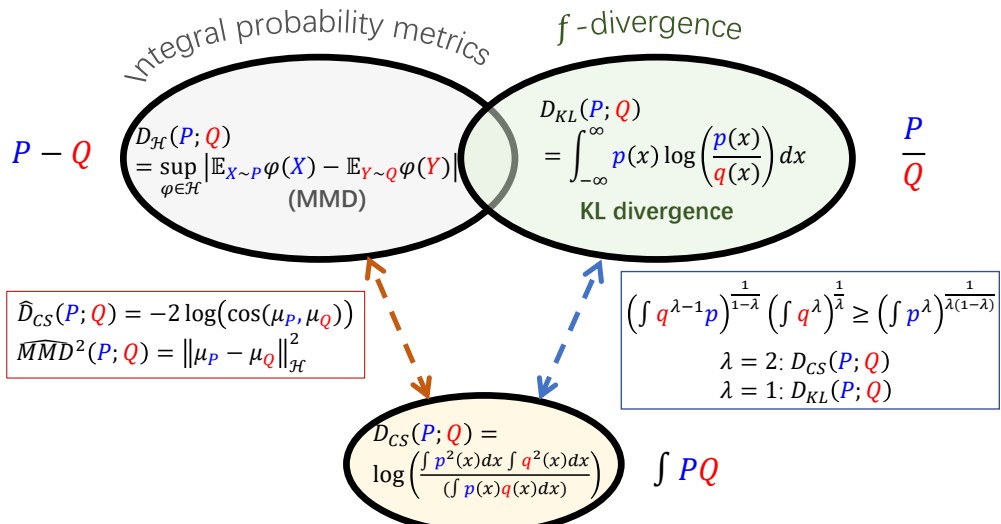

Figure 4: Connection between CS divergence with respect to MMD and KL divergence

There are two popular ways to define a valid divergence measure for two densities $p$ and $q$: the integral probability metrics (IPMs) aim to measure "$p - q$"; whereas the Rényi's divergences with order $\alpha$ aim to model the ratio of $\frac{p}{q}$ (see Fig. 4). The general idea of CS divergence is totally different: it neither horizontally computes $p - q$ nor vertically evaluates $\frac{p}{q}$. Rather, it quantifies the "distance" of two distributions by quantify the tightness (or gap) of an inequality associated with the integral of densities. That is, it is hard to say the CS divergence belongs to either IRM or the traditional Rényi's divergence family.

One of the inequalities one can consider here is the CS inequality:

$$| \int p(x)q(x)dx| \leq \left( \int p^2(x)dx \right)^{1/2} \left( \int q^2(x)dx \right)^{1/2}, \tag{85}$$

which directly motivates our CS divergence.

In fact, the CS inequality can be generalized to $L_p$ space by making use of the Hölder inequality:

$$| \int p(x)q(x)dx| \leq \left( \int p^a(x)dx \right)^{1/a} \left( \int q^b(x)dx \right)^{1/b} \quad \forall a, b > 1 \quad \text{and} \quad \frac{1}{a} + \frac{1}{b} = 1, \tag{86}$$

from which we can obtain a so-called Hölder divergence:

$$D_H = -\log \left( \frac{| \int p(x)q(x)dx|}{\left( \int p^a(x)dx \right)^{1/a} \left( \int q^b(x)dx \right)^{1/b}} \right), \tag{87}$$

although $D_H$ introduces one more hyperparameter.

One immediate advantage to define divergence in such a way (i.e., Eq. (10) and Eq. (87)) is that, compared to the basic KL divergence (or majority of divergences by measuring the ratio of $\frac{p}{q}$), the CS divergence is much more stable in the sense that it relaxes the constraint on the distribution supports, which makes it hard to reach a value of infinity.

In fact, for any two square-integral densities $p$ and $q$, the CS divergence goes to infinity if and only if there is no overlap on the supports of $p$ and $q$, i.e., $\text{supp}(p) \cap \text{supp}(q) = \emptyset$, since $\log 0$ is undefined. For $D_{\text{KL}}(p; q)$, it has finite values only if $\text{supp}(p) \subseteq \text{supp}(q)$ (note that, $p(x) \log \left( \frac{p(x)}{0} \right) \to \infty$ (Cover,

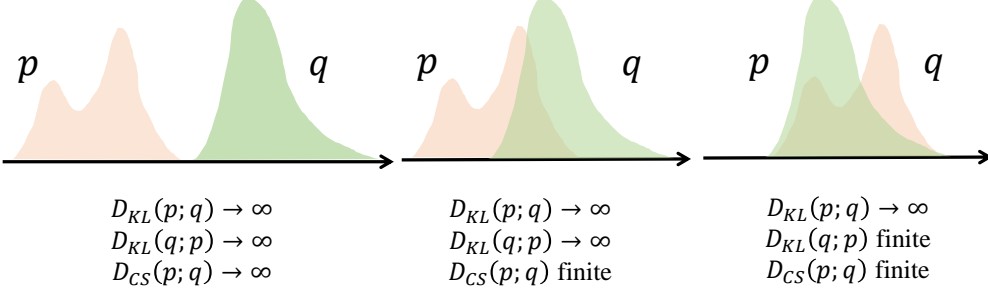

Figure 5: KL divergence is infinite even though there is an overlap between $\text{supp}(p)$ and $\text{supp}(q)$, but neither is a subset of the other. CS divergence does not has such support constraint.

1999)); whereas for $D_{\text{KL}}(q;p)$, it has finite values only if $\text{supp}(q) \subseteq \text{supp}(p)$. Please see Fig. 5 for an illustration.

Finally, we would like to emphasize here that, although CS divergence does not belong to either IRMs and $f$-divergence family. It has close connections to both of these two. On the one hand, if we rely on another inequality associated with two densities:

$$\left(\int q^{\lambda-1}p\right)^{\frac{1}{1-\lambda}} \left(\int q^{\lambda}\right)^{\frac{1}{\lambda}} \geq \left(\int p^{\lambda}\right)^{\frac{1}{\lambda(1-\lambda)}}. \tag{88}$$

It can be shown that when $\lambda = 2$, we obtain CS divergence; whereas when $\lambda = 1$, we obtain KL divergence.

On the other hand, from Remark 2, the empirical estimator of CS divergence measures the cosine similarity between (empirical) kernel mean embeddings of two densities; whereas MMD uses Euclidean distance.

## B.2 THE BIAS ANALYSIS

We discuss in this section the bias induced by variational approximations in previous literature. Let the conditional probability $p_\theta(\mathbf{t}|\mathbf{x})$ denote a parameterized encoding map with parameter $\theta$. Also let the conditional probability $p_\phi(y|\mathbf{t})$ denote a parameterized decoding map with parameter $\phi$.

### B.2.1 THE UPPER BOUND OF $I(\mathbf{x};\mathbf{t})$

$I_\theta(\mathbf{x};\mathbf{t})$ can be expressed as:

$$\begin{aligned} I_\theta(\mathbf{x};\mathbf{t}) &= \int p_\theta(\mathbf{x},\mathbf{t})\log\left(\frac{p_\theta(\mathbf{t}|\mathbf{x})}{p_\theta(\mathbf{t})}\right)d\mathbf{x}d\mathbf{t} \\ &\leq \int p_\theta(\mathbf{x},\mathbf{t})\log\left(\frac{p_\theta(\mathbf{t}|\mathbf{x})}{p_\theta(\mathbf{t})}\right)d\mathbf{x}d\mathbf{t} + \int p_\theta(\mathbf{t})\log\left(\frac{p_\theta(\mathbf{t})}{r(\mathbf{t})}\right)d\mathbf{t} \\ &= \int p_\theta(\mathbf{x},\mathbf{t})\log\left(\frac{p_\theta(\mathbf{t}|\mathbf{x})}{r(\mathbf{t})}\right)d\mathbf{x}d\mathbf{t}, \end{aligned} \tag{89}$$

in which $r(\mathbf{t})$ is a variational approximation to the marginal distribution $p_\theta(\mathbf{t}) = \int p(\mathbf{x})p_\theta(\mathbf{t}|\mathbf{x})d\mathbf{x}$.

As can be seen, the approximation error is exactly $D_{\text{KL}}(p_\theta(\mathbf{t}); r(\mathbf{t})) = \int p_\theta(\mathbf{t})\log\left(\frac{p_\theta(\mathbf{t})}{r(\mathbf{t})}\right)d\mathbf{t}$.

In VIB (Alemi et al., 2017) and lots of its downstream applications such as (Mahabadi et al., 2021), $r(\mathbf{t})$ is set as simple as a Gaussian. However, by our visualization in Fig. 6, $p_\theta(\mathbf{t})$ deviates a lot from a Gaussian.

Regarding the upper bound of $I(\mathbf{x};\mathbf{t})$ used in other existing methods, such as NIB (Kolchinsky et al., 2019b) and its extensions like exp-NIB (Rodríguez Gálvez et al., 2020), our argument is that these

methods may have similar bias than our non-parametric estimator on $I_{CS}(\mathbf{x}; \mathbf{t})$. This is because their estimators can also be interpreted from a kernel density estimation (KDE) perspective.

In fact, in NIB and exp-IB, $I_\theta(\mathbf{x}; \mathbf{t})$ is upper bounded by (Kolchinsky & Tracey, 2017):

$$I_\theta(\mathbf{x}; \mathbf{t}) \leq -\frac{1}{N} \sum_{i=1}^{N} \log \frac{1}{N} \sum_{j=1}^{N} \exp\left(-D_{KL}(p(t|x_i); p(t|x_j))\right), \tag{90}$$

which, in practice, is estimated as (Saxe et al., 2018):

$$I_\theta(\mathbf{x}; \mathbf{t}) \leq -\frac{1}{N} \sum_{i=1}^{N} \log \frac{1}{N} \sum_{j=1}^{N} \exp\left(-\frac{\|\mathbf{h}_i - \mathbf{h}_j\|_2^2}{2\sigma^2}\right), \tag{91}$$

in with $\mathbf{h}$ is the hidden activity corresponding to each input sample or the Gaussian center, i.e., $\mathbf{t} = \mathbf{h} + \epsilon$, and $\epsilon \sim \mathcal{N}(0, \sigma^2 I)$.

We argue that Eq. (91) is closely related to the KDE on the Shannon entropy of $\mathbf{t}$. Note that, the Shannon's differential entropy can be expressed as:

$$H(\mathbf{t}) = -\int p(\mathbf{t}) \log p(\mathbf{t}) d\mathbf{t} = -\mathbb{E}_p(\log p) \approx -\frac{1}{N} \sum_{i=1}^{N} \log p(\mathbf{t}_i). \tag{92}$$

By KDE with a Gaussian kernel of kernel size $\sigma$, we have:

$$p(\mathbf{t}_i) = \frac{1}{N} \sum_{i=1}^{N} \frac{1}{\sqrt{2\pi}\sigma} \exp\left(-\frac{\|\mathbf{t}_i - \mathbf{t}_j\|_2^2}{2\sigma^2}\right). \tag{93}$$

Hence,

$$H(\mathbf{t}) = -\frac{1}{N} \sum_{i=1}^{N} \log \frac{1}{N} \sum_{j=1}^{N} \frac{1}{\sqrt{2\pi}\sigma} \exp\left(-\frac{\|\mathbf{t}_i - \mathbf{t}_j\|_2^2}{2\sigma^2}\right). \tag{94}$$

Comparing Eq. (91) with Eq. (94), $\mathbf{h}$ is replaced with $\mathbf{t} = \mathbf{h} + \epsilon$, and there is a difference on the constant $\frac{1}{\sqrt{2\pi}\sigma}$.

From our perspective, using $\mathbf{t}$ rather than $\mathbf{h}$ seems to be a better choice. This is because,

$$\begin{aligned}
I(\mathbf{x}; \mathbf{t}) &= H(\mathbf{t}) - H(\mathbf{t}|\mathbf{x}) \\
&= H(\mathbf{t}) - H(\epsilon) = H(\mathbf{t}) - d(\log \sqrt{2\pi}\sigma + \frac{1}{2}).
\end{aligned} \tag{95}$$

Hence, if we keep $\sigma$ unchanged (during training) or take a value of $1/(\sqrt{2\pi}e)$, minimizing $H(\mathbf{t})$ amounts to minimize $I(\mathbf{x}; \mathbf{t})$. The only bias comes from the way to estimate $H(\mathbf{t})$.

To summarize, previous methods either have much more obvious bias than ours (severe mismatch between $r(\mathbf{t})$ and $p_\theta(\mathbf{t})$) or have similar bias (due to the same KDE in essence). But some improper choices (e.g., using $\mathbf{h}$ or $\mathbf{t}$, or varying noise variance) may incur other bias.

### B.2.2 THE LOWER BOUND OF $I(y; \mathbf{t})$

$I_\theta(y; \mathbf{t})$ can be expressed as:

$$\begin{aligned}
I_\theta(y; \mathbf{t}) &= H(p(y)) - H(p_\theta(y|\mathbf{t})) \\
&\geq H(p(y)) - H(p_\theta(y|\mathbf{t})) - D_{KL}(p_\theta(y|\mathbf{t}); p_\phi(y|\mathbf{t})) \\
&= H(p(y)) + \mathbb{E}_{p_\theta(y,\mathbf{t})}(p_\phi(y|\mathbf{t})),
\end{aligned} \tag{96}$$

where $p_\phi(y|\mathbf{t})$ is a variational approximation to $p_\theta(y|\mathbf{t})$. The smaller the KL divergence $D_{KL}(p_\theta(y|\mathbf{t}); p_\phi(y|\mathbf{t}))$, the smaller the approximation error.

Further, if we assume $p_\phi(y|\mathbf{t})$ follows a Gaussian, then we obtain MSE loss; whereas if we assume $p_\phi(y|\mathbf{t})$ follows a Laplacian, we obtain MAE loss.

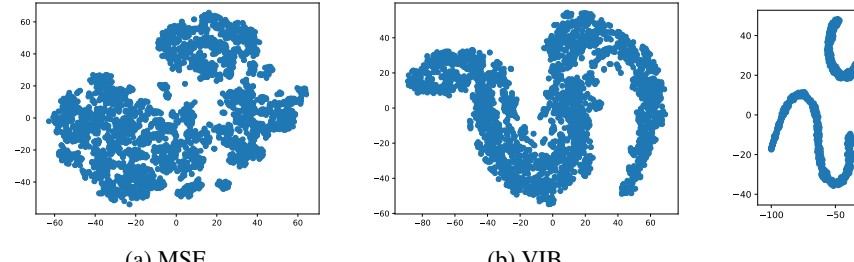

(a) MSE        (b) VIB        (c) NIB

Figure 6: PCA projection of bottleneck layer activations for models trained with (a) MSE loss only, (b) VIB, and (c) NIB on California housing.

Hence, the bias on estimating $I(y; \mathbf{t})$ simply by MSE comes from two sources: the closeness between $p_\phi(y|\mathbf{t})$ and $p_\theta(y|\mathbf{t})$, and the suitability of parametric Gaussian assumption.

Unfortunately, it is hard to justify or investigate if the bias from the first source is small or not. This is because (Alemi et al., 2017):

$$p_\theta(y|\mathbf{t}) = \int \frac{p_\theta(y|\mathbf{x})p_\theta(\mathbf{z}|\mathbf{x})p_\theta(\mathbf{x})}{p_\theta(\mathbf{z})} d\mathbf{x}, \tag{97}$$

which is intractable. On the other hand, although Gaussian assumption on $p_\phi(y|\mathbf{t})$ is the most popular choice, it is very likely that MAE (or losses induced by other parametric assumptions) has better performance.

To summarize, the bias term in estimating $I(y; \mathbf{t})$ by variational approximation is a bit hard to control. Moreover, there is no theoretical guarantee on properties like consistency. Fortunately, CS divergence with KDE does not have such limitations. Interested readers can refer to Section B.4.

## B.3    ADVANTAGES OF THE PREDICTION TERM $\widehat{D}_{\mathrm{CS}}(p(y|\mathbf{x}); q_\theta(\hat{y}|\mathbf{x}))$

Given observations $\{(\mathbf{x}_i, y_i, \hat{y}_i)\}_{i=1}^N$, where $\mathbf{x} \in \mathbb{R}^p$ denotes a $p$-dimensional input variable, $y$ is the desired response, and $\hat{y}$ is the predicted output generated by a model $h_\theta$. Let $K$, $L^1$ and $L^2$ denote, respectively, the Gram matrices for the variable $\mathbf{x}$, $y$, and $\hat{y}$ (i.e., $K_{ij} = \kappa(\mathbf{x}_i, \mathbf{x}_j)$, $L_{ij}^1 = \kappa(y_i, y_j)$ and $L_{ij}^2 = \kappa(\hat{y}_i, \hat{y}_j)$). Further, let $L^{21}$ denote the Gram matrix between $\hat{y}$ and $y$ (i.e., $L_{ij}^{21} = \kappa(\hat{y}_i, y_j)$). The prediction term $D_{\mathrm{CS}}(p(y|\mathbf{x}); q_\theta(\hat{y}|\mathbf{x}))$ is given by:

$$\widehat{D}_{\mathrm{CS}}(p(y|\mathbf{x}); q_\theta(\hat{y}|\mathbf{x})) = \log \left( \sum_{j=1}^N \left( \frac{\sum_{i=1}^N K_{ji} L_{ji}^1}{(\sum_{i=1}^N K_{ji})^2} \right) \right)$$
$$+ \log \left( \sum_{j=1}^N \left( \frac{\sum_{i=1}^N K_{ji} L_{ji}^2}{(\sum_{i=1}^N K_{ji})^2} \right) \right) - 2 \log \left( \sum_{j=1}^N \left( \frac{\sum_{i=1}^N K_{ji} L_{ji}^{21}}{(\sum_{i=1}^N K_{ji})^2} \right) \right). \tag{98}$$

We focus our analysis on the last term, as it directly evaluates the difference between $y$ and $\hat{y}$, which should governs the quality of prediction.

By the second order Taylor expansion, we have:

$$L_{ij}^{21} = \kappa(\hat{y}_i, y_j) = \exp \left( -\frac{(\hat{y}_i - y_j)^2}{2\sigma^2} \right) \approx 1 - \frac{1}{2\sigma^2}(\hat{y}_i - y_j)^2, \tag{99}$$

For a fixed $j$, we have:

$$
\frac{\sum_{i=1}^{N} K_{ji} L_{ji}^{21}}{(\sum_{i=1}^{N} K_{ji})^2} = \frac{K_{j1}}{(\sum_{i=1}^{N} K_{ji})^2} \cdot L_{j1}^{21} + \frac{K_{j2}}{(\sum_{i=1}^{N} K_{ji})^2} \cdot L_{j2}^{21} + \cdots + \frac{K_{jN}}{(\sum_{i=1}^{N} K_{ji})^2} \cdot L_{jN}^{21}
$$

$$
\approx \frac{1}{\sum_{i=1}^{N} K_{ji}} - \frac{1}{2\sigma^2} \left[ \frac{K_{j1}}{(\sum_{i=1}^{N} K_{ji})^2} \cdot (\hat{y}_j - y_1)^2 + \frac{K_{j2}}{(\sum_{i=1}^{N} K_{ji})^2} \cdot (\hat{y}_j - y_2)^2 \right.
$$

$$
\left. + \cdots + \frac{K_{jj}}{(\sum_{i=1}^{N} K_{ji})^2} \cdot (\hat{y}_j - y_j)^2 + \cdots + \frac{K_{jN}}{(\sum_{i=1}^{N} K_{ji})^2} \cdot (\hat{y}_j - y_N)^2 \right]. \tag{100}
$$

That is, compared with the naïve mean squared error (MSE) loss that only minimizes $(\hat{y}_j - y_j)^2$, our prediction term additionally considers the squared distance between $\hat{y}_j$ and $y_i (i \neq j)$. The weight on $(\hat{y}_j - y_i)^2$ is determined by $K_{ji}$, i.e., the kernel distance between $\mathbf{x}_j$ and $\mathbf{x}_i$: if $\|\mathbf{x}_j - \mathbf{x}_i\|_2^2$ is small, there is a heavy weight on $(\hat{y}_j - y_i)^2$; otherwise, the weight on $(\hat{y}_j - y_i)^2$ is very light or negligible. Hence, our prediction term encourages that if two points $\mathbf{x}_i$ and $\mathbf{x}_j$ are sufficiently close, their predictions ($\hat{y}_i$ and $\hat{y}_j$) will be close as well.

To empirically support this, we trained a neural network on the California Housing data with respectively the MSE loss and our conditional CS divergence loss (i.e., Eq. (98)). In the test set, we measure all the pairwise distances between the $i$-th sample and the $j$-th sample (i.e., $\|\mathbf{x}_i - \mathbf{x}_j\|_2$) and their predictions (i.e., $|\hat{y}_i - \hat{y}_j|$). From Fig. 7, we observe that our conditional CS divergence loss encourages much higher correlation between $\|\mathbf{x}_i - \mathbf{x}_j\|_2$ and $|\hat{y}_i - \hat{y}_j|$ than MSE, which corroborates our analysis. That is, close points have closer predictions.

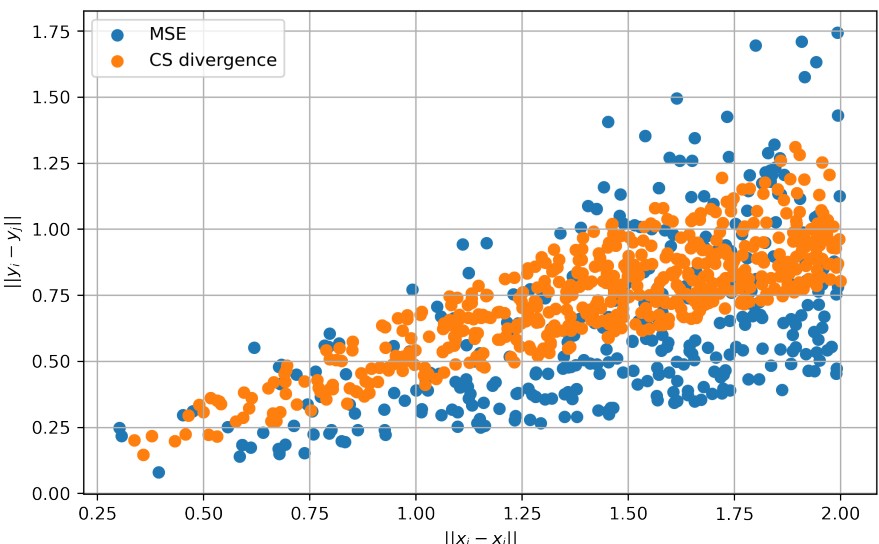

Figure 7: $\|\mathbf{x}_i - \mathbf{x}_j\|_2$ with respect to $|\hat{y}_i - \hat{y}_j|$.

On the other hand, when $\sigma \to 0$, we have $K_{jj} = 1$ and $K_{ji} \to 0 (i \neq j)$. In this case, our prediction term reduces to the MSE.

If we measure the divergence between $p(y|\mathbf{x})$ and $q_\theta(\hat{y}|\mathbf{x})$ with conditional MMD by (Ren et al., 2016). The empirical estimator is given by

$$
\widehat{D}_{\text{MMD}}(p(y|\mathbf{x}); q_\theta(\hat{y}|\mathbf{x})) = \text{tr}(K\tilde{K}^{-1}L^1\tilde{K}^{-1}) + \text{tr}(K\tilde{K}^{-1}L^2\tilde{K}^{-1}) - 2\,\text{tr}(K\tilde{K}^{-1}L^{21}\tilde{K}^{-1}), \tag{101}
$$

in which $\tilde{K} = K + \lambda I$.

By comparing conditional MMD in Eq. (101) with our conditional CS in Eq. (98), we observe that conditional CS avoids introducing an additional hyperparametr $\lambda$ and the necessity of matrix inverses, which is computationally efficient and more stable.

To justify our argument, we additionally compared conditional MMD with conditional CS as a loss function to train a neural network also on California Housing data. The result is in Fig. 8, which clearly corroborates our analysis.

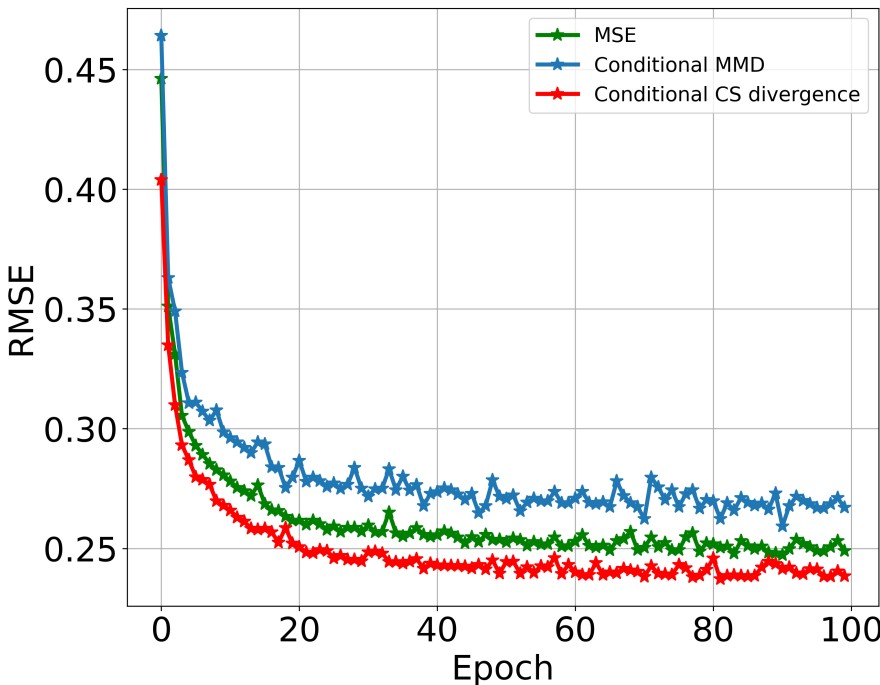

Figure 8: Comparison of learning curves of conditional MMD, MSE, and conditional CS divergences on the California Housing test data. The learning curve of conditional CS is much smoother than that that of conditional MMD (almost no jitter).

Finally, one can also non-parametrically measure the closeness between $p(y|\mathbf{x})$ and $q_\theta(\hat{y}|\mathbf{x})$ with $\widehat{D}_{\mathrm{KL}}(p(y|\mathbf{x}); q_\theta(\hat{y}|\mathbf{x}))$. By the same KDE technique, we obtain:

$$
\widehat{D}_{\mathrm{KL}}(p(y|\mathbf{x}); q_\theta(\hat{y}|\mathbf{x})) = \widehat{D}_{\mathrm{KL}}(p(\mathbf{x}, y); q_\theta(\mathbf{x}, \hat{y})) = \frac{1}{N} \sum_{i=1}^{N} \log\left( \frac{p(\mathbf{x}_i, y_i)}{q_\theta(\mathbf{x}_i, \hat{y}_i)} \right)
$$

$$
= \frac{1}{N} \sum_{i=1}^{N} \log\left( \frac{\sum_{j=1}^{N} \exp\left( \| \begin{bmatrix} \mathbf{x}_i \\ y_i \end{bmatrix} - \begin{bmatrix} \mathbf{x}_j \\ y_j \end{bmatrix} \|_2^2 / 2\sigma^2 \right)}{\sum_{j=1}^{N} \exp\left( \| \begin{bmatrix} \mathbf{x}_i \\ \hat{y}_i \end{bmatrix} - \begin{bmatrix} \mathbf{x}_j \\ y_j \end{bmatrix} \|_2^2 / 2\sigma^2 \right)} \right). \tag{102}
$$

Unfortunately, we found that the above formula is hard to converge or achieve a promising result. One possible reason is perhaps the support constraint on the KL divergence to obtain a finite value: in our case, there is no guarantee that $\mathrm{supp}(p(\mathbf{x}, y)) \subseteq \mathrm{supp}(q_\theta(\mathbf{x}, \hat{y}))$.

### B.4 ASYMPTOTIC PROPERTY OF OUR CS DIVERGENCE ESTIMATOR

For simplicity, we analyze the asymptotic property of our KDE estimator on the basic CS divergence between two distributions $p$ and $q$, which has the following form:

$$
D_{\mathrm{CS}}(p; q) = \log\left( \int p^2 d\mu \right) + \log\left( \int q^2 d\mu \right) - 2\log\left( \int pq\, d\mu \right) \tag{103}
$$

$$
= \log\left( \mathbb{E}_p(p) \right) + \log\left( \mathbb{E}_q(q) \right) - \log\left( \mathbb{E}_p(q) \right) - \log\left( \mathbb{E}_q(p) \right).
$$

Note that, the term $2\log\left( \int pq\, d\mu \right)$ can be expressed as:

$$
w_1 \log\left( \mathbb{E}_p(q) \right) + w_2 \log\left( \mathbb{E}_q(p) \right) \quad \forall w_1 + w_2 = 2, w_1 \geq 0, w_2 \geq 0. \tag{104}
$$

In the following, we consider two distributions with equal number of samples and hence set $w_1 = w_2 = 1$. Given observations $X := \{x_1, x_2, \cdots, x_N\}$ and $Y := \{y_1, y_2, \cdots, y_N\}$ drawn *i.i.d.* from $p$ and $q$, respectively, the empirical estimator of $D_{\text{CS}}(p; q)$ is given by:

$$\widehat{D}_{\text{CS}}(p(x); q(y)) = \log\left(\frac{1}{N}\sum_{i=1}^{N}\widehat{p}(x_i)\right) + \log\left(\frac{1}{N}\sum_{i=1}^{N}\widehat{q}(y_i)\right) - \log\left(\frac{1}{N}\sum_{i=1}^{N}\widehat{q}(x_i)\right) - \log\left(\frac{1}{N}\sum_{i=1}^{N}\widehat{p}(y_i)\right)$$

$$= \log\left(\frac{1}{N^2}\sum_{i=1}^{N}\sum_{j=1}^{N}\kappa_\sigma(x_j - x_i)\right) + \log\left(\frac{1}{N^2}\sum_{i=1}^{N}\sum_{j=1}^{N}\kappa_\sigma(y_j - y_i)\right)$$

$$- \log\left(\frac{1}{N^2}\sum_{i=1}^{N}\sum_{j=1}^{N}\kappa_\sigma(y_j - x_i)\right) - \log\left(\frac{1}{N^2}\sum_{i=1}^{N}\sum_{j=1}^{N}\kappa_\sigma(x_j - y_i)\right),$$

(105)

where the second equation is by KDE with a kernel function $\kappa$ of width $\sigma$.

Note that, our CS divergence estimator in Eq. (105) is different to that derived in (Jenssen et al., 2006).

In fact, quantities like $\int p^2 d\mu$ and $\int pq d\mu$ can be estimated in a couple of ways (Beirlant et al., 1997). Our estimator in Eq. (105) is also called the *resubstitution estimator*; whereas the authors in (Jenssen et al., 2006) use the *plug-in estimator* that simply inserting KDE of the density into the formula, i.e.,

$$\int p^2 d\mu \approx \int \widehat{p}^2(x)dx = \int \left(\frac{1}{N}\sum_{i=1}^{N}\kappa_\sigma(x_i - x)\right)^2 dx$$

$$= \frac{1}{N^2}\sum_{i=1}^{N}\sum_{j=1}^{N}\int \kappa_\sigma(x_i - x) \times \kappa_\sigma(x_j - x)dx.$$

(106)

Authors of (Jenssen et al., 2006) then assume a Gaussian kernel and rely on the property that the integral of the product of two Gaussians is exactly evaluated as the value of the Gaussian computed at the difference of the arguments and whose variance is the sum of the variances of the two original Gaussian functions. Hence,

$$\int p^2 d\mu \approx \frac{1}{N^2}\sum_{i=1}^{N}\sum_{j=1}^{N}\int \kappa_\sigma(x_i - x) \times \kappa_\sigma(x_j - x)dx = \frac{1}{N^2}\sum_{i=1}^{N}\sum_{j=1}^{N}\kappa_{\sqrt{2}\sigma}(x_i - x_j). \quad (107)$$

To our knowledge, other kernel functions, however, do not result in such convenient evaluation of the integral because the Gaussian maintains the functional form under convolution.

By contrast, we estimate $\int p^2 d\mu$ as:

$$\int p^2 d\mu = \mathbb{E}_p(p) = \frac{1}{N}\sum_{i=1}^{N}p(x_i) = \frac{1}{N}\sum_{i=1}^{N}\left(\frac{1}{N}\sum_{j=1}^{N}\kappa_\sigma(x_i - x_j)\right)dx = \frac{1}{N^2}\sum_{i=1}^{N}\sum_{j=1}^{N}\kappa_\sigma(x_i - x_j).$$

(108)

Although Eq. (108) only differs from Eq. (107) by replacing $\sqrt{2}\sigma$ with $\sigma$, our estimator offers two immediately advantages over that in (Jenssen et al., 2006): 1) our estimator is generalizable to all valid kernel functions; 2) the asymptotic property of our estimator can be guaranteed, whereas such analysis in (Jenssen et al., 2006) is missing.

Having explained the difference between *resubstitution estimator* and *plug-in estimator*, we analyze the asymptotic bias and variance of our estimator. As an example, we focus our analysis on the cross-term $\int pq d\mu = \mathbb{E}_p(q)$.

### B.4.1 BIAS OF $\int pqd\mu = \mathbb{E}_p(q)$

The estimator of $\mathbb{E}_p(q)$ is given by:

$$\mathbb{E}_p(q) \approx \frac{1}{N^2} \sum_{i=1}^{N} \sum_{j=1}^{N} \kappa_\sigma(y_j - x_i). \tag{109}$$

Let us denote $S = \mathbb{E}_p(q)$ and $\widehat{S} = \frac{1}{N^2} \sum_{i=1}^{N} \sum_{j=1}^{N} \kappa_\sigma(y_j - x_i)$, the bias of $\widehat{S}$ can be obtained by expectation:

$$
\begin{aligned}
\mathbb{E}(\widehat{S}) &= \mathbb{E}\left[ \frac{1}{N^2} \sum_{i=1}^{N} \sum_{j=1}^{N} \kappa_\sigma(y_j - x_i) \right] \\
&= \frac{1}{N^2} \sum_{i=1}^{N} \sum_{j=1}^{N} \mathbb{E}\left[ \kappa_\sigma(y_j - x_i) \right] = \mathbb{E}\left[ \kappa_\sigma(y_j - x_i) \right].
\end{aligned}
\tag{110}
$$

Further,

$$
\begin{aligned}
\mathbb{E}\left[ \kappa_\sigma(y_j - x_i) \right] &= \iint \kappa_\sigma(y_j - x_i) p(x_i) q(y_j) dx_i dy_j \\
&= \int p(x_i) \left[ \int \kappa_\sigma(y_j - x_i) q(y_j) dy_j \right] dx_i \\
&= \int p(x_i) \left[ \int \kappa_\sigma(s) q(x_i + \sigma s) d(\sigma s) \right] dx_i \quad \text{Let us denote} \quad s = \frac{y_j - x_i}{\sigma}. \\
&= \int p(x_i) \left[ \int \kappa(s) q(x_i + \sigma s) ds \right] dx_i \quad \text{Note:} \quad \kappa_\sigma(s)\sigma = \frac{1}{\sqrt{2\pi}} \exp(-s^2/2) = \kappa(s) \\
&= \int p(x) \left[ \int \kappa(s) [q(x) + \sigma s q'(x) + \frac{1}{2}\sigma^2 s^2 q''(x) + \mathcal{O}(\sigma^2)] ds \right] dx \quad \text{Taylor expansion} \\
&= \int p(x) \left[ q(x) \underbrace{\int \kappa(s) ds}_{=1} + \sigma q'(x) \underbrace{\int s\kappa(s) ds}_{=0} + \frac{1}{2}\sigma^2 q''(x) \int s^2 \kappa(s) ds + \mathcal{O}(\sigma^2) \right] dx \\
&= \int p(x) [q(x) + \frac{1}{2}\sigma^2 q''(x) \mu_\kappa] dx + \mathcal{O}(\sigma^2),
\end{aligned}
\tag{111}
$$

where $\mu_\kappa = \int s^2 \kappa(s) ds$ (for Gaussian kernel, $\mu_\kappa = 1$).

Namely, the bias of $\widehat{S}$ is:

$$
\begin{aligned}
\mathbf{bias}(\widehat{S}) &= \mathbb{E}(\widehat{S}) - \int pqd\mu \\
&= \frac{1}{2}\sigma^2 \mu_\kappa \int pq'' d\mu + \mathcal{O}(\sigma^2) \\
&= \frac{1}{2}\sigma^2 \mathbb{E}_p(q'') \mu_\kappa + \mathcal{O}(\sigma^2).
\end{aligned}
\tag{112}
$$

We see that the bias of $\widehat{S}$ increases proportionally to the square of the kernel size multiplied by the expected value of the second derivative of $q$ (under distribution $p$). This result also reveals an interesting factor: the bias is caused by the *curvature* (second derivative) of the density function, which coincides with KDE.

### B.4.2 VARIANCE OF $\int pq d\mu = \mathbb{E}_p(q)$

For the analysis of variance, we can obtain an upper bound straightforwardly:

$$
\begin{aligned}
\mathbf{var}(\widehat{S}) &= \mathbf{var}\left( \frac{1}{N^2} \sum_{i=1}^{N} \sum_{j=1}^{N} \kappa_\sigma(y_j - x_i) \right) \\
&= \mathbf{var}\left( \frac{1}{N^2\sigma} \sum_{i=1}^{N} \sum_{j=1}^{N} \kappa\left( \frac{y_j - x_i}{\sigma} \right) \right) \\
&= \frac{1}{N^4\sigma^2} \mathbf{var}\left( \sum_{i=1}^{N} \sum_{j=1}^{N} \kappa\left( \frac{y_j - x_i}{\sigma} \right) \right) \\
&= \frac{1}{N^2\sigma^2} \mathbf{var}\left( \kappa\left( \frac{y_j - x_i}{\sigma} \right) \right) \quad \text{independence assumption} \\
&\leq \frac{1}{N^2\sigma^2} \mathbb{E}\left( \kappa^2\left( \frac{y_j - x_i}{\sigma} \right) \right) \\
&= \frac{1}{N^2\sigma^2} \iint \kappa^2\left( \frac{y_j - x_i}{\sigma} \right) p(x_i)q(y_j) dx_i dy_j \\
&= \frac{1}{N^2\sigma^2} \int p(x) \left[ \int \kappa^2\left( \frac{y - x}{\sigma} \right) q(y)dy \right] dx \\
&= \frac{1}{N^2\sigma} \int p(x) \left[ \int \kappa^2(s)q(x + \sigma s)ds \right] dx \quad \text{Let us denote} \quad s = \frac{y - x}{\sigma}. \\
&= \frac{1}{N^2\sigma} \int p(x) \left[ \int \kappa^2(s)[q(x) + \sigma s q'(x) + \mathcal{O}(\sigma)]ds \right] dx \\
&= \frac{1}{N^2\sigma} \int p(x) \left[ q(x) \int \kappa^2(s)ds + \sigma q'(x) \underbrace{\int s\kappa^2(s)ds}_{=0} + \mathcal{O}(\sigma) \right] dx \quad \text{We assume symmetric kernel.} \\
&= \frac{1}{N^2\sigma} \int p(x)q(x)\sigma_\kappa^2 + \mathcal{O}(\frac{1}{N^2}),
\end{aligned}
$$
(113)

where $\sigma_\kappa^2 = \int \kappa(s)ds$ (for Gaussian kernel, $\sigma_\kappa^2 = \frac{1}{2\sqrt{\pi}}$).

So, from the above analysis, we conclude that the variance of $\widehat{S}$ will decrease inversely proportional to $N^2$, which is a comfortable result for estimation.

The asymptotic mean integrated square error (AMISE) of $\widehat{S}$ is therefore:

$$
\begin{aligned}
\mathbf{AMISE}(\widehat{S}) &= \mathbb{E}\left[ \int (\widehat{S} - S)^2 \right] \\
&= \frac{\sigma^4}{4} \mu_\kappa^2 \mathbb{E}_p^2(q'') + \frac{1}{N^2\sigma} \mathbb{E}_p(q)\sigma_\kappa^2,
\end{aligned}
$$
(114)

in which $\mu_\kappa = \int s^2 \kappa(s)ds$ and $\sigma_\kappa^2 = \int \kappa(s)ds$.

To summarize, AMISE will tend to zero when the kernel size $\sigma$ goes to zero and the number of samples goes to infinity with $N^2\sigma \to 0$, that is, $\widehat{S}$ is a consistent estimator of $S$.

Finally, one should note that, the $\log$ operator does not influence the convergence of $\widehat{S}$:

$$
N\sqrt{\sigma}(\widehat{S} - S) = \mathcal{O}_p(1)
$$
(115)

$$
N\sqrt{\sigma}(\log(\widehat{S}) - \log(S)) = N\sqrt{\sigma} \log\left( 1 + \frac{N\sqrt{\sigma}(\widehat{S} - S)}{N\sqrt{\sigma}S} \right) = \mathcal{O}_p(1)
$$
(116)

Additionally, this result can also be obtained by the delta method (Ferguson, 2017).

The bias and variance of other terms such as $\log \mathbb{E}_p(p)$ and $\log \mathbb{E}_q(q)$ in CS divergence can be quantified similarly.

The same result also applies for CS-QMI, i.e., $I_{CS}(\mathbf{x}; \mathbf{y})$. This is because we can construct a new concatenated variable $\mathbf{z} = (\mathbf{x}, \mathbf{y})^T$ in the joint space of $\mathbf{x}$ and $\mathbf{y}$, and let $p(\mathbf{z}) = p(\mathbf{x}, \mathbf{y})$ and $q(\mathbf{z}) = p(\mathbf{x})p(\mathbf{y})$, then:

$$I_{CS}(\mathbf{x}; \mathbf{y}) = D_{CS}(p(\mathbf{z}); q(\mathbf{z})). \tag{117}$$

### B.5 EXTENSION OF THEOREM 1 AND ITS IMPLICATION

Theorem 1 suggests that, under Gaussian assumptions, $D_{CS}(p; q) \leq D_{KL}(p; q)$. In this section, we show that such conclusion can be generalized to two arbitrary square-integral density functions. We also provide its implication.

#### B.5.1 CS DIVERGENCE IS USUALLY SMALLER THAN KL DIVERGENCE

**Proposition 5.** *For any density functions $p$ and $q$ that are square-integral, let $|K|$ denote the length of the integral's integration range $K$ with $|K| \gg 0$, we have:*

$$C_1 \left[ D_{CS}(p; q) - \log |K| + 2 \log C_2 \right] \leq D_{KL}(p; q), \tag{118}$$

*in which $C_1 = \int_K p(\mathbf{x})d\mathbf{x} \approx 1$ and $C_2 = \frac{\int_K p(\mathbf{x})d\mathbf{x}}{\left( \int_K p^2(\mathbf{x})d\mathbf{x} \int_K q^2(\mathbf{x})d\mathbf{x} \right)^{1/4}} \approx \frac{1}{\left( \int_K p^2(\mathbf{x})d\mathbf{x} \int_K q^2(\mathbf{x})d\mathbf{x} \right)^{1/4}}$.*

*Proof.* The following results hold for multivariate density functions. For straightforward illustration, we prove the results for the univariate case.

We first present Lemma 5, which is also called the Jensen weighted integral inequality.

**Lemma 5.** *(Dragomir et al., 2003) Assume a convex function $f : I \mapsto \mathbb{R}$ and $g, h : [x_1, x_2] \mapsto \mathbb{R}$ are measurable functions such that $g(x) \in I$ and $h(x) \geq 0 \quad , \forall x \in [x_1, x_2]$. Also suppose that $h$, $gh$, and $(f \circ g) \cdot h$ are all integrable functions on $[x_1, x_2]$ and $\int_{x_1}^{x_2} h(x)dx > 0$, then*

$$f \left( \frac{\int_{x_1}^{x_2} g(x)h(x)dx}{\int_{x_1}^{x_2} h(x)dx} \right) \leq \frac{\int_{x_1}^{x_2} (f \circ g)(x)h(x)dx}{\int_{x_1}^{x_2} h(x)dx}. \tag{119}$$

Let us set $h(x) = b(x)$ and $g(x) = \frac{a(x)}{b(x)}$ and $f = x \log(x)$ which is a convex function, by applying Lemma 5, we obtain:

$$\left( \int_{x_1}^{x_2} a(x)dx \right) \log \left( \frac{\int_{x_1}^{x_2} a(x)dx}{\int_{x_1}^{x_2} b(x)dx} \right) \leq \int_{x_1}^{x_2} a(x) \log \frac{a(x)}{b(x)} dx. \tag{120}$$

The inequality above holds for any integration range, provided the Riemann integrals exist. Moreover, this inequality can be easily extended to general ranges, including possibly disconnected sets, using Lebesgue integrals. In fact, Eq. (120) can be understood as a continuous extension of the well-known log sum inequaity. For simplicity, we denote $\int_{x_1}^{x_2} a(x)dx = \int_K a(x)dx$, in which $|K| = x_2 - x_1 \gg 0$ refers to the length of the integral's interval.

Now, suppose we are given two distributions $p(x)$ and $q(x)$, let us construct the following two functions:

$$a(x) = \frac{p(x)}{C_2} = \frac{p(x)}{\int_K p(x)dx} \left( \int_K p^2(x)dx \int_K q^2(x)dx \right)^{1/4}; \tag{121}$$
$$b(x) = \sqrt{p(x)q(x)}.$$

Clearly,

$$\sqrt{\frac{p(x)}{q(x)}} = \frac{a(x)}{b(x)} C_2. \tag{122}$$

We have,

$$
\begin{aligned}
D_{\text{KL}}(p;q) &= \int_K p(x) \log \frac{p(x)}{q(x)} dx \\
&= 2 \int_K p(x) \log \sqrt{\frac{p(x)}{q(x)}} dx \\
&= 2 \int_K a(x) C_2 \log \left( \frac{a(x)}{b(x)} C_2 \right) dx \\
&= 2 C_2 \left[ \int_K a(x) \log \left( \frac{a(x)}{b(x)} \right) dx + \log C_2 \int_K a(x) dx \right] \\
&\geq 2 C_2 \left[ \left( \int_K a(x) dx \right) \log \left( \frac{\int_K a(x) dx}{\int_K b(x) dx} \right) + \log C_2 \int_K a(x) dx \right] \\
&= 2 C_2 \int_K a(x) dx \left[ \log \left( \frac{\int_K a(x) dx}{\int_K b(x) dx} \right) + \log C_2 \right],
\end{aligned}
\tag{123}
$$

in which the fifth line is due to Eq. (120).

Note that,

$$
\int_K a(x) dx = \int_K \frac{p(x)}{C_2} dx = \frac{1}{C_2} \int_K p(x) dx = \left( \int_K p^2(x) dx \int_K q^2(x) dx \right)^{1/4},
\tag{124}
$$

and, using the Cauchy-Schwarz inequality,

$$
\left( \int_K b(x) dx \right)^2 = \left( \int_K \sqrt{p(x)q(x)} \cdot 1 dx \right)^2 \leq \left( \int_K p(x)q(x) dx \right) \left( \int_K 1 dx \right) = \left( \int_K p(x)q(x) dx \right) |K|.
\tag{125}
$$

By plugging Eqs. (124) and (125) into Eq. (123), we have:

$$
\begin{aligned}
D_{\text{KL}}(p;q) &\geq 2 C_2 \int_K a(x) dx \left[ \log \left( \frac{\int_K a(x) dx}{\int_K b(x) dx} \right) + \log C_2 \right] \\
&= \int_K p(x) dx \left[ \log \left( \frac{\int_K a(x) dx}{\int_K b(x) dx} \right)^2 + 2 \log C_2 \right] \\
&= \int_K p(x) dx \left[ \log \left( \frac{\left( \int_K p^2(x) dx \int_K q^2(x) dx \right)^{1/2}}{\left( \int_K b(x) dx \right)^2} \right) + 2 \log C_2 \right] \\
&\geq \int_K p(x) dx \left[ \log \left( \frac{\left( \int_K p^2(x) dx \int_K q^2(x) dx \right)^{1/2}}{\left( \int_K p(x)q(x) \right) |K|} \right) + 2 \log C_2 \right] \\
&= \int_K p(x) dx \left[ D_{\text{CS}}(p;q) - \log |K| + 2 \log C_2 \right] \\
&= C_1 \left[ D_{\text{CS}}(p;q) - \log |K| + 2 \log C_2 \right].
\end{aligned}
\tag{126}
$$

in which $C_1 = \int_K p(x) dx \approx 1$.

$\square$

### B.5.2 Empirical Justification

We also provide an empirical justification, showing that in general cases, the following relationship largely holds:

$$
D_{\text{CS}} \lesssim D_{\text{KL}},
\tag{127}
$$

in which $p$ and $q$ need not be Gaussian, and the symbol $\lesssim$ denotes "less than or similar to".

We focus our justification on discrete $p$ and $q$ for simplicity. This is because, unlike the CS divergence, KL divergence does not have closed-form expression for mixture-of-Gaussians (MoG) (Kampa et al., 2011). Hence, it becomes hard to perform Monte Carlo simulation on the continuous regime.

For two discrete distributions $p$ and $q$ on the finite set $\mathcal{X} = \{x_1, x_2, \ldots, x_K\}$ (i.e., there are $K$ different discrete states), let us denote $p(x_i) = p(x = x_i)$, we have:

$$D_{\text{KL}}(p; q) = \sum_{i=1}^{K} p(x_i) \log \left( \frac{p(x_i)}{q(x_i)} \right)$$
$$\text{s.t.} \sum_{i=1}^{K} p(x_i) = \sum_{i=1}^{K} q(x_i) = 1 \tag{128}$$

$$D_{\text{CS}}(p; q) = -\log \left( \frac{\sum p(x_i) q(x_i)}{\sqrt{\sum p(x_i)^2} \sqrt{\sum q(x_i)^2}} \right) \tag{129}$$

To empirically justify our analysis, for each value of $K$, we randomly generate $1,000$ pairs of distributions $p$ and $q$. Fig. 9 demonstrates the values of $D_{\text{KL}}$ with respect to $D_{\text{CS}}$ when $K = 2$, $K = 3$ and $K = 10$, respectively.

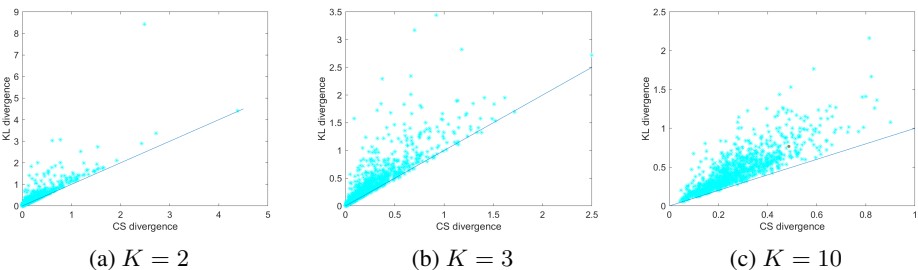

|  |  |  |
|---|---|---|
| (a) $K = 2$ | (b) $K = 3$ | (c) $K = 10$ |

Figure 9: Values of $D_{\text{KL}}$ with respect to $D_{\text{CS}}$ for $1,000$ pairs of randomly generated $p$ and $q$ when $K = 2$, $K = 3$ and $K = 10$, respectively. The diagonal indicates $D_{\text{KL}} = D_{\text{CS}}$.

### B.5.3 Tighter Generalization Error Bound in Unsupervised Domain Adaptation

In the problem of unsupervised domain adaptation, we aim to learn a classifier from samples in a source distribution $p_s$ that is generalizable to a related and different target distribution $p_t$. Suppose we learn a latent representation $\mathbf{t}$ such that $-\log \hat{p}(y|\mathbf{t})$ is bounded by a constant $M$[8].

From (Nguyen et al., 2022), the loss $l_{\text{test}}$ in the target domain can be upper bounded by the loss $l_{\text{train}}$ in the source domain as:

$$l_{\text{test}} \le l_{\text{train}} + \frac{M}{\sqrt{2}} \sqrt{D_{\text{KL}}(p_t(\mathbf{t}, y); p_s(\mathbf{t}, y))}. \tag{130}$$

Eq. (130) implies that the generalization gap from source to target domain is upper bounded by the mismatch on the joint distributions $p_t(\mathbf{t}, y)$ and $p_s(\mathbf{t}, y)$. The exact management of the KL divergence is usually hard. This drawback motivates a possibility to replace KL divergence with CS divergence, which, by Theorem 1, may enable tighter generalization error bound. We leave a systematic evaluation of this proposal as future work.

---

[8]In classification, we can enforce this condition easily by augmenting the output softmax of the classifier so that each class probability is always at least $\exp(-M)$ (Nguyen et al., 2022). For example, if we choose $M = 4$, then $\exp(-M) \approx 0.02$.

## C EXPERIMENTAL DETAILS AND ADDITIONAL RESULTS

### C.1 EFFECTS OF $I_{\text{CS}}(\mathbf{x}; \mathbf{t})$ ON GENERALIZATION

For the correlation experiment in Section 3.2.1, we generate a nonlinear regression data with 30-dimensional input, in which the input variable $\mathbf{x}$ is generated $i.i.d.$ from an isotropic multivariate Gaussian distribution, i.e., $\mathbf{x} \sim \mathcal{N}(0, I_{30})$. The corresponding output $y$ is generated as $y = \sin(\mathbf{w}^T \mathbf{x}) + \log_2(\mathbf{w}^T \mathbf{x})$, where $\mathbf{w} \sim \mathcal{N}(0, I_{30})$. We generate $5,000$ samples and use $4,000$ for training and remain the rest $1,000$ for test. We also select the real-world California housing dataset, and randomly split into $70\%$ training samples, $10\%$ validation samples, and $20\%$ testing samples.

For the first data, we use fully-connected networks and sweep over the following hyperparameters: (i) the depth (1 hidden layer or 2 hidden layers); (ii) the width (first hidden layer with number of neurons $\{256, 224, 192, 164, 128, 96, 64\}$; second hidden layer with number of neurons $\{64, 48, 32, 16\}$); (iii) batch size ($\{128, 96, 64\}$); (iv) learning rate $(0.1, 0.05)$. We train every model with SGD with a hard stop at 200 epochs. We only retain models that have a stable convergence.

For the second data, We also use fully-connected networks and sweep over the following hyperparameters: (i) the depth (2, 4, 6 or 8 hidden layers); (ii) the width (16, 32, 64, or 128) and keep the number of neurons the same for all the hidden layers; (iii) batch size (64, 128, 256, or 512); (iv) learning rate (0.001, 0.0005, 0.0001). We train 200 epochs for each model with Adam and only retain converged models. In total, we have nearly 100 models on two NVIDIA V100 GPUs.

For each resulting model, we compute three quantities: 1) the CS-QMI (i.e., $I_{\text{CS}}(\mathbf{x}; \mathbf{t})$ between input $\mathbf{x}$ and hidden layer representation $\mathbf{t}$); the conditional mutual information between input $\mathbf{x}$ and hidden layer representation $\mathbf{t}$ given response variable $y$ (i.e., $I_{\text{CS}}(\mathbf{x}; \mathbf{t}|y)$); 3) the generalization gap (i.e., the performance difference in training and test sets in terms of rooted mean squared error).

To evaluate $I_{\text{CS}}(\mathbf{x}; \mathbf{t}|y)$, we follow the chain rule in (Federici et al., 2020): $I_{\text{CS}}(\mathbf{x}; \mathbf{t}) = I_{\text{CS}}(\mathbf{x}; \mathbf{t}|y) + I_{\text{CS}}(y; \mathbf{t})$, in which $I_{\text{CS}}(\mathbf{x}; \mathbf{t}|y)$ is also called the superfluous information, and $I_{\text{CS}}(y; \mathbf{t})$ the predictive information.

We evaluate the dependence between $I_{\text{CS}}(\mathbf{x}; \mathbf{t})$ and the generalization gap, and the dependence between $I_{\text{CS}}(\mathbf{x}; \mathbf{t}|y)$ and the generalization gap. Two kinds of dependence measures are used: Kendall's $\tau$ and maximal information coefficient (MIC) (Reshef et al., 2011). For Kendall's $\tau$, values of $\tau$ close to 1 indicate strong agreement of two rankings for samples in variables $x$ and $y$, that that is, if $x_i > x_j$, then $y_i > y_j$. Kendall's $\tau$ matches our motivation well, since we would like to evaluate if a small value of $I_{\text{CS}}(\mathbf{x}; \mathbf{t})$ (or $I_{\text{CS}}(\mathbf{x}; \mathbf{t}|y)$) is likely to indicate a smaller generalization gap. Compared to Kendall's $\tau$, MIC is able to capture more complex and nonlinear dependence relationships.

According to Table 1 and Fig. 10, we can conclude that both $I_{\text{CS}}(\mathbf{x}; \mathbf{t})$ and $I_{\text{CS}}(\mathbf{x}; \mathbf{t}|y)$ have a positive correlation with empirical generalization error gap. This result is in line with (Kawaguchi et al., 2023; Galloway et al., 2023), although these two works focus on classification setup.

### C.2 EXPERIMENTAL SETUP IN REAL-WORLD REGRESSION DATASETS

We first provide more details on the used datasets in the main paper.

**California Housing**[9]**:** This dataset contains $20,640$ samples of 8 real number input variables like the longitude and latitude of the house. The output is the house price. A log-transformed house price was used as the target variable, and those 992 samples with a house price greater than $\$500,000$ were dropped. The data were normalized with zero mean and unit variance and randomly split into $70\%$ training samples, $10\%$ validation samples, and $20\%$ test samples.

**Appliance Energy**[10]**:** This dataset contains $12,630$ samples of appliance energy use in a low-energy building. Energy data was logged every 10 minutes for about $4.5$ months. In the dataset, each record has 14 features, such as air pressure, outside temperature and humidity, wind speed, visibility, dew point, energy use of light, and kitchen, laundry, and living room temperature and humidity. We select $80\%$ for training and $20\%$ for testing, normalize the data between 0 and 1 with MInMaxscaler.

---

[9]https://scikit-learn.org/stable/modules/generated/sklearn.datasets.fetch_california_housing.html

[10]https://www.kaggle.com/datasets/loveall/appliances-energy-prediction1

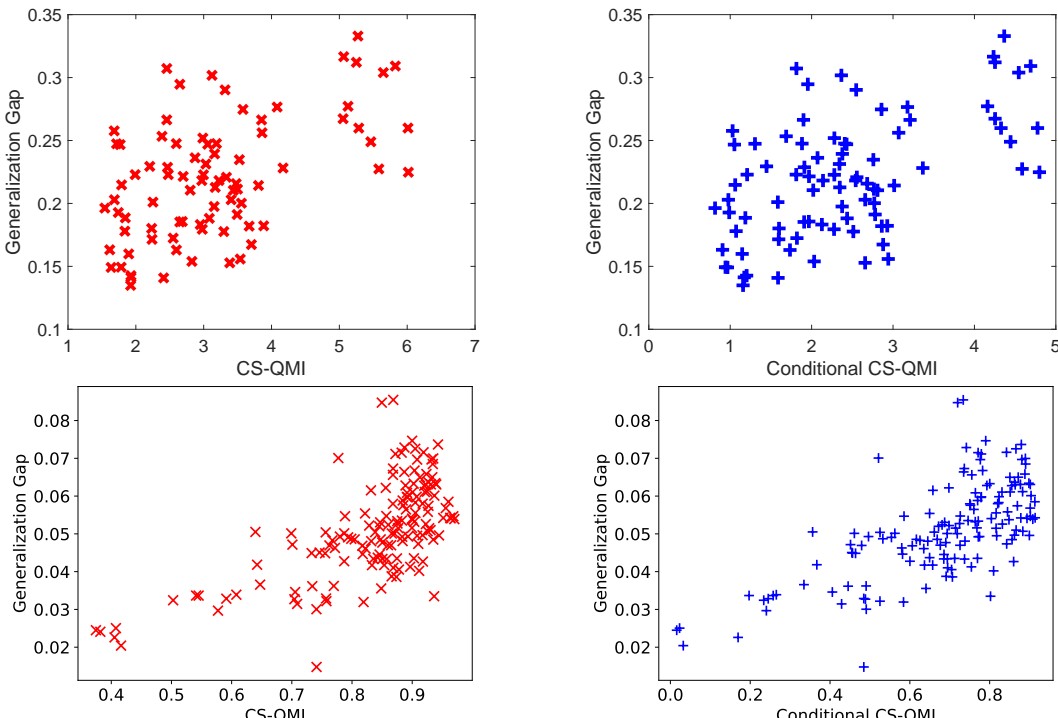

Figure 10: Scatter plot of $I_{CS}(\mathbf{x}; \mathbf{t})$ (left) and $I_{CS}(\mathbf{x}; \mathbf{t}|y)$ (right) with respect to the generalization gap in synthetic data (first row) and California housing (second row).

**Beijing PM2.5**[11]: This dataset collected PM 2.5 between Jan 1st, 2010 to Dec 31st, 2014 in Beijing. Each record in the dataset has 7 features, including dew point, temperature, pressure, combined wind direction, wind speed, hours of snow and hours of rain. The dataset contains $41,757$ samples. We selected $80\%$ for training and $20\%$ for test.

**Bike Sharing**[12]: This dataset includes the hourly and daily count of rental bikes between 2011 and 2012 in the Capital Bikeshare system, along with weather and seasonal information. In this paper, we utilize the hourly count data and each record in the dataset includes the following features: holiday, weekday, workingday, weathersite, temperature, feeling temperature, wind speed, humidity count of casual users, and count of registered users. Consisting of $17,379$ samples, the data was collected over two years, and can be partitioned by year and season. We use the first three seasons samples as training data and the forth season samples as test data.

**Rotation MNIST**[13]: The rotation MNIST contains synthetic images of handwritten digits together with the corresponding angles (in degrees) by which each image is rotated. The input image is of size $28 \times 28$, the output is in the range $[-\pi/4, \pi/4]$. The dataset consists of $10,000$ samples. We use $5,000$ for training and $5,000$ for test.

**UTKFace**[14]: For UTKFace, the input is grayscale face images with the size of $91 \times 91$, and the output is the corresponding age of each face. The original dataset includes samples of people with an age ranging from 0 to 116 years old. It also includes additional personal information such as gender and ethnicity. In our study, we use $7,715$ samples for training and $1,159$ samples for test, the used dataset includes people aged in a range of 0 to 80 with no additional information.

For a fair comparison, we consider the same architecture in (Kolchinsky et al., 2019b). Specifically, for California Housing and Bike Sharing datasets, the encoder $f_{\text{enc}}$ is a 3-layer fully-connected

---

[11] https://www.kaggle.com/datasets/djhavera/beijing-pm25-data-data-set
[12] https://archive.ics.uci.edu/ml/datasets/bike+sharing+dataset
[13] https://www.mathworks.com/help/deeplearning/ug/train-a-convolutional-neural-network-for-regression.html
[14] https://github.com/aicip/UTKFace

encoder with 128 ReLU hidden units, and the decoder $g_{\text{dec}}$ is a fully-connected layer with 128 ReLU units followed by an output layer with 1 linear unit. For the Appliance Energy and Beijing PM2.5 datasets, we utilize the past 4 days of data to predict the data of the next day. The decoder remains the same, while the encoder is a 3-layer LSTM with 32 hidden units followed by a fully-connected layer with 128 units. The IB regularization is added to the output of the encoder. The backbone architecture for both rotation MNIST and UTKFace is VGG-16, rather than the basic fully-connected network or LSTM with only a few layers.

All datasets are normalized between $[0, 1]$ with MinMaxscaler, and we set the kernel width $\sigma = 1$ for CS-IB and HSIC-bottleneck, which is actually a common choice for HSIC literature (Greenfeld & Shalit, 2020). In all experiments, we train networks with the Adam (Kingma & Ba, 2015) optimizer for 100 epochs and set the batch size to 128.

In our implementation, we use the normalized CS-QMI motivated by the formulation of centered kernel alignment (CKA) (Cortes et al., 2012) to guarantee the dependence value between $\mathbf{x}$ and $\mathbf{t}$ is bounded between $[0, 1]$:

$$\tilde{I}_{\text{CS}}(\mathbf{x}; \mathbf{t}) = \frac{I_{\text{CS}}(\mathbf{x}; \mathbf{t})}{\sqrt{I_{\text{CS}}(\mathbf{x}; \mathbf{x}) \cdot I_{\text{CS}}(\mathbf{t}; \mathbf{t})}}. \tag{131}$$

This strategy is also used in HSIC-bottleneck (Ma et al., 2020; Wang et al., 2021).

For each competing method, the hyperparameters are selected as follows. We first choose the default values of $\beta$ (the balance parameter to adjust the importance of the compression term) and the learning rate $lr$ mentioned in its original paper. Then, we select hyperparameters within a certain range of these default values that can achieve the best performance in terms of RMSE. For VIB, we search within the range of $\beta \in [10^{-3}, 10^{-5}]$. For NIB, square-NIB, and exp-NIB, we search within the range of $\beta \in [10^{-2}, 10^{-5}]$. For HSIC-bottleneck, we search $\beta \in [10^{-2}, 10^{-5}]$. For our CS-IB, we found that the best performance was always achieved with $\beta$ between $10^{-2}$ to $10^{-3}$. Based on this, we sweep the $\beta$ within this range and select the best one. The learning rate range for all methods was set as $[10^{-3}, 10^{-4}]$. We train all methods for 100 epochs on all datasets except for $PM2.5$, which requires around 200 epochs of training until converge. All the hyperparameter tuning experiments are conducted on the validation set. In Table 4, we record all hyperparameters for each deep IB approach that achieve the best RMSE. We train models with such hyperparameter setting to evaluate their adversarial robustness performances, as shown in Table 3.

Table 4: Hyperparameters for different IB approaches that achieve best RMSE on six real-world regression datasets.

| Dataset | param. | VIB | NIB | Square-NIB | exp-NIB | HSIC | CS-IB |
|---|---|---|---|---|---|---|---|
| Housing | learning rate | $1 \times 10^{-4}$ | $1 \times 10^{-3}$ | $1 \times 10^{-3}$ | $1 \times 10^{-3}$ | $1 \times 10^{-3}$ | $1 \times 10^{-4}$ |
| | epochs | 100 | 100 | 100 | 100 | 100 | 100 |
| | $\beta$ | $1 \times 10^{-5}$ | $3 \times 10^{-2}$ | $1 \times 10^{-2}$ | $1 \times 10^{-2}$ | $3 \times 10^{-3}/5 \times 10^{-3}$ | $1 \times 10^{-3}$ |
| Energy | learning rate | $1 \times 10^{-3}$ | $1 \times 10^{-4}$ | $1 \times 10^{-4}$ | $1 \times 10^{-4}$ | $1 \times 10^{-4}$ | $1 \times 10^{-4}$ |
| | epochs | 100 | 100 | 100 | 100 | 100 | 100 |
| | $\beta$ | $1 \times 10^{-4}$ | $3 \times 10^{-4}$ | $2 \times 10^{-4}$ | $2 \times 10^{-4}$ | $1 \times 10^{-3}/4 \times 10^{-2}$ | $5 \times 10^{-2}$ |
| PM2.5 | learning rate | $1 \times 10^{-3}$ | $1 \times 10^{-3}$ | $1 \times 10^{-3}$ | $1 \times 10^{-3}$ | $1 \times 10^{-4}$ | $1 \times 10^{-3}$ |
| | epochs | 200 | 200 | 200 | 200 | 200 | 200 |
| | $\beta$ | $5 \times 10^{-5}$ | $1 \times 10^{-5}$ | $1 \times 10^{-5}$ | $3 \times 10^{-5}$ | $1 \times 10^{-4}/8 \times 10^{-3}$ | $5 \times 10^{-3}$ |
| Bike | learning rate | $1 \times 10^{-4}$ | $1 \times 10^{-4}$ | $1 \times 10^{-4}$ | $1 \times 10^{-4}$ | $1 \times 10^{-4}$ | $1 \times 10^{-3}$ |
| | epochs | 100 | 100 | 100 | 100 | 100 | 100 |
| | $\beta$ | $3 \times 10^{-5}$ | $5 \times 10^{-2}$ | $6 \times 10^{-3}$ | $2 \times 10^{-3}$ | $1 \times 10^{-4}/4 \times 10^{-3}$ | $1 \times 10^{-2}$ |
| Rotation MNIST | learning rate | $1 \times 10^{-3}$ | $1 \times 10^{-3}$ | $1 \times 10^{-3}$ | $1 \times 10^{-3}$ | $1 \times 10^{-3}$ | $1 \times 10^{-3}$ |
| | epochs | 100 | 100 | 100 | 100 | 100 | 100 |
| | $\beta$ | $1 \times 10^{-5}$ | $1 \times 10^{-2}$ | $1 \times 10^{-2}$ | $1 \times 10^{-2}$ | $1 \times 10^{-3}/4 \times 10^{-2}$ | $1 \times 10^{-3}$ |
| UTKFace | learning rate | $1 \times 10^{-3}$ | $1 \times 10^{-3}$ | $1 \times 10^{-3}$ | $1 \times 10^{-3}$ | $1 \times 10^{-4}$ | $1 \times 10^{-3}$ |
| | epochs | 100 | 100 | 100 | 100 | 100 | 100 |
| | $\beta$ | $1 \times 10^{-5}$ | $1 \times 10^{-2}$ | $1 \times 10^{-2}$ | $1 \times 10^{-5}$ | $1 \times 10^{-3}/1 \times 10^{-5}$ | $1 \times 10^{-2}$ |

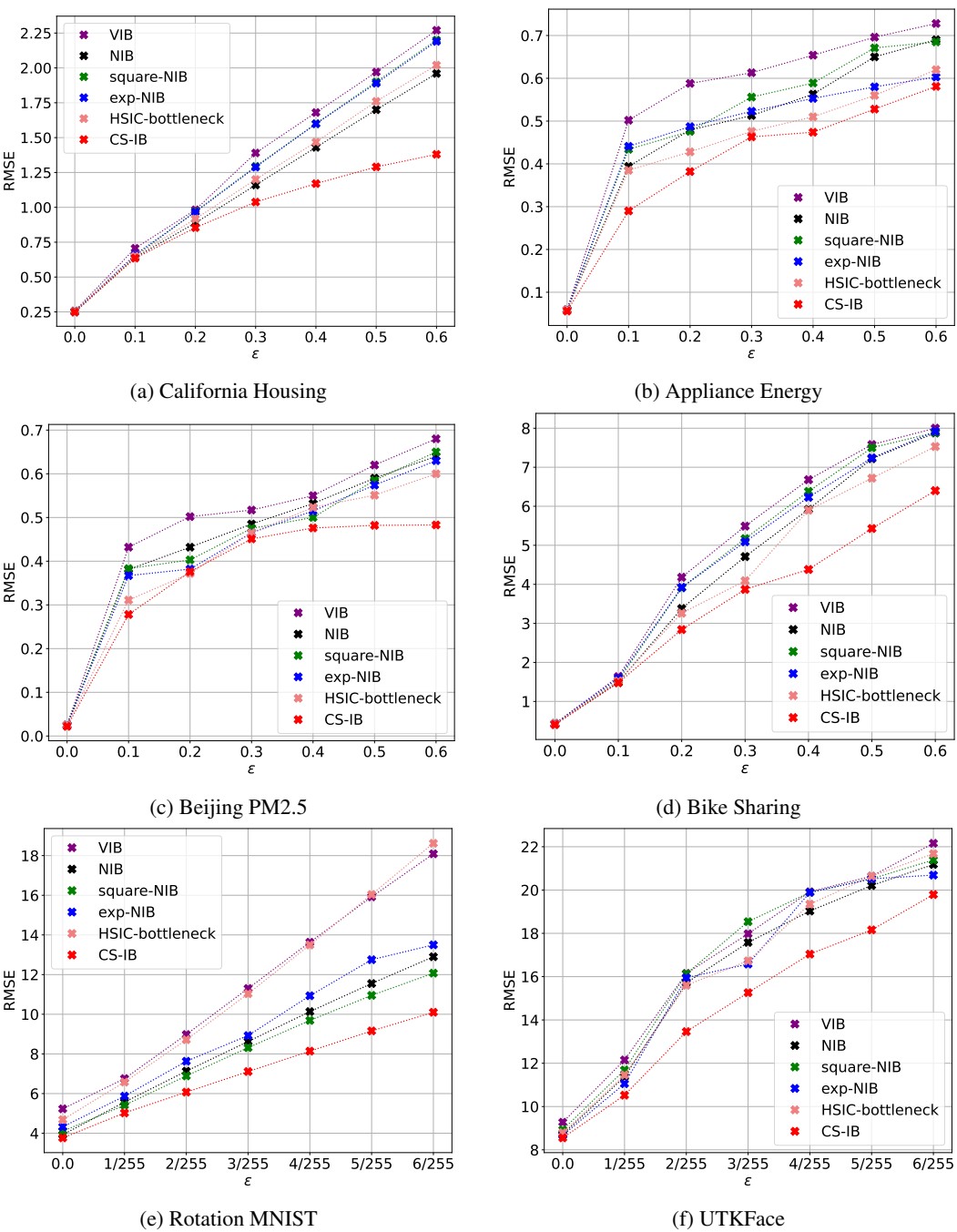

Figure 11: Test RMSE under FGSM attack with different $\epsilon$ on six regression datasets. Our CS-IB is consistently better that other competing approaches.

## C.3 ADVERSARIAL ROBUSTNESS

We further evaluate the adversarial robustness by comparing the behaviors of different IB approaches under FGSM attack with different perturbations $\epsilon = [0.1, 0.2, 0.3, 0.4, 0.5, 0.6]$. We use the same parameter configuration in Table 4, and only add adversarial attack on the test set. Our CS-IB outperforms other IB approaches with different perturbation strengths on all datasets as shown in Fig. 11.

### C.4 ADDITIONAL RESULTS OF SECTION 4.1 AND AN ABLATION STUDY

In Tables 8 and 9, we complete Table 2 and provide the standard deviation (std) for each method over 5 independent runs with different network initialization. In Table 7, we additionally provide the best performance achieved with $r \neq 0$ (i.e., $\beta > 0$) for different deep IB approaches.

We also conduct an ablation study on California Housing dataset to investigate the effectiveness of combining our prediction loss (the conditional CS divergence $D_{CS}(p(y|\mathbf{x}); q_\theta(\hat{y}|\mathbf{x}))$ in Eq. (18)) with $I(\mathbf{x}; \mathbf{t})$ regularization that is calculated by mutual information estimators in other IB approaches. For instance, "CS-div+VIB" refers to the combination of conditional CS divergence with $I(\mathbf{x}; \mathbf{t})$ calculated by variational lower bound. In Table 6, we observe that replacing MSE with our conditional CS divergence $D_{CS}(p(y|\mathbf{x}); q_\theta(\hat{y}|\mathbf{x}))$ improves the performance for all competing IB approaches. This result indicates that our conditional CS divergence is more helpful than MSE in extracting useful information from $\mathbf{x}$ to predict $y$.

### C.5 COMPARING CS DIVERGENCE WITH VARIATIONAL KL DIVERGENCE

We aim to verify the advantage of CS divergence over variational KL divergence when we want to approximate an unknown distribution $p(\mathbf{x})$ with another distribution $q(\mathbf{x})$.

Specifically, we assume that $p(\mathbf{x})$ as the true distribution is consists of two Gaussians with mean vectors $[4, 4]^T$ and $[-4, -4]^T$ and covariance matrix $\mathbf{I}_2$. Suppose we approximate $p(\mathbf{x})$ with a single Gaussian $q_\phi(\mathbf{x})$ by minimizing the KL divergence $D_{KL}(q_\phi(\mathbf{x}); p(\mathbf{x}))$ (the reverse KL divergence is also in different variational IB approaches) with respect to $q$'s variational parameters $\phi$. We initialize the variational parameters as:

$$\phi_\mu = [0, 0]^T \quad \phi_\Sigma = \mathbf{I}_2, \tag{132}$$

and optimize the following objective with gradient descent:

$$D_{KL}(q_\phi(\mathbf{x}); p(\mathbf{x})) = -H(q_\phi(\mathbf{x})) - \mathbb{E}_{q_\phi}(\log p(\mathbf{x})). \tag{133}$$

As can be seen in Fig. 12(b), the minimization forces $q_\phi(\mathbf{x})$ to be zero where $p(\mathbf{x})$ is zero and hence makes it concentrate on one of the modes. This phenomenon is called mode-seeking.

Note that, a single Gaussian is a common choice for variational inference with KL divergence, this is because KL divergence does not have closed-form expression for mixture-of-Gaussians (MoG). We would like to point out here that the CS divergence has closed-form expression for MoG (Kampa et al., 2011), which may further strength its utility. Although this property is not used in our paper.

Now, we consider using CS divergence to approximate $p(\mathbf{x})$ with $q(\mathbf{x})$. To do this, we sample $N_1 = 400$ samples from $p(\mathbf{x})$ and initialize $q(\mathbf{x})$ by sampling $N_2 = 200$ samples from a standard Gaussian. Then, we optimize samples in $q(\mathbf{x})$ by minimizing the CS divergence. Because CS divergence is estimated in a non-parametric way that does not make any distributional assumptions, it can fit two Gaussians perfectly as shown in Fig. 12(d).

## D LIMITATIONS AND FUTURE WORK

Given the promising results demonstrating the usefulness of CS-IB, it also has limitations.

First, we still would like to mention a limitation of CS divergence with respect to the KL divergence. That is, we did not find a dual representation for the CS divergence. The dual representation of KL divergence, such as the well-known Donsker-Varadhan representation (Donsker & Varadhan, 1983), enables the use of neural networks to estimate the divergence itself, which significantly improves estimation accuracy in high-dimensional space. A notable example is the mutual information neural estimator (MINE) (Belghazi et al., 2018).

Second, although kernel density estimator (KDE) offers elegant expressions for both $I(\mathbf{x}; \mathbf{t})$ and $D(p(y|\mathbf{x}); q_\theta(\hat{y}|\mathbf{x}))$, its performance depends heavily on a proper choice of kernel width $\sigma$. In this paper, we normalized our data and observed that $\sigma = 1$ is always a reliable heuristic. We do observe that our estimators provide consistent performance gain in a reasonable range of kernel size (e.g., $1 - 3$), as illustrated in Fig. 15.

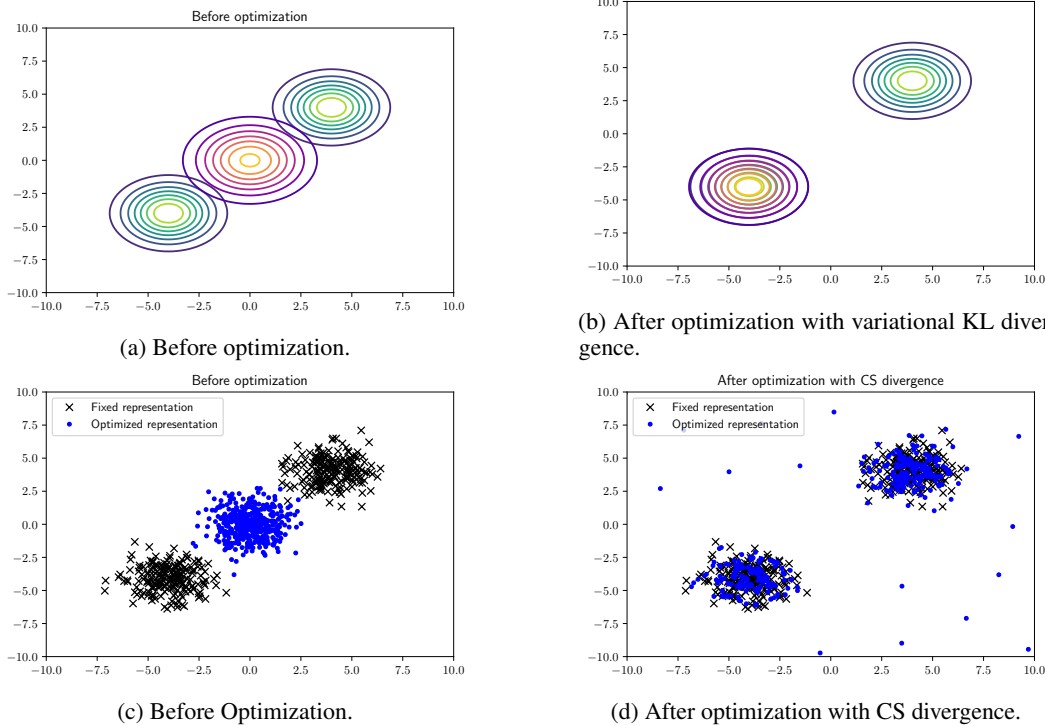

Figure 12: Comparison between CS-divergence and variational KL divergence on the simulated dataset.

There are multiple avenues for future work:

1) We would like to mention another two intriguing mathematical properties of CS divergence that both KL divergence and traditional Rényi's divergence do not have.

- CS divergence has closed-form expression for mixture-of-Gaussians (MoG) (Kampa et al., 2011).

- CS divergence is invariant to scaling. That is, given $\lambda_1, \lambda_2 > 0$, we always have:

$$D_{\text{CS}}(p; q) = D_{\text{CS}}(\lambda_1 p; \lambda_2 q). \tag{134}$$

Although both properties are not used in this paper. They should be beneficial to other statistical or machine learning tasks. For example, the first property has been leveraged in (Tran et al., 2022) to train variational autoencoders by replacing the single Gaussian prior distribution with MoG; whereas the second property has recently been used in (Jenssen, 2024) to project and visualize high-dimensional data.

2) We would like to extend CS-IB framework to other types of data, such as graphs. Some initial works have been done.

Here, we present our initial study to predict the age of patients based on their brain functional MRI (fMRI) data with a graph neural network (GNN), but leave a comprehensive and in-depth investigation as future work.

To this end, we rely on the brain information bottleneck (BrainIB) framework (Liu et al., 2023) framework as shown in Fig. 13. We use the original code from its authors[15] but replace the graph encoder with a graph transformer network (Shi et al., 2021). We add the information bottleneck regularization on the extracted graph representations, i.e., $g$ and $g_{\text{sub}}$; and train the whole network by

---

[15]https://github.com/SJYuCNEL/brain-and-Information-Bottleneck

the following objective:

$$\min D_{\text{CS}}(p(y|\mathbf{g}); p(\hat{y}|\mathbf{g})) + \beta I_{\text{CS}}(\mathbf{g}; \mathbf{g}_{\text{sub}}). \tag{135}$$

For other deep IB approaches, the objective is simply:

$$\min \mathbb{E}(\|y - \hat{y}\|_2^2) + \beta I(\mathbf{g}; \mathbf{g}_{\text{sub}}). \tag{136}$$

We train all models on the Autism Brain Imaging Data Exchange (ABIDE) dataset[16], which contains the fMRI data from $1,028$ patients (ages 7-64 years, median $14.7$). We split the data with 60% training, 20% validation, 20% test, and normalize the age between $0$ and $1$. For all IB approaches, we choose $\beta$ from four values: $0.0001$, $0.001$, $0.01$, and $0.1$, and select the one that achieves the best prediction accuracy. Fig. 14 demonstrates the predicted age with respect to true age for MSE loss only, NIB, HSIC-bottleneck, and our CS-IB. Table 5 records the quantitative evaluation results, in which the "prediction loss" indicates using $D_{\text{CS}}(p(y|\mathbf{g}); p(\hat{y}|\mathbf{g}))$ only (as a surrogate of MSE).

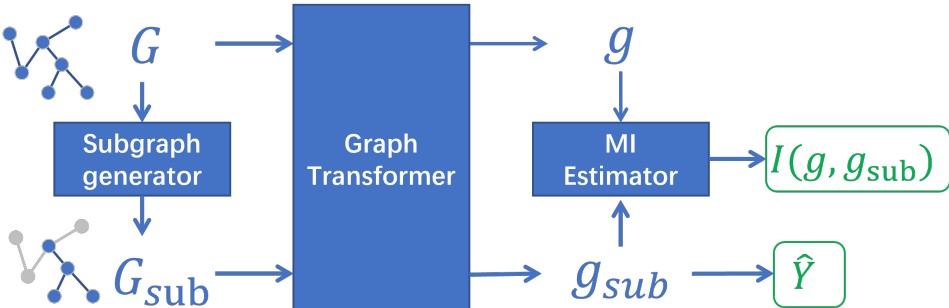

Figure 13: Subgraph information bottleneck framework for age prediction. $G$ is the original brain network, $G_{\text{sub}}$ is the identified brain sub-network that reduces irrelevant edge information for prediction. $g$ and $g_{\text{sub}}$ refers to graph vector representations corresponding to $G$ and $G_{\text{sub}}$, respectively, which are learned from a joint graph encoder with shared parameters. $g_{\text{sub}}$ is used for prediction.

Table 5: The mean absolute error (MAE) and pearson correlation coefficient (PCC) between predicted age and true age.

| Method | MAE | PCC |
|---|---|---|
| MSE | 0.079 | 0.564 |
| Prediction Loss | 0.067 | 0.571 |
| NIB | 0.072 | 0.573 |
| HSIC-bottleneck | 0.079 | 0.615 |
| CS-IB | **0.067** | **0.631** |

---

[16]http://fcon_1000.projects.nitrc.org/indi/abide/abide_I.html

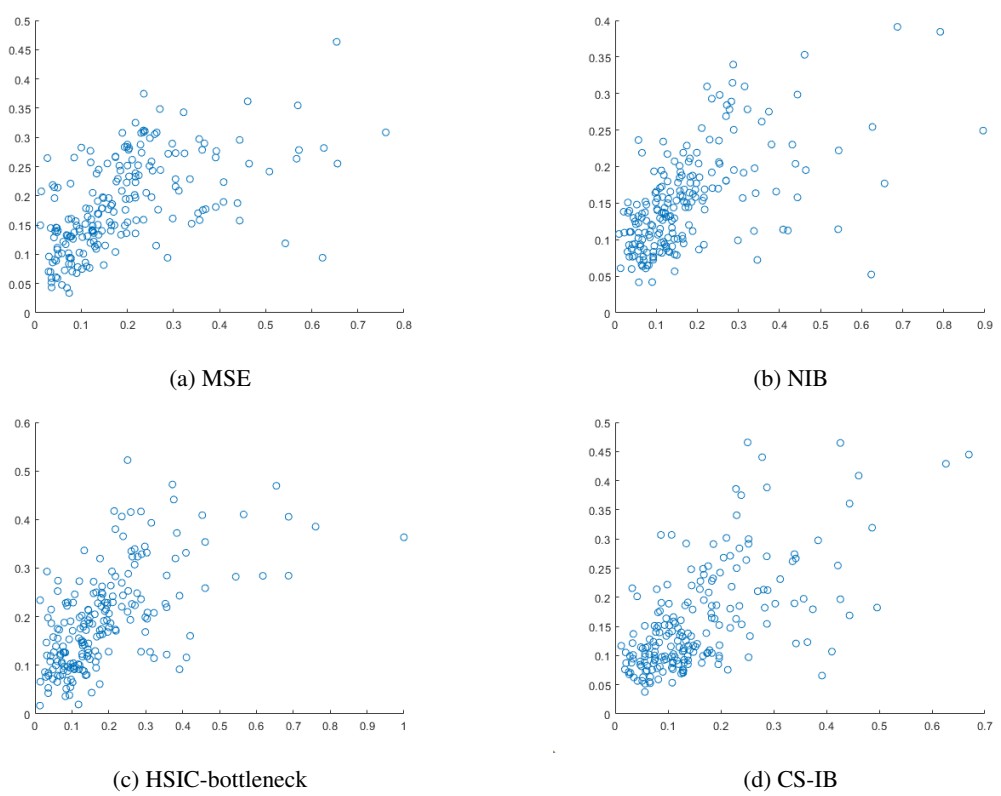

(a) MSE

(b) NIB

(c) HSIC-bottleneck

(d) CS-IB

Figure 14: Scatter plot of predicted age (y-axis) with respect to true age (x-age) for each patient in the test set.

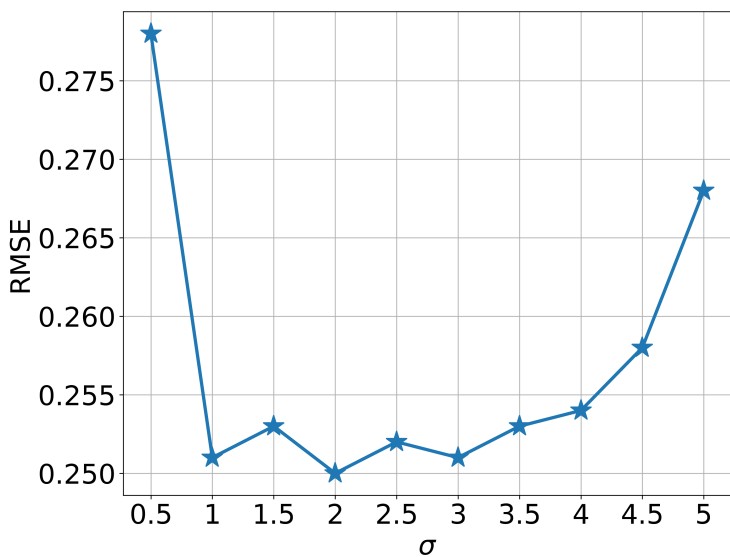

Figure 15: Performance of CS-IB with different kernel width $\sigma$ on California Housing dataset.

Table 6: Ablation study of conditional CS divergence on California Housing dataset. The best performance is highlighted. By replacing MSE with our conditional CS divergence in Eq. (18), all models have a performance gain.

| Method | Test RMSE |
|---|---|
| VIB | 0.257 |
| CS-div+VIB | 0.255 |
| NIB | 0.251 |
| CS-div+NIB | 0.248 |
| Square-NIB | 0.255 |
| CS-div+Square-NIB | 0.251 |
| Exp-NIB | 0.253 |
| CS-div+Exp-NIB | 0.250 |
| HSIC-bottleneck | 0.254 |
| CS-div+HSIC | 0.252 |
| CS-IB | **0.244** |

Table 7: The best performance achieved with $r \neq 0$ (i.e., $\beta > 0$) for different deep IB approaches.

| Method | Housing | Energy | PM2.5 | Bike | Rotation MNIST | UTKFace |
|---|---|---|---|---|---|---|
| VIB | 0.257 | 0.063 | 0.024 | 0.421 | 5.213 | 9.274 |
| NIB | 0.251 | 0.058 | 0.021 | 0.414 | 3.926 | 8.712 |
| Square-NIB | 0.255 | 0.060 | 0.023 | 0.420 | 4.102 | 8.673 |
| Exp-NIB | 0.253 | 0.058 | 0.024 | 0.423 | 4.325 | 8.927 |
| HSIC-bottleneck | 0.254 | 0.062 | 0.029 | 0.417 | 4.685 | 8.793 |
| CS-IB | **0.244** | **0.054** | **0.020** | **0.392** | **3.765** | **8.552** |

Table 8: RMSE for different deep IB approaches on California Housing, Appliance Energy, and Beijing PM2.5 with compression ratio $r = 0$ and $r = 0.5$. When $r = 0$, CS-IB uses prediction term $D_{CS}(p(y|\mathbf{x}); q_\theta(\hat{y}|\mathbf{x}))$ in Eq. (18), whereas others use MSE.

| Model | Housing | | Energy | | PM2.5 | |
|---|---|---|---|---|---|---|
| | 0 | 0.5 | 0 | 0.5 | 0 | 0.5 |
| VIB | 0.258±0.002 | 0.347±0.015 | 0.059±0.004 | 0.071±0.010 | 0.025±0.003 | 0.038±0.008 |
| NIB | 0.258±0.002 | 0.267±0.010 | 0.059±0.004 | 0.060±0.008 | 0.025±0.003 | 0.034±0.005 |
| Square-NIB | 0.258±0.002 | 0.293±0.008 | 0.059±0.004 | 0.063±0.005 | 0.025±0.003 | 0.028±0.004 |
| Exp-NIB | 0.258±0.002 | 0.287±0.010 | 0.059±0.004 | 0.061±0.006 | 0.025±0.003 | 0.030±0.006 |
| HSIC-bottlenck | 0.258±0.002 | 0.371±0.006 | 0.059±0.004 | 0.065±0.005 | 0.025±0.003 | 0.031±0.008 |
| CS-IB | **0.251**±0.001 | **0.245**±0.010 | **0.056**±0.002 | **0.058**±0.005 | **0.022**±0.002 | **0.027**±0.005 |

Table 9: RMSE for different deep IB approaches on Bike Sharing, Rotation MNIST, and UTK-Face with compression ratio $r = 0$ and $r = 0.5$. When $r = 0$, CS-IB uses prediction term $D_{CS}(p(y|\mathbf{x}); q_\theta(\hat{y}|\mathbf{x}))$ in Eq. (18), whereas others use MSE.

| Model | Bike | | Rotation MNIST | | UTKFace | |
|---|---|---|---|---|---|---|
| | 0 | 0.5 | 0 | 0.5 | 0 | 0.5 |
| VIB | 0.428±0.005 | 0.523±0.010 | 4.351±0.025 | 5.358±0.030 | 8.870±0.050 | 9.258±0.080 |
| NIB | 0.428±0.005 | 0.435±0.008 | 4.351±0.025 | 4.102±0.020 | 8.870±0.050 | 8.756±0.060 |
| Square-NIB | 0.428±0.005 | 0.447±0.006 | 4.351±0.025 | 4.257±0.010 | 8.870±0.050 | 8.712±0.050 |
| Exp-NIB | 0.428±0.005 | 0.458±0.005 | 4.351±0.025 | 4.285±0.015 | 8.870±0.050 | 8.917±0.050 |
| HSIC-bottlenck | 0.428±0.005 | 0.451±0.007 | 4.351±0.025 | 4.573±0.055 | 8.870±0.050 | 8.852±0.055 |
| CS-IB | **0.404**±0.004 | **0.412**±0.005 | **4.165**±0.015 | **3.930**±0.020 | **8.702**±0.035 | **8.655**±0.043 |

