# OpenReview forum: "Cauchy-Schwarz Divergence Information Bottleneck for Regression"
_ICLR.cc/2024/Conference — ICLR 2024 poster_

### Official Review · Reviewer_aubm · 2023-10-25

**Soundness:** 2 fair
**Presentation:** 3 good
**Contribution:** 3 good
**Rating:** 6
**Confidence:** 4

**Summary:**

The paper discusses an information bottleneck (IB) approach for neural network-based regression, using Cauchy-Schwarz divergence in place of Kullback-Leibler divergences and estimating these using kernel methods. The authors connect their approach to several methods from the literature (MMD, HSIC, etc.) and illustrate in their results that their approach is able to outperform IB approaches based on other approximations of (KL-based) mutual information.

**Strengths:**

The paper belongs to an interesting branch of literature, namely deep learning with information-theoretic cost functions. While the use of kernel methods in this regard is not new, the usage of Cauchy-Schwarz divergence is novel as far as I know. The paper is written quite accessibly, and important connections are made between the proposed approach and the existing related work. The results seem to indicate that the proposed methods can outperform existing approaches based on IB and its approximations.

**Weaknesses:**

There are a few aspects of the work that prevent me from recommending publication at this stage. I look forward to reading the authors replies to my concerns, upon which I may improve my score.
* In Remark 3, the authors clearly state that a kernel width of $\sigma=0$ reduces the CS divergence between the true posterior and the decoder distribution to classical MSE. With this in light, it is not clear in what respect CS divergence is fundamentally different from estimating $I(y;t)$ using a Gaussian variational distribution (limits novelty). Also, it is not clear why the CS-IB outperforms other methods also for $r=0$.
* In Remark 4, the authors claim that the use of KL divergence combined with a Gaussian assumption is "likely to induce inductive bias". The same can be said about using CS divergence with Gaussian kernels (which resonates with Remark 3, for which CS divergence reduces to MSE for $\sigma=0$). Specifically, at a very superficial glance it appears that the inductive biases of KL+Gauss and CS+Gauss are not too different. I would appreciate a paragraph explaining the fundamental differences (from the perspective of inductive bias).
* While I agree that CS-QMI is a rigorous definition, I do not agree that the proposed estimator measures mutual dependence in bits. As has been known for some time (see, e.g., Th. 1 in Amjad and Geiger or the work of Goldfeld et al.), for deterministic networks true MI between $x$ and $t$ is infinite. I doubt that $D_{CS}$ remains finite in such a setting (e.g., Renyi divergence is infinite if the joint and product of marginal distributions are singular w.r.t. each other, cf. Th. 24 in van Erven and Harremoes). From that perspective, I do not see how your estimators based on CS divergence have substantial mathematical advantages over existing estimators (although I acknowledge that the estimation may still be useful from a practical perspective; also, I understand that your networks are stochastic, because $t$ is obtained by adding Gaussian noise to a decoder output).
* One of my main concerns is the comparison of methods in Table 2 and Fig. 2. As far as I can see, $r$ is defined via the respective estimators (i.e., CS divergences for CS-IB, variational costs for VIB, etc.). This makes a comparison difficult, as the same $r$ indeed may lead to different compression ratios when compression is measured in a common currency (e.g., CS divergence for all methods). While this is problematic only for Table 2, in Fig. 2 the same holds for $I(y,t)$. Specifically, CS-IB is the only method that does not use the MSE, for the estimation of $I(y;t)$ which explains the difference at $r=0$. Essentially, the red line may be just above all others because $I(y;t)$ is measured differently (namely, using CS divergence). Also here a common currency would be better, such as MSE or RMSE. (Further, using the RMSE would allow for a better comparison with Table 2.)

_EDIT:_ Improved my score after rebuttal phase.

**Questions:**

See above.

---

> ### Author Response · Authors · 2023-11-22
> **Reply to your questions 1 and 2**
>
> First, thanks for careful reading our submission. Please let us reply to your questions one by one.
>
> ## [Q1 The fundamental difference between (conditional) CS divergence and $I(y;t)$ with Gaussian variational assumption]
>
> In fact, “CS+Gaussian” and “KL+Gaussian” are fundamentally different. Please refer to our reply to your [Q2].
>
> We now reply to your second question "why the CS-IB outperforms others also for $r=0$". In fact, when the compression ratio $r=0$, all competing methods do not have the compression term, our CS-IB optimizes $D_{CS}(p(y|x);q_\theta(\hat{y}|x))$ which reduces to MSE when kernel size $\sigma=0$; whereas other methods optimize MSE directly. For our method, we never select kernel size $\sigma=0$. This is because by our analysis in **Appendix B.4** in the revised submission, $\sigma=0$ significantly increases the estimation variance (which make the results highly unstable).
>
> There are two reasons why we are better: 1) we do not make parametric assumption on $q_\theta(\hat{y}|x)$. According to our analysis in **Appendix B.2**, a parametric assumption like Gaussian might not be the optimal choice, and the estimation bias is somewhat hard to control. In fact, our performance is better further justifying that there is a large bias introduced by the Gaussian assumption. Note that, for variational KL divergence estimator, the bias is exactly $D_{KL}(p(y|x);q_\theta(\hat{y}|x))$ or $D_{KL}(p(y|t);q_\theta(\hat{y}|t))$ (please refer to Eq.(96) in **Appendix B.2**).
>
> Another reason why $D_{KL}(p(y|x);q_\theta(\hat{y}|x))$ performs better than MSE when $\sigma\neq0$ is because, CS divergence encourages that if two points $x_i$ and $x_j$ are sufficiently close, their predictions ($\hat{y}_i$ and $\hat{y}_j$) will be close as well, which essentially enhances the numerical stability of the trained model. When $\sigma=0$, there is no such regularization.
>
> ## [Q2 The inductive bias between KL+Gaussian and CS+Gaussian]
> In fact, “KL+Gaussian” and “CS+Gaussian” are fundamentally different: the former is a parametric estimator that assumes the underlying distribution is Gaussian; whereas the latter is a non-parametric estimator that does not make any assumption on underlying distribution. A Gaussian kernel in kernel density estimator (KDE) does not mean we assume the underlying data distribution is Gaussian.
>
> When the underlying distribution is not Gaussian, “CS+Gaussian” still works, whereas “KL+Gaussian” suffers from poor performance. We added a toy example in **Appendix C.5** to illustrate this point.
>
> During the rebuttal, we also systematically analyze the bias and variance of our non-parametric estimators and show that they are consistent with a rate of convergence of $N\sqrt{\sigma}$, in which $N$ is the number of samples, $\sigma$ is the kernel width. Please refer to our discussion in **Appendix B.4**.
>
> Moreover, we also analyze the bias of previous variational KL divergence-based methods. Please refer to our discussion in **Appendix B.2**. Our argument is that previous literature either has obvious much more bias than ours or similar to us.
>
> To summarize, for our estimator, the asymptotic property is guaranteed. But the use of Gaussian kernel in KDE may introduce some inductive bias, since other kernels may perform better (e.g., smaller variance) or more robust, such as Epanechnikov kernel. But for “KL+Gaussian”, the Gaussian assumption alone for the underlying distribution makes the estimation is highly likely to be biased.

---

> ### Author Response · Authors · 2023-11-22
> **Reply to your questions 3 and 4**
>
> ## [Q3 substantial mathematical advantages by CS divergence]
> Thanks for the good question. First, we agree with you that in a deterministic network, the true CS-QMI between $x$ and $t$ may also become infinity. To check why, let us consider the CS-QMI between two Gaussian signals $x$ and $t$ with correlation coefficient $\rho$, in which $I_{CS}(x;t)$ is given by:
> $I_{CS}(x;t) = \frac{1}{2}\log\left( \frac{4-\rho^2}{4\sqrt{1-\rho^2}} \right)$.
> Suppose $x$ passes through an identity mapping, that is $t=x$, and $\rho=1$, we have $I_{CS}(x;t)\rightarrow\infty$.
>
> However, we would like to make a clarification that the CS divergence is not a special case of the well-known Renyi’s divergence developed by Renyi with the following expression:
> $D_\alpha$$(p;q) = \frac{1}{\alpha-1}\log\int p^\alpha q^{1-\alpha}du=\frac{1}{\alpha-1}\log\int p\left(\frac{q}{p}\right)^{1-\alpha}du=\frac{1}{\alpha-1}\log\mathbb{E}_p\left(\frac{q}{p}\right)^{1-\alpha}$
>
> When $\alpha=2$, we obtain a new divergence $-\log(1-\chi^2(p;q))$ that is proportional to the $\chi^2$ divergence, rather than our CS divergence mentioned in the paper.
>
> To our knowledge, there are two big ideas to define divergence: the integral probability metrics (IRM) aim to measure $p-q$, whereas the Renyi’s divergences with order $\alpha$ aim to model the ratio of $\frac{p}{q}$. The general idea of CS divergence is totally different: it neither horizontally computes $p-q$ nor vertically evaluates $\frac{p}{q}$.  Rather, it quantifies the “distance” of two distributions by quantify the tightness (or gap) of an inequality associated with integral of densities (i.e., $\int pq$). That is, CS divergence does not belong to either IRM or the traditional Renyi’s divergence family. Please refer to our discussion on **Appendix B.1** for more details.
>
> This uniqueness makes CS divergence has many desirable properties compared to KL. One of them is that it relaxes the constraint on the support. Note that, $D_{KL}(p;q)$ has finite values if and only if support of $p$ is a subset of support of $q$. This condition is very strong, which also explains why we use KL divergence combined with non-parametric estimator, the model is hard to train (please also refer to our discussion in **Appendix B.3**, especially the last paragraphs marked with blue).
>
> Finally, we would also like to mention two important and special properties of CS divergence over KL divergence (or Renyi divergence), although these two properties are not explicated used in our method, but it would be beneficial and helpful to the community. (1) the CS divergence has closed-form expression for mixture-of-Gaussians (MoG); (2) CS divergence is invariant to scaling. That is, given $\lambda_1,\lambda_2>0$, we always have $D_{\text{CS}} (p;q) = D_{\text{CS}} (\lambda_1 p;\lambda_2 q)$. Please also refer to our **Appendix D** for details.
>
> ## [Q4 Comparison of methods in Table 2 and Fig. 2]
> We would like to clarify that in Fig.2, the $y$-axis that quantities $I(y;t)$ is indeed measured with a common currency (to provide a fair comparison). Specifically, we follow (Kolchinsky et al., 2019b), and measure the true value of $I(y;t)$ by $1/2 \log⁡(\text{var}(y)/MSE)$. Hence, $I(y;t)$ in Fig.2 is not measured with respective estimators of different methods.
>
> Going back to the Table 2, yes, the compression ratio $r$ is indeed computed with respective estimators. However, we would like to clarify here that $r$ is a relative quantity, rather than an absolute quantity. Specifically, $r$ is defined as:
> $r = 1- \frac{a}{b}$, in which $a=I(x;t)$ when $\beta=\beta^*$ and $b=I(x;t)$ when $\beta=0$ (i.e., no compression).
>
> Hence, for each method, the physical meaning of $r$ is how much information has been compressed (when $\beta=\beta^*$) relatively to no compression (i.e., $\beta=0$).
>
> We use $r$ rather than $I(x;t)$ in both Table 2 and Fig 2. This is just because different IB approaches differ in the way to (define) or estimate $I(x;t)$. Most literatures use Shannon’s mutual information (MI), our CS-QMI estimates a generalized MI. By contrast, HSIC-bottleneck uses HSIC, which does not have physical meaning of mutual information in terms of bits or nats. Hence, if we use $I(x;t)$, different approaches have different units, which may result in an unfair comparison.
>
> Finally, we would like to emphasize that, for each method, we additionally report the best performance achieved when $r\neq0$ (i.e., $β\neq0$) in Table 7 in **Appendix C.4**. This result suggests that, under an optimal compression ratio (in which the $r$ or $\beta$ is different for different methods), our CS-IB is significantly and consistently better.

---

> > ### Comment · Reviewer_aubm · 2023-11-22
> > **Thanks!**
> >
> > Thank you very much for the clarifications. I appreciate your responses and will increase my score.

---

> > > ### Author Response · Authors · 2023-11-22
> > >
> > > Thanks for your reply. We are very glad that we clarified your concerns.

---

### Official Review · Reviewer_uyPi · 2023-10-29

**Soundness:** 3 good
**Presentation:** 3 good
**Contribution:** 3 good
**Rating:** 6
**Confidence:** 2

**Summary:**

I’m unfamiliar with entropy-based methods. So my reviews are purely based on reading the paper without comparison to the related literature.

This paper uses the CS divergence of IB for regression estimation. It can be regarded as empirical CS divergence minimization on the conditional distribution $p(y|x)$ with a regularization term on $I(x,t)$. This problem can be efficiently estimated via kernel density estimator. The effectiveness of this metric is demonstrated on four benchmark datasets and two image datasets.

**Strengths:**

- the CS divergence is smaller than KL divergence under Gaussian distribution as well as some relatively general setting (e.g., $p$ and $q$ are sufficiently small)
- The rationality of the regularization term is verified on generalization and adversarial robustness

**Weaknesses:**

Since I’m not an expert in this community, it appears difficult for me to evaluate the contribution. I only have few questions on this work:

- The CS divergence is based on the Cauchy-Schwartz inequality, a special case of Holder inequality. Does this metric can be extended in the general $L_p$ space?
- Theorem 1 is restricted to Gaussian distribution. Though this assumption can be relaxed if $p$ and $q$ are sufficiently small, it would be possible to extend to the sub-Gaussian, sub-exponential case?
- For the generalization bound, the authors obtain a tighter bound but $I_{CS}(x;t)$ and $I(x;t)$ are still in the same constant order?

**Questions:**

See the above

---

> ### Author Response · Authors · 2023-11-22
> **Reply to your questions**
>
> First, thanks for your positive comments. Please let us reply to your questions one by one.
>
> ## [Q1 A generalization of Cauchy-Schwarz divergence]
> Thanks for the good question. Yes, you are right, the general idea of CS divergence can be generalized to $L_p$ space by taking use of the Holder inequality:
> $|\int p(x)q(x)dx|\leq (\int|p(x)|^a d(x))^{1/a} (\int |q(x)|^b d(x))^{1/b}$, in which $a,b>1$ and $\frac{1}{a}+\frac{1}{b}=1$.
>
> Hence, a generalized way to quantify the closeness between $p$ and $q$ can be expressed as:
> $D = -\log\left( \frac{|\int p(x)q(x)dx|}{(\int|p(x)|^a d(x))^{1/a} (\int |q(x)|^b d(x))^{1/b}} \right)$.
>
> CS divergence is obtained by setting $a=b=2$. The generalized divergence can also be estimated by kernel density estimator.
>
> ## [Q2 generalization of Theorem 1]
> Thanks for the good suggestion. We generalized our Theorem 1 to any two square-integral densities $p$ and $q$. Please refer to **Appendix B.5** and Proposition 5 in the revised submission.
>
> ## [Q3 the same constant order of $I_{CS}(x;t)$ and $I(x;t)$]
> Yes, the same constant order. By definition, $I_{CS}(x;t) = D_{CS}(p(x;t);p(x)p(t))$ and $I(x;t) = D_{KL}(p(x;t);p(x)p(t))$, our justifications in **Appendix B.5** suggest that for any two square-integral densities $p$ and $q$, $D_{\text{CS}}(p;q) \lesssim D_{\text{KL}}(p;q)$. There is no other scaling factors in this inequality. By letting $p=p(x;t)$ and $q=p(x)p(t)$, we obtain $I_{CS}(x;t) \lesssim I(x;t)$.

---

> > ### Comment · Reviewer_uyPi · 2023-11-23
> >
> > thanks for your response. It addressed my concern. Since I'm not the expert in this community, after reading other reviewers' comments, I remain my score unchanged.

---

### Official Review · Reviewer_MhfL · 2023-10-31

**Soundness:** 3 good
**Presentation:** 2 fair
**Contribution:** 2 fair
**Rating:** 5
**Confidence:** 4

**Summary:**

- The authors considered using the Couchy--Schwarz (CS) divergence to parameterize the information-bottleneck (IB) with deep neural networks (DNNs), which is beyond the MSE (or MAE) loss based on Gaussian (or Laplace) variational distribution.
- Furthermore, they proposed using the non-parametric estimation by plugging in the output of DNNs through the kernel density estimator (KDE) for the following two terms in their objective: the prediction term (CS divergence) $D_{\mathrm{CS}}(p(y|x); q_{\theta}(\widehat{y}|x))$ and the compression term (CS-QMI) $I_{\mathrm{CS}}$, which allows us directly estimate the MI instead of its upper bound as in existing approaches.
- They also provided a discussion regarding the relationship between CS and MMD-based estimators.
- They offered theoretical and empirical analyses for generalization error and adversarial robustness based on the work of [Kawaguchi et al. in 2023] and [Wang et al. in 2021].
- They empirically confirmed the predictive and robustness performance of their CS-IB method.

**Strengths:**

- They introduced a new choice of loss function based on CS divergence for the IB method, which was often used with MSE based on Gaussian settings or MAE loss based on Laplace distribution.
- They pointed out the challenges in estimating MI and the adoption of indirect methods, such as estimating upper bounds. They proposed an IB framework based on direct estimation by performing non-parametric estimation using KDE.
- They pointed out that existing methods adopt indirect methods for MI estimation, such as estimating upper bounds, and they proposed an IB framework based on direct estimation by performing non-parametric estimation using KDE.
- They attempted empirical verification regarding the correlation between CS-QMI and generalization error, providing insights into the generalization analysis of the IB method based on CS divergence.
- They conducted performance evaluation experiments on a wide range of real-world data, including California Housing, MNIST, Beijing PM2.5, and UTKFace.

**Weaknesses:**

I would like to express my sincere respect for all the efforts the authors have invested in this paper. Unfortunately, however, I cannot strongly recommend this paper as an ICLR 2024 accepted paper for the following reasons: (1) a misalignment between the claims of contribution, the assumptions of theoretical analysis, and the content of theoretical analysis; (2) a lack of theoretical guarantees on the properties of the proposed estimations, and the unclear discussion of the pros and cons between direct estimation and upper bound estimation approaches, or the absence of sufficient comparative experiments to complement it; and (3) concerns regarding the reliability of the experimental results. The details are provided below. If there are any misunderstandings, I apologize, and I would appreciate it if you could explain them to me.

## A misalignment between the claims of contribution, the assumptions of theoretical analysis, and the content of theoretical analysis
- The authors claim that by leveraging the CS divergence, they can perform non-linear IB regression for any distribution $p(x, y)$ without heavy reliance on variational approximations and without making distribution assumptions. How accurate is this claim? The independence from the variational distribution seems to be achieved not so much due to the properties of the CS divergence but rather through non-parametric estimation using Kernel Density Estimation (KDE) on the outputs from neural networks, treating them as samples from the training data. Therefore, the motivation for introducing the CS divergence may lack a clear and solid rationale, leaving room for discussion regarding its validity in the context of this presentation.
- Furthermore, while the authors claim that their contribution lies in the ability to estimate without making distribution assumptions, the actual theoretical analysis deals with a limited setting where both the model distribution and variational distribution are assumed to be normal distributions. This limitation is acknowledged as a one of limitations in Appendix D. However, it can be considered a significant weakness that the proposed method's theoretical guarantees are limited, given the broad claim of removing the necessity of distribution assumptions. This disparity should ideally be addressed in the main part of the paper.
- For one of the contributions mentioned, I believe the explanation of the adversarial robustness aspect is rather too concise. Besides Lemma 1, there should be a proper proof for [Wang et al., 2021] Theorem 2's Corollary as well. In the current presentation, while one may vaguely understand the content being discussed, the explanations are too informal, making it difficult to accurately determine whether the validity is established. The theoretical part of the proof should ideally be self-contained within the main paper, rather than relying on related literature.
- The assumptions underlying generalization analysis and adversarial robustness have not been well-organized. It is recommended to consolidate these assumptions.

## A lack of theoretical guarantees on the properties of the proposed estimations, and the unclear discussion of the pros and cons between direct estimation and upper bound estimation approaches, or the absence of sufficient comparative experiments to complement it
- When constructing nonparametric estimators, a critical concern always revolves around whether (asymptotic) unbiasedness or (asymptotic) consistency is guaranteed. This theoretical assurance holds true for nonparametric estimations like Kernel Density Estimation (KDE) and k-Nearest Neighbors (k-NN) based on KL divergence [1,2], among others. In the context of the IB method under focus here, I presume that unbiasedness and consistency play a significant role in determining its performance. The importance of analyzing this aspect lies in the bias introduced by parameter estimation in variational approaches dependent on the variational distribution, as well as the bias in the estimators of the proposed method. Deciding which of these biases is smaller - the one induced by parameter estimation or the one introduced by the proposed method - is a crucial perspective in determining whether a direct estimation approach, like the proposed method, or a variational estimation approach is more useful. Unfortunately, this paper lacks comprehensive discussions on this aspect. The lack of convincing arguments on why a direct estimation approach like the proposed method is effective. This issue holds immense significance in the comparison between the proposed method and conventional methods, and it remains one of the unresolved problems in this paper, marked as a limitation. Relevant questions pertaining to this matter are summarized in the Question section.
- When comparing the bias introduced by the variational approach with that introduced by nonparametric estimations such as the proposed method, it would be beneficial to investigate a toy example with an increased sample size to determine which approach exhibits higher estimation accuracy. Unfortunately, this paper did not provide such experimental confirmations.
- The authors, I believe, are understood, but the theory in Section 3.2.1 assumes that the generalization error upper bound is derived in a manner that includes the objective function of the proposed method, which alone does not guarantee generalization performance, as is evident. As mentioned in the Strength section, the attempt to experimentally verify this aspect is a fascinating endeavor. However, the empirical correlations have only been validated on toy data. To increase the persuasiveness of the empirical evidence regarding the correlation with generalization performance, it would have been necessary to confirm the experimental results on benchmark data adopted in Section 4.

## Concerns regarding the reliability of the experimental results.
- Due to the absence of reports on measures of dispersion, such as standard deviations, for all the experimental results in this paper, it is difficult to determine whether the proposed method consistently achieves superior performance compared to other methods. This raises concerns about the validity of the claim made by the authors as one of their contributions, that CS-IB outperforms other methods. Since the proposed method is built on nonparametric estimation, it is conceivable that the variance of the estimates can become large when there are not enough samples, leading to increased variability in performance. In essence, to support the claim that the proposed method is superior, it would be challenging to agree without considering factors such as the randomness of model initialization, mini-batch datasets, and adversarial perturbations. It is advisable to repeat the experiments several times, report the average and standard error of the obtained prediction accuracy, and evaluate performance while taking variability into account.
- I find it puzzling that some of the reported numerical values in the experiments are identical in every aspect. For instance, in Table 2, the RMSE for the existing methods when $r=0$ is a complete match across all experimental settings. This further emphasizes the need for reporting both the average prediction accuracy and the standard error. While the average prediction accuracy may be the same, the standard error might differ between methods. If they do indeed match perfectly, it is advisable to check for any implementation bugs. At the very least, there are concerns regarding the credibility of the currently reported numerical values.
- I apologize if I missed any details earlier, but it does appear that there is a lack of explanation regarding hyperparameter settings, such as in Table 4. This omission can give the impression that there might be arbitrariness in the experimental results. If you have indeed eliminated arbitrariness, it would be beneficial to provide more detailed explanations about how you selected the reported hyperparameters. This would help enhance the transparency and credibility of your findings.

**Questions:**

In connection with the weaknesses mentioned above, I would like to pose several questions related to the concerns raised. I would appreciate your responses.

- It seems that this nonparametric approach could also be applicable to the conventional objective function based on KL divergence. Why was it not discussed, proposed, or included as a subject for comparative experiments?
- As mentioned in Remark 1, CS divergence is a special case of Rényi divergence. Under this premise, it could have been considered as a generalization of the conventional KL-based methods using Rényi divergence, especially from the perspective that it converges to KL divergence as α approaches. Why was the emphasis placed on this particular special case?
- The uniqueness of the KDE estimator in CS divergence has already been provided by [3]. I understand the need to construct an estimator consisting of three elements: predictions, input data, and label data, in the context of the IB method. However, it seems that there might not be such a fundamental difference between the basic nonparametric estimation and the proposed method. If these aspects are positioned as variations based on the ideas from related research, it would be beneficial to explicitly state this and discuss the differences, if any. What are your thoughts on this?
- Can you guarantee the asymptotic properties of the two estimation methods in the proposed approach? For instance, is asymptotic unbiasedness or consistency ensured?
- Why do the performance values for existing methods in Table 2 match perfectly?
- Why was it not considered to conduct repeated experiments and report the mean and standard deviation?
- How were the hyperparameters selected?

## Citation
(Note: I am not the author of the following papers)

[1]: Q. Wang, S. R. Kulkarni, and S. Verdú. Divergence estimation for multidimensional densities via k-nearestneighbor distances. IEEE Transactions on Information Theory, 55(5), 2009.
https://www.princeton.edu/~kulkarni/Papers/Journals/j068_2009_WangKulVer_TransIT.pdf

[2]: F. Perez-Cruz, Kullback-Leibler divergence estimation of continuous distributions. 2008 IEEE International Symposium on Information Theory, Toronto, ON, Canada, 2008, pp. 1666-1670, doi: 10.1109/ISIT.2008.4595271.
https://www.tsc.uc3m.es/~fernando/bare_conf3.pdf

[3]: R. Jenssen, J. C. Principe, D. Erdogmus, T. Eltoft, The Cauchy–Schwarz divergence and parzen windowing: Connections to graph theory and mercer kernels, Journal of the Franklin Institute 343 (6) (2006) 614–629.
https://www.sciencedirect.com/science/article/abs/pii/S0016003206000767


================ AFTER REBUTTAL & DISCUSSION ================

The authors diligently worked to enhance their paper, successfully addressing certain concerns, such as the comparison of a non-parametric estimator.
Considering the substantial revisions and additional analysis, I am of the opinion that the manuscript should undergo another round of peer review to validate these changes.
I raised my score, but with the perspective that it is just below the acceptance borderline.

**Details Of Ethics Concerns:**

I believe that this work does not raise any ethical concerns because it is a methodological study focused on information bottleneck.

---

> ### Author Response · Authors · 2023-11-22
> **Reply to your question 1 and question 2**
>
> First, we would like to appreciate your careful review about our submission, and raise some insightful questions and suggestions. In the rebuttal, we systematically analyze the asymptotic property of our CS divergence estimator (please refer to **Appendix B.4**). We also analyze the sources of bias of previous variational KL divergence-based deep information bottleneck approaches (please refer to **Appendix B.2**).  The advantage and distinctions of CS divergence over traditional distance measures like KL divergence is further introduced in *Appendix B.1*.
>
> Moreover, per your request, we strengthened our theoretical results. First, we relax Gaussian assmption on the Theorem 1, we show in the **Appendix B.5** that the relationship between CS divergence and KL divergence still holds for two arbitrary square-integral densities. An empirical justification is also provided. Second, we add Proposition 4, showing that our CS-QMI, i.e., $I_{CS}(x;t)$ bounds the expected value of $|h_\theta(x+\delta)-h_\theta(x)|$ with a probability of nearly $1$ (given sufficient samples). We also provide the assumptions under which this proposition holds.
>
> Below, please let us reply to your questions one by one. We will then address your other concerns also one by one.
>
> ## [Q1: non-parametric estimator of KL divergence]
> Yes, the reviewer is correct that the conventional information bottleneck (IB) objective based on the KL divergence (i.e., Eq.~(16)) can also be evaluated non-parametrically with kernel density estimator (KDE).
> $D_{KL} (p(y|x);q(\hat{y}|x)) + \beta I(x;t)$.
>
> The above objective actually equals to (by the chain rule of KL divergence):
> $D_{KL} (p(y,x);q(\hat{y},x)) + \beta I(x;t)$
>
> The first term can be estimated by KDE with:
> $D_{KL} (p(y,x);q(\hat{y},x)) = \frac{1}{N} \sum_{i=1}^N \log \left( \frac{p(x_i,y_i)}{q(x_i,\hat{y}_i)} \right)$
>
> In a Gaussian noise channel, $t=f(x)+\epsilon$ and $\epsilon \sim N(0,\sigma^2I)$. Hence, the second term equals to:
> $I(x;t)=H(t)-H(t|x)=H(t)-H(\epsilon)=H(t)-d(\log \sqrt{2\pi}\sigma + 1/2)$,
> in which $H(t)$ can also be measured with KDE.
>
> We discussed this baseline in our revised submission. Please refer to our discussions on **Appendix B.3** (especially the last paragraphs marked with blue). Unfortunately, we found that KL divergence estimated in a non-parametric way is hard to convergence or achieve a compelling performance.
>
> One possible reason is perhaps the support constraint on the KL divergence to obtain a finite value: there is no guarantee that support of $q_\theta$ is a subset of the support of $p$. Our CS divergence does not have this issue, it relaxes the constraint on the support (unless two supports have no overlap). Please also refer to our discussion on **Appendix B.1**, in which we also plot Fig. 5 to clarify this point.
>
>
> ## [Q2: CS divergence is not a special case of the well-known Renyi’ divergence]
>
> First, we would like to make a clarification that the CS divergence is not a special case of the well-known Renyi’s divergence developed by Renyi with the following expression:
> $D_\alpha(p;q) = \frac{1}{\alpha-1}\log\int p^\alpha q^{1-\alpha}du=\frac{1}{\alpha-1}\log\int p\left(\frac{q}{p}\right)^{1-\alpha}du=\frac{1}{\alpha-1}\log\mathbb{E}_p\left(\frac{q}{p}\right)^{1-\alpha}$
>
> When $\alpha=2$, we obtain a new divergence $-\log(1-\chi^2(p;q))$ that is proportional to the $\chi^2$ divergence, rather than our CS divergence mentioned in the paper.
>
> To our knowledge, there are two big ideas to define divergence: the integral probability metrics (IRM) aim to measure $p-q$, whereas the Renyi’s divergences with order $\alpha$ aim to model the ratio of $\frac{p}{q}$.
> The general idea of CS divergence is totally different: it neither horizontally computes $p-q$ nor vertically evaluates $\frac{p}{q}$.  Rather, it quantifies the “distance” of two distributions by quantify the tightness (or gap) of an inequality associated with integral of densities (i.e., $\int pq$). That is, CS divergence does not belong to either IRM or the traditional Renyi’s divergence family. Please refer to our discussion on **Appendix B.1** for more details.
>
> Our CS divergence belongs to a special case of the generalized divergence defined by Lutwak [Lutwak et al., 2005], which uses a modification of the Holder inequality. Lutwak et al. still use the name of “Renyi” maybe because their divergence also provides a one-parameter generalization of the KL divergence. But Lutwak’s definition is totally different from the traditional Renyi’ divergence.
>
> To our knowledge, the use of the above Holder divergence or CS divergence is very scarce in the community. However, this family of divergence indeed has many desirable properties (please also refer to our discussion in **Appendix D**). We choose the most popular one in this new family of divergence in our paper. We also revise some relevant descriptions to avoid confusion.

---

> ### Author Response · Authors · 2023-11-22
> **Reply to your questions 3-7**
>
> ## [Q3: Fundamental difference with respect to KDE estimator in [3]].
>
> Thanks for pointing out this good question; also sorry for missing this relevant discussion in the submission. Let us take the (marginal) CS divergence as an example, which can be expressed as:
> $D_{cs}(p;q)=\log\int p(x)^2dx+\log\int q(x)^2dx-2\log\int p(x)q(x)dx$
>
> We focus our discussion on the first term inside the log, i.e. $\int p^2(x)dx$.
>
> In fact, we estimate $\int p^2(x)dx$ as $\mathbb{E}_p(p(x))$. which is then estimated by $\frac{1}{N} \sum_i^N \hat{p}(x_i)$. This estimator is also called the **resubstitution estimator**; whereas [3] uses the **plug-in estimator** that inserting KDE of $p(x)$ into $p^2(x)$. That is, they estimate $\int p^2(x)dx$ as $\int\hat{p}^2(x_i)$, and use the property that “the integral of the product of two Gaussians is exactly evaluated as the value of the Gaussian computed at the difference of the arguments and whose variance is the sum of the variances of the two original Gaussian functions”. To our knowledge, this is a unique property that only holds for Gaussian kernel.
>
> Please also refer to our discussion in **Appendix B.4**, especially the sentences marked with brown.
>
> Based on the above analysis, we summarize the following two big differences:
>
> 1) although the final expression of CS divergence is similar. Our resubstitution estimator can be proven consistent (see also refer to Appendix B.4 in the revised submission), such property is missing in [3].
>
> 2) Moreover, our method is more general that is not restricted to Gaussian kernel (we do not to use any property associated with kernel function, such as convolution of two Gaussian).
>
> ## [Q4: asymptotic properties of our estimator]
> Yes, the asymptotic property of our estimator can be guaranteed. Pease refer to our **Appendix B.4**.
>
> We start our analysis on the basic CS divergence:
> $D_{cs}(p;q)=\log\int p(x)^2dx+\log\int q(x)^2dx-2\log\int p(x)q(x)dx$
>
> Proof sketch: we first provide the asymptotic property of each term inside the “log” operator. We obtained that for $\int p^2(x)dx$, $\int q^2(x)dx$ and $\int p(x)q(x)dx$, our empirical estimator combined with KDE has the asymptotic mean integrated square error (AMISE) tends to zero when the kernel size $\sigma$ goes to zero and the number of samples $N$ goes to infinity with $N^2\sigma\rightarrow\infty$. That is, each term has a consistent estimator. We then show that, the “log” operator does not influence the asymptotic property.
>
> ## [Q5: Why performances of existing methods match perfectly in Table 2?]
> First, there is no bug in our code. The performances of all existing methods are the same when the compression ratio $r=0$, i.e., no compression term exists in the IB objective, i.e., $\beta=0$. In this case, all existing methods just minimize a MSE loss (without any regularization on $I(x;t)$) with the same network architecture. Hence, their performances are the same.
>
> Our CS-IB outperforms consistently when $r=0$. This results also demonstrate the advantage of using $D_{CS} (p(y|x);q_\theta(\hat{y}|x))$ alone over the traditional MSE loss.
>
> ## [Q6: the standard deviation of results]
> We provide the full results which include standard deviation (over 5 independent runs) in Tables 8 and 9 in **Appendix C.4**. Due to page limit, we only demonstrate the mean value in main paper.
>
> ## [Q7: How hyperparameters in Table 4 were selected?]
> For each competing method, the hyperparameters are selected as follows. We first choose the default values of $\beta$ (the balance parameter to adjust the importance of the compression term) and the learning rate $lr$ mentioned in its original paper. Then, we select hyperparameters within a certain range of these default values that can achieve the best performance in terms of RMSE. For VIB, we search within the range of $\beta \in [10^{-3}, 10^{-5}]$. For NIB, square-NIB, and exp-NIB, we search within the range of $\beta \in [10^{-2}, 10^{-5}]$. For HSIC-bottleneck, we search $\beta \in [10^{-2}, 10^{-5}]$. For our CS-IB, we found that the best performance was always achieved with $\beta$ between $10^{-2}$ to $10^{-3}$. Based on this, we sweep the $\beta$ within this range and select the best one. The learning rate range for all methods was set as $[10^{-3}, 10^{-4}]$. We train all methods for $100$ epochs on all datasets except for $PM2.5$, which requires around $200$ epochs of training until converge. All the hyperparameter tuning experiments are conducted on the validation set.

---

> ### Author Response · Authors · 2023-11-22
> **On your other concerns, such as the mismatch of the claim, the lack of theoretical guarantee, and the reliability of the results**
>
> # [On the misalignment between the claims of contribution]
> ## [P1: How accurate is the claim?]
> Thanks for raising such a good question. More precisely, we feel that the independence to variational approximation and distributional assumption can be attributed to the estimation of CS divergence in a non-parametric way. Per your request, we analyzed how to non-parametrically implement KL divergence-based IB approach, from which we found the model is hard to train (see our reply to your [Q1]). We also point out that one possible reason why the non-parametric estimation of CS divergence successes is maybe because CS divergence relax the constraint on the support of distributions, in which we seldom has the situation $p(x) \log(\frac{p(x)}{0}) = \infty$.
>
> ## [P2: Extending Theorem 1 to general distributions without Gaussian assumptions]
> Theorem 1 can be generalized to arbitrary square-integral distributions without Gaussian assumptions. Please refer to Proposition 5 in the **Appendix B.5** of the revised submission.
>
> ## [P3: Strengthen the justification on adversarial robustness]
> Thanks for your suggestion. We added Proposition 4 in the revised submission, in which we bound demonstrate that the probability that CS-QM, i.e., $I_{CS}(x;t)$ bounds the expected value of $|h_\theta(x+\delta)-h_\theta(x)|$ is at least $0.5$ and can reach to nearly $1$ (given sufficient samples). We also provide the assumptions under which this proposition holds.
>
> # [A lack of theoretical guarantees on the properties]
> ## [P1&P2 The bias from non-parametric estimator and variational approximation]
> First, thanks for raising such a good question. The bias and asymptotic property of our estimator has been discussed in **Appendix B.4**. The bias of existing variational approximation is also discussed in **Appendix B.2** in the revised submission. In general, our argument is that previous methods either have much more obvious bias than ours or similar bias to us. We also provide a toy example in **Appendix C.5** to show CS divergence estimated non-parametrically performs better than the variational KL divergence.
>
> ## [P3 Justification of correlation with generalization performance in real-world data]
> Thanks for the good suggestion. We justified the positive correlation on California housing data and also trained nearly 100 models with various depth, width, or mini-batch size. The results are demonstrated in Table 1 and Fig. 12. The experimental details are described in **Appendix C.1**.
>
> # [Concerns on regarding the reliability of results]
> We added the standard deviation in Tables 8 and 9 in **Appendix C.4**. We explained in reply to your [Q5], there is no bug in the implementation. We also explained how hyperparameters in Table 4 are obtained in reply to your [Q7]. Please also refer to **Appendix C.2**.
>
> Overall, many thanks for your careful reading of our submission. We hope that our reply and the revised submission can address your concerns.

---

> > ### Comment · Reviewer_MhfL · 2023-11-23
> > **Acknowledgement**
> >
> > Thank you for your considerate reply to my concerns. I've carefully examined all the responses and briefly reviewed the revised drafts. Nevertheless, due to substantial revisions and additional analyses, I am of the opinion that the manuscript should undergo another round of peer review to validate these changes. Consequently, while I plan to elevate the score, my outlook regarding the ICLR acceptance this time is somewhat cautious.
> >
> > While this might be a disappointing outcome for the authors, it's important to highlight that the revisions stemming from this peer review have significantly enhanced your paper, particularly in terms of soundness. Even if the paper doesn't secure acceptance from ICLR this time, I anticipate a high likelihood of acceptance in the subsequent opportunity.

---

### Official Review · Reviewer_zqzB · 2023-11-01

**Soundness:** 3 good
**Presentation:** 3 good
**Contribution:** 2 fair
**Rating:** 6
**Confidence:** 2

**Summary:**

The authors study information bottleneck (IB) methods for regression tasks using a new divergence (Cauchy-Schwarz). The authors showed how to algorithmically design an IB approach based on this new divergence, analyzed the theoretical properties of the new divergence, and numerically demonstrated a visible advantage over existing approaches.

**Strengths:**

The use of Cauchy-Schwarz divergence in information bottleneck approaches is reasonable and novel to my understanding.

The authors derived an efficient algorithm for training IB approaches based on the Cauchy-Schwarz divergence.

The authors demonstrate visible improvement over existing approaches.

**Weaknesses:**

The improvement over existing approaches is fairly limited. In many tasks, the improvement is as small as 0.1 RMSE (where the relative improvement is close to 0.01~0.04). I am not sure if the limited improvement on these datasets is particularly meaningful.

It remains conceptually unclear to me why we want to use the Cauchy-Schwarz divergence. It is shown to be always <= KL divergence, but it is not clear how much smaller would this be (maybe only minimally).

**Questions:**

- Could the authors provide a high-level idea/motivation in the Introduction to describe why one should consider the Cauchy-Schwarz divergence instead of the more traditional KL divergence? Why is it better to avoid variational approximations?

- Are there toy problems where Cauchy-Schwarz divergence performs significantly better than other traditional approaches?

---

> ### Author Response · Authors · 2023-11-22
> **Reply to weaknesses and questions**
>
> First, thanks for your positive comments. Please let us reply to your concerns one by one.
>
> ## [Q1: minor performance improvement]
> Yes, on some relative simpler data, our prediction performance improvement seems a bit marginal. However, we would like to emphasize the obvious performance improvement on the adversarial robustness in Table 3.
>
> Moreover, in the rebuttal period, we additionally added a new application: to predict the age of patients based on their brain functional MRI (fMRI) data with a graph neural network (GNN), which, to the best of our knowledge, is seldom investigated (prevalent literatures always rely on structural MRI (sMRI) data with convolutional neural networks) and is a much more challenging task. Our new results in  **Appendix D** also demonstrate an obvious performance gain. Moreover, the success of our method in GNN also showcase the broader applicability and generalizability of our estimator.
>
> ## [Q2: The gap between CS divergence and KL divergence]
> $D_{KL}(p;q)$ is easy to reach infinity if the support of $p$ is not a subset of the support of $q$, which means that $D_{KL}(p;q)$ only has finite values if $\text{supp}(p) \subseteq \text{supp}(q)$.
> In fact, even if the support of $p$ and $q$ has large overlap, but $\text{supp}(p) \not\subset \text{supp}(q)$ (this situation is common in practice), KL is still infinity. Given two square integral densities $p$ and $q$, its value is infinite for CS divergence if and only if there is no overlap on the supports, i.e., $\text{supp}(p) \cap \text{supp}(q) = \emptyset$. In this sense, the CS divergence actually relaxes the support constraint of the KL divergence. Please also refer to our **Appendix B.1** for more details.
>
> ## [Q3: Why Cauchy-Schwarz divergence]
> Thanks for the good question.
>
> On the one hand, in **Appendix B.2**, we show that the approximation error of variational approximation is hard or intractable to control, or obvious larger than ours. By contrast, our estimation error and other properties can be fully analyzed as shown in **Appendix B.4**. That is, our estimator provides better theoretical guarantee.
>
> On the other hand, some intrinsic limitations of KL divergence, such as support constraint, also motivates us to use other divergence. However, we would also like to emphasize here that the CS divergence also has some other desirable mathematical properties over KL divergence (as has been discussed in **Appendix D**), which we feel should be interesting and helpful to community.
>
> ## [Q4: Toy examples where CS divergence performs slightly better]
> Thanks for the good question. We consider two toy examples in the revised submission in **Appendix C.5**. The first example demonstrates when the true underlying distribution is a mixture of Gaussian, our CS divergence optimization performs better than the reverse KL divergence with variational approximation. The second example demonstrates that CS-QMI (the Cauchy-Schwarz quadratic mutual information) has similar or slightly better ability to identify complex dependence patterns compared with popular approaches like HSIC and distance covariance (dCov).

---

### Official Review · Reviewer_SV68 · 2023-11-01

**Soundness:** 4 excellent
**Presentation:** 3 good
**Contribution:** 3 good
**Rating:** 6
**Confidence:** 4

**Summary:**

The authors of this paper introduce a novel approach to regression using the IB principle, leveraging the Cauchy-Schwarz divergence for parameterization. This departure from MSE-based regression eliminates the need for variational approximations or distributional assumptions, leading to improved generalization and strong adversarial robustness. Their CS-IB method outperforms other deep IB approaches across six real-world regression tasks, achieving an optimal balance between prediction accuracy and compression ratio in the information plane.

**Strengths:**

- The authors show the connections between CS divergence to MMD and HSIC.
- The effect of CS divergence to generalization and adversarial robustness is well quantified.
- Thorough discussions are provided for most of remarks or theoretic findings.

**Weaknesses:**

- The CS divergence estimation is based on the Gaussian kernel assumption, which will depend on the parameter $\sigma$. What is the effect of $\sigma$ to the IB performance is not shown clearly.
- The KL IB using variational approach is friendly to optimization based methods (gradient-based approaches). On the other hand, CS IB method is based on Gaussian kernel assumption, which may require the tuning of $\sigma$.
- I think it’s better to have a section of identifying what are some potential disadvantages of CS-IB method compared to KL-IB method.

**Questions:**

The complexity of CS-divergence estimation approach is O(N^2). How is it compared to that of the traditional KL divergence based IB method?

---

> ### Author Response · Authors · 2023-11-22
> **Reply to weaknesses and questions**
>
> First, thanks for your positive comments. Please let us reply to your concerns one by one.
>
> ## [Q1 The computational complexity of traditional KL-divergence based IB method]
> Thanks for the good question. For HSIC-bottleneck, the dependence between $x$ and $t$ is measured with HSIC, which also takes a computational complexity of $O(N^2)$. For nonlinear information bottleneck (NIB) and square-NIB and exp-NIB, $I(x;t)$ is upper bounded as:
> $I(x;t) \leq -\frac{1}{N} \sum_{i=1}^N \log \frac{1}{N} \sum_{j=1}^N \exp\left( -\frac{\|h_i-h_j\|_2^2}{2\sigma^2} \right)$.
> in with $h$ is the hidden activity corresponding to each input sample, i.e., $t$ equals $h$ plus Gaussian noise (to avoid infinite mutual information value). Hence, the complexity is also $O(N^2)$.
>
> ## [Q2 The effect of $\sigma$ to IB performance]
> We demonstrate in Fig. 19 in the **Appendix C.4**, showing that our estimators provide consistent performance gain in a reasonable range of kernel size.
>
> Additionally, we also demonstrate in **Appendix B.4** that the effects of $\sigma$ to the bias and variance of our CS divergence estimator. The conclusion is that estimation bias is shrinking at a rate of $O(\sigma^2)$, and the variance shrinks at rate $O(1/(N^2\sigma))$. So there is a trade-off between bias and variance, which makes sense.
>
> ## [Q3 The tuning of hyperparameter $\sigma$]
> We partially agree with this point. For the basic variational information bottleneck (VIB) based on KL divergence, it is indeed very friendly for optimization. However, we would like to emphasize here that, for advanced KL-based IB approaches such as NIB, square-NIB and HSIC-bottleneck, they all require a careful treatment on the choice of some hyperparameter similar to kernel size $\sigma$. Different to these approaches, we systematically analyzed how $\sigma$ affects the bias and variance of our estimator in **Appendix B.4**. To our knowledge, such analysis is missing in previous literature.
>
> ## [Q4 Potential disadvantage of CS-IB]
> Thanks for the good question. We would like to mention a limitation of CS divergence with respect to the KL divergence. That is, we did not find a dual representation for the CS divergence. The dual representation of KL divergence, such as the well-known Donsker-Varadhan representation, enables the use of neural networks to estimate the divergence itself, which significantly improves estimation accuracy in high-dimensional space. A notable example is the mutual information neural estimator (MINE). We leave it as a future work and mention this point in **Appendix D**.

---

### Meta-Review · Area_Chair_Y68c · 2023-12-02

**Metareview:**

This paper considers Cauchy-Schwarz (CS) divergence in Information Bottleneck. Efficient algorithm for for training with CS divergence has been provided, generalisation and robustness issues have been discussed. I believe the result would of great interest of ICLR audience and benefit the community. After vibrant discussions, most of the reviewer's concerns have been resolved. I would recommend to accept the paper. However, as reviewer MhfL pointed out, the discussion has extensively improved the content of the manuscript and I would strongly recommend the authors integrate the discussion points into the revised manuscript to improve the paper further.

**Justification For Why Not Higher Score:**

Additional clarification and modification on discussion points are needed.

**Justification For Why Not Lower Score:**

The results presented are interesting enough for the ICLR audience.

---

### Decision · Program_Chairs · 2024-01-16

Accept (poster)